# Safe Meta-Reinforcement Learning via Dual-Method-Based Policy Adaptation: Near-Optimality and Anytime Safety Guarantee

## Abstract

This paper studies the safe meta-reinforcement learning (safe meta-RL) problem where anytime safety is ensured during the meta-test. We develop a safe meta-RL framework that consists of two modules, safe policy adaptation and safe meta-policy training, and propose efficient algorithms for the two modules. Beyond existing safe meta-RL analyses, we prove the anytime safety guarantee of policy adaptation and provide a lower bound of the expected total reward of the adapted policies compared with the optimal policies, which shows that the adapted policies are nearly optimal. Our experiments demonstrate three key advantages over existing safe meta-RL methods: (i) superior optimality, (ii) anytime safety guarantee, and (iii) high computational efficiency.

## 1 Introduction

Reinforcement learning (RL) (Sutton & Barto, 2018) has achieved significant successes in various domains, from video games (Mnih et al., 2015; Silver et al., 2016; Lee et al., 2018) to robotics (Levine et al., 2016; Lee et al., 2020; Margolis et al., 2021; 2024). The RL problem is formulated as a Markov decision process (MDP) and aims to maximize the expected total reward. Safe RL (Yu et al., 2019; Xu et al., 2021; Ding et al., 2021; Yu et al., 2022) addresses additional safety requirements, such as collision avoidance for robots (Xu & Zhu, 2022; Huang, 2021) and operation restrictions in financial management (Abe et al., 2010). Typically, the safe RL problem is formulated as a constrained MDP (CMDP) (Altman, 2021), which aims to maximize the expected total reward while ensuring that the expected safety costs are below given thresholds. As noted in (Ding et al., 2021), the goals of reward maximization and constraint enforcement are not completely aligned, aggravating the challenge of the inherent trade-off between exploration and exploitation.

Meta-reinforcement learning (meta-RL) (Beck et al., 2023) aims to extract common knowledge from multiple existing RL tasks, accelerating the learning process and increasing the data efficiency of RL algorithms. Safe meta-RL (Khattar et al., 2023; Xu et al., 2021) integrates safe RL and meta-RL and inherits the benefits of both. On the other hand, existing safe meta-RL methods face three new challenges: optimality, computational efficiency, and anytime safety. Meta-CRPO (Khattar et al., 2023) considers an online safe meta-RL problem. In each round, it computes the task-specific policy by CRPO (Xu et al., 2021) and updates the meta-policy that has the minimal average distance to the task-specific policies of all previous tasks. However, the meta-training does not optimize the performance of the task-specific policy adaptation, and the policies adapted from the learned meta-policy may be sub-optimal for new tasks. Meta-CPO (Cho & Sun, 2024) optimizes the policies adapted from the meta-policy by constraint policy optimization (CPO) (Achiam et al., 2017). Nevertheless, its computational complexity is high in both the meta-training and meta-test stages. Specifically, during the meta-training, meta-CPO aims to solve a constrained bilevel optimization problem (Xu & Zhu, 2023a) where the constraints are present at both the upper level and lower level. It requires to compute the inverse of Hessian, which is computationally expensive. During the meta-test, each policy adaptation step solves a nonconvex constrained optimization problem.

In applications of (safe) meta-RL (Nagabandi et al., 2018; Belkhale et al., 2021), during the meta-test, the agent collects the rewards/costs of state-action pairs by exploring a new, unknown CMDP and optimizes the policy based on the collected data. Therefore, it is important to guarantee anytime

Table 1: Comparison with existing safe meta-RL methods

| Methods | Theoretical results | | | Experimental results | |
| | Safety | | Bounded optimality gap | Efficiency | Optimality |
| | Constraint violation | Target policy | | | |
| --- | --- | --- | --- | --- | --- |
| (Khattar et al., 2023) | Positive | Safety for final policy | ✓ | Low | Low |
| (Cho & Sun, 2024) | Positive | Safety for adapted policy | × | Low | Medium |
| This paper | Zero | Anytime safety | ✓ | High | High |

safety, i.e., the safety constraints must be satisfied for every policy used for the exploration. However, the anytime safety is overlooked in all existing safe meta-RL algorithms (Khattar et al., 2023; Cho & Sun, 2024). Specifically, during the meta-test, they start with the meta-policy and repeatedly adapt the most recent policy into a new one by the policy adaptation algorithm, which generates a sequence of policies. Except for the final policy in the sequence, each policy, including the initial meta-policy, is used to explore the environment and collect data. Meta-CRPO (Khattar et al., 2023) only quantifies the safety constraint violation of the final convergent policy in the sequence, neglecting that of intermediate policies for data collection. Meta-CPO (Cho & Sun, 2024) applies the CPO (Achiam et al., 2017) as the policy adaptation algorithm, which can quantify the safety constraint violation of policies that have undergone at least one adaptation step. However, the safety of the meta-policy is ignored. Moreover, both meta-CRPO and meta-CPO provide positive upper bounds of the constraint violation, which do not guarantee zero violation of the safety constraints.

**Main contribution.** In this paper, we develop a safe meta-RL framework consisting of two modules: safe policy adaptation and safe meta-policy training. Specifically, the safe policy adaptation is to maximize an approximate accumulated reward function under approximate constraint functions. The safe meta-policy training is to maximize the meta-objective function of the meta-policy, i.e., the expected accumulated reward of the task-specific policies adapted from the meta-policy, while the meta-policy satisfies the safety constraints. Then, we derive efficient algorithms for the two modules. In particular, to solve the safe policy adaptation, we derive its close-formed solution under certain Lagrangian multipliers, and propose a dual-method-based algorithm to solve the multipliers. For the safe meta-policy training, we derive the meta-gradient, i.e., the gradient of the meta-objective, simplify its computation by exploiting the softmax form of the adapted policy, and propose a Hessian-free meta-training algorithm.

The proposed algorithms offer three key advantages over existing safe meta-RL methods. (i) **Superior optimality.** Our safe meta-policy training algorithm maximizes the expected accumulated reward of the policies adapted from the meta-policy, and then improves the optimality of meta-CRPO (Khattar et al., 2023) and naive transfers from meta-RL, which do not consider the task-specific safe policy adaptation in the meta-training. (ii) **Anytime safety guarantee** during the meta-test. The safe meta-policy training produces a safe initial meta-policy by imposing the safety constraint on it. The safe policy adaptation imposes a constraint on the upper bound of the total cost, and thus is guaranteed to produce a safe policy for each iteration when the initial policy is safe. By integrating these two modules, anytime safety is achieved. (iii) **High computational efficiency** in both the meta-test and meta-training stages. In the meta-test, the derivation of the close-formed solution makes it much more efficient than those in meta-CRPO (Khattar et al., 2023) and meta-CPO (Cho & Sun, 2024), which solve constrained optimization problems. In the meta-training, the close-formed solution of the policy adaptation is used to derive a Hessian-free meta-gradient and reduces the computation complexity of the proposed algorithm to approach that in the single-level optimization, making it more efficient than meta-CPO (Cho & Sun, 2024) and many meta-RL algorithms (Finn et al., 2017; Liu et al., 2019b) with the bi-level optimization steps and the computation of Hessian and Hessian inverse. We conduct experiments on seven scenarios including navigation tasks with collision avoidance and locomotion tasks to verify these advantages of the proposed algorithms.

Another major contribution of the paper is that it is the first to derive a comprehensive theoretical analysis regarding near optimality and anytime safety guarantees for safe meta-RL. First, we establish the theoretical basis of the algorithm design that guarantees anytime safety, i.e., zero constraint violation for any policy used for exploration. Second, we derive a lower bound of the expected accumulated reward of the adapted policies compared to that of the task-specific optimal policies, which shows the near optimality of the proposed safe meta-RL framework. Finally, we demonstrate a trade-off between the optimality bound and constraint violation when the allowable constraint violation varies, which enables the algorithm to be adjusted to prioritize either safety or optimality.

Table 1 compares both the theoretical and experimental results between this paper and previous works (Khattar et al., 2023; Cho & Sun, 2024). First, this paper considers the anytime safety and provides a zero constraint violation guarantee. In previous works, they only provided positive upper bounds for

the constraint violation, and the upper bounds only work for the final policy (Khattar et al., 2023) or the adapted policies (Cho & Sun, 2024). Second, although (Khattar et al., 2023) provides an upper bound of the optimality gap, the experimental optimality is the worst. On the other hand, (Cho & Sun, 2024) does not provide an optimality bound. In contrast, the proposed method exhibits high optimality and provides a near-optimality guarantee, outperforming existing approaches in terms of both experimental and theoretical outcomes. Third, the proposed method is more efficient than the existing approaches (Khattar et al., 2023; Cho & Sun, 2024).

**Related works.** Due to the space limit, we include a section of related works in Appendix A.

## 2 PROBLEM STATEMENT

**CMDP.** A CMDP $\mathcal{M} \triangleq \{\mathcal{S}, \mathcal{A}, \gamma, \rho, P, r, \{c_i\}_{i=1}^p, \{d_i\}_{i=1}^p\}$ is defined by the state space $\mathcal{S}$, the action space $\mathcal{A}$, the discount factor $\gamma$, the initial state distribution $\rho$ over $\mathcal{S}$, the transition probability $P(s'|s,a) : \mathcal{S} \times \mathcal{A} \times \mathcal{S} \to [0,1]$, the reward function $r : \mathcal{S} \times \mathcal{A} \times \mathcal{S} \to [0, r^{max}]$, $p$ cost functions where the $i$-th cost function is defined as $c_i : \mathcal{S} \times \mathcal{A} \times \mathcal{S} \to [0, c_i^{max}]$ for $i = 1, \cdots, p$, and the constant $d_i$, which is the limit of constraint $i$. The state space $\mathcal{S}$ could be either a discrete space or a bounded continuous space. The action space $\mathcal{A}$ could be either discrete or continuous.

**Policy.** A stochastic policy $\pi : \mathcal{S} \to \mathbb{P}(\mathcal{A})$ is a mapping from states to probability distributions over action. When $\mathcal{A}$ is discrete, $\pi(a|s)$ denotes the probability of choosing action $a$ in state $s$; when $\mathcal{A}$ is continuous, $\pi(a|s)$ denotes the probability density. Denote the policy space as $\Pi$. In addition, a softmax policy parameterized by $\theta \in \mathbb{R}^n$ is denoted as $\pi_\theta$, where $\pi_\theta(a|s) \triangleq \frac{\exp(f_\theta(s,a))}{\int_{\mathcal{A}} \exp(f_\theta(s,a'))da'}$, $\forall(s,a) \in \mathcal{S} \times \mathcal{A}$, for continuous action space $\mathcal{A}$, or $\pi_\theta(a|s) \triangleq \frac{\exp(f_\theta(s,a))}{\sum_{a' \in \mathcal{A}} \exp(f_\theta(s,a'))}$, for discrete action space $\mathcal{A}$, and $f_\theta : \mathcal{S} \times \mathcal{A} \to \mathbb{R}$ is an approximation function.

**Safe RL.** For a policy $\pi$, the value function is defined as $V^\pi(s) \triangleq \mathbb{E}[\sum_{t=0}^\infty \gamma^t r(s_t, a_t, s_{t+1})|s_0 = s, \pi]$. The action-value function is defined as $Q^\pi(s,a) \triangleq \mathbb{E}[\sum_{t=0}^\infty \gamma^t r(s_t, a_t, s_{t+1})|s_0 = s, a_0 = a, \pi]$. The advantage function is defined as $A^\pi(s,a) \triangleq Q^\pi(s,a) - V^\pi(s)$. The accumulated reward function is $J(\pi) \triangleq \mathbb{E}_{s \sim \rho}[V^\pi(s)]$. Similarly, for each $i = 1, \cdots, p$, we define $V_{c_i}^\pi(s) \triangleq \mathbb{E}[\sum_{t=0}^\infty \gamma^t c_i(s_t, a_t, s_{t+1})|s_0 = s, \pi]$, $Q_{c_i}^\pi(s,a) \triangleq \mathbb{E}[\sum_{t=0}^\infty \gamma^t c_i(s_t, a_t, s_{t+1})|s_0 = s, a_0 = a, \pi]$, $A_{c_i}^\pi(s,a) \triangleq Q_{c_i}^\pi(s,a) - V_{c_i}^\pi(s)$, and $J_{c_i}(\pi) \triangleq \mathbb{E}_{s \sim \rho}[V_{c_i}^\pi(s)]$. The discounted state visitation distribution of $\pi$ is defined as $\nu^\pi(s) \triangleq (1 - \gamma)\mathbb{E}_{s_0 \sim \rho}[\sum_{t=0}^\infty \gamma^t \mathbb{P}(s_t = s|\pi)]$. The safe RL problem is to maximize the accumulated reward function while the accumulated cost functions satisfy the constraints, i.e., solving the problem $\max_{\pi \in \Pi} J(\pi)$ s.t. $J_{c_i, \tau}(\pi) \le d_i, \forall i = 1, \cdots, p$.

**Safe meta-RL with anytime safety.** Safe meta-RL targets multiple safe RL tasks. Consider a space of safe RL tasks $\Gamma$, where each task $\tau \in \Gamma$ is modeled by a CMDP $\mathcal{M}_\tau \triangleq \{\mathcal{S}, \mathcal{A}, \gamma, \rho_\tau, P_\tau, r_\tau, \{c_{i,\tau}\}_{i=1}^p, \{d_{i,\tau}\}_{i=1}^p\}$. Following the notions in the above subsections, the notations $\rho_\tau, P_\tau, r_\tau, c_{i,\tau}, d_{i,\tau}$, as well as $V_\tau^\pi, V_{c_i,\tau}^\pi, Q_\tau^\pi, Q_{c_i,\tau}^\pi, A_\tau^\pi, A_{c_i,\tau}^\pi, J_\tau, J_{c_i,\tau}$, and $\nu_\tau^\pi$ are defined for task $\tau$. Consider a set of safe RL tasks in $\Gamma$ following a probability distribution $\mathbb{P}(\Gamma)$. Safe meta-RL aims to learn the meta-prior from $\mathbb{P}(\Gamma)$ which can be used to train a policy for an unseen task $\tau_{new} \sim \mathbb{P}(\Gamma)$ by a small number of new environment explorations with anytime safety. In specific, during the meta-training, tasks can be sampled from $\mathbb{P}(\Gamma)$, i.e., $\{\tau_j\}_{j=1}^T \sim \mathbb{P}(\Gamma)$ and the tasks' CMDPs $\{\mathcal{M}_{\tau_j}\}_{j=1}^T$ can be explored. During the meta-test, a new task $\tau_{new}$ is given, and the agent explores the CMDP $\mathcal{M}_{\tau_{new}}$ and produces the task-specific policy. Note that we consider the meta-training to be an offline stage, e.g. done in simulated environments, the safety constraints may be violated. In contrast, the policies are deployed to practical environments during the meta-test. Any policy used to explore $\mathcal{M}_{\tau_{new}}$ or used to execute the task $\tau_{new}$ should satisfy the safety constraints.

## 3 SAFE META-RL FRAMEWORK

The proposed safe meta-RL framework aims to learn a meta-policy $\pi_\phi$ such that it can adapt to an unseen task with anytime safety guarantee. The framework includes two modules: safe policy adaptation and safe meta-policy training. During the meta-training, the task-specific policy $\pi^\tau$ for each training task $\tau$ is adapted from the meta-policy $\pi_\phi$ by using the safe policy adaptation. Then,

the meta-parameter $\phi$ is optimized by using the safe meta-policy training. During the meta-test, the learned meta-policy $\pi_\phi$ is adapted to new tasks by the safe policy adaptation.

We propose the safe policy adaptation in Section 3.1, which can address the issues of safety guarantee in (Khattar et al., 2023) and high computational complexity in (Cho & Sun, 2024), and propose the safe meta-policy training in Section 3.2 to obtain a safe and optimal meta-policy. The integration between these two modules ensures the anytime safety.

## 3.1 SAFE POLICY ADAPTATION

We first derive the optimization problem to achieve safe policy adaptation from the meta-policy. For task $\tau$, the policy $\pi^\tau$ is adapted from the meta-policy $\pi_\phi$ by the safe policy adaptation $\mathcal{A}^s$, which is defined by $\pi^\tau = \mathcal{A}^s(\pi_\phi, \Lambda, \Delta, \tau) \triangleq$

$$\underset{\pi \in \Pi}{\mathrm{argmax}} \; \mathbb{E}_{s \sim \nu_\tau^{\pi_\phi}, a \sim \pi(\cdot|s)} \left[ A_\tau^{\pi_\phi}(s, a) \right] - \lambda \, \mathbb{E}_{s \sim \nu_\tau^{\pi_\phi}} \left[ D_{KL} \left( \pi(\cdot|s) \| \pi_\phi(\cdot|s) \right) \right], \tag{1}$$

$$\text{s.t. } J_{c_i, \tau}(\pi_\phi) + \mathbb{E}_{\substack{s \sim \nu_\tau^{\pi_\phi} \\ a \sim \pi(\cdot|s)}} \left[ \frac{A_{c_i, \tau}^{\pi_\phi}(s, a)}{1 - \gamma} \right] + \lambda_{c_i} \, \mathbb{E}_{s \sim \nu_\tau^{\pi_\phi}} \left[ D_{KL} \left( \pi(\cdot|s) \| \pi_\phi(\cdot|s) \right) \right] \leq d_{i, \tau} + \delta_{c_i},$$

where $i = 1, \cdots, p$, $\Lambda \triangleq \{\lambda, \lambda_{c_1}, \cdots, \lambda_{c_p}\}$ and $\Delta \triangleq \{\delta_{c_1}, \cdots, \delta_{c_p}\}$ are the hyper-parameters of $\mathcal{A}^s$. The safe policy adaptation $\mathcal{A}^s$ in problem (1) is inspired by the derivation of CPO (Achiam et al., 2017), where both problem (1) and CPO aim to approximate the original safe RL problem. Specifically, the objective and constraint functions of problem (1) serve as upper bounds of the true objective and constraint functions $J_\tau(\pi)$ and $J_{c_i, \tau}(\pi)$ of the safe RL problem. More details about the upper bounds will be discussed in Lemma 1 of Section 5.1. More importantly, considering that the explorations for the task $\tau$ are limited, problem (1) only needs to collect state-action data points and evaluate $A_\tau^{\pi_\phi}$ for a single policy $\pi_\phi$, which keeps the same requirement of data collection as one-step of gradient ascent in MAML (Finn et al., 2017). Therefore, we denote $\mathcal{A}^s$, i.e., collecting data on the meta-policy and solving the optimal solution of problem (1) as the one step of the policy adaptation. Moreover, considering a single gradient ascent in MAML is usually insufficient to identify a policy with good performance and safety, $\mathcal{A}^s$ is to completely solve the problem (1).

The existence of the solution, the safety, and the monotonic improvement are guaranteed for $\mathcal{A}^s$. Specifically, when setting $\Delta = 0$, given that the meta-policy $\pi_\phi$ is safe for task $\tau$, i.e., $J_{c_i, \tau}(\pi_\phi) \leq d_{i, \tau}, \forall i = 1, \cdots, p$, for an appropriate hyper-parameter $\Lambda$, we have following properties: (i) the feasibility set of problem (1) is not empty; (ii) $\pi^\tau$ is safe for task $\tau$, i.e., $J_{c_i, \tau}(\pi^\tau) \leq d_{i, \tau}, \forall i = 1, \cdots, p$; (iii) the performance of $\pi^\tau$ is better than the meta-policy $\pi_\phi$, i.e., $J_\tau(\pi^\tau) \geq J_\tau(\pi_\phi)$. The complete statements and proofs of property (i) are shown in Proposition 1 of Section 4.1; properties (ii) and (iii) under selected hyper-parameter $\Lambda$ are shown in Section 5. Moreover, when the requirement of the constraint satisfaction is not strict, setting $\delta_{c_i} = 0$ for all $i$ in problem (1) may overly restrict the policy update step. To enhance the algorithm's flexibility, we set $0 \leq \delta_{c_i} \leq \delta_{max}$ as an allowable constraint violation in problem (1).

As mentioned in the above properties (ii) and (iii), both CPO and problem (1) can achieve policy improvement and safety guarantee. However, the computational complexity of directly solving CPO or the constrained optimization problem of (1) is high. CPO (Achiam et al., 2017) and meta-CPO (Cho & Sun, 2024) solve an approximate problem to mitigate the issue, but the computational complexity is still high, meanwhile the safety constraint violation cannot be avoided in theory and also usually appears in practice. In contrast, the safe policy adaptation in problem (1) is designed to have the closed-form solution under certain Lagrangian multipliers, and then can be efficiently solved by the dual method, which will be discussed in Section 4.1.

Note that problem (1), for the first time, simultaneously offers two key advantages: (a) constraint satisfaction guarantee for a single policy optimization step (policy optimization using data collected on a single policy), which enables anytime safety in each policy adaptation step during the meta-test, and (b) the closed-form solution, which significantly reduces the computational complexity of the meta-policy training. The details of the two benefits to the safe meta-RL problem will be discussed in Sections 4.1 and 5. Consequently, it is particularly well-suited for the safe meta-RL problem formulation. As the existing safe policy optimization algorithms, such as primal-dual-based algorithm in RCPO Tessler et al. (2018b), PPO-Lagrangian Ray et al. (2019), and CRPO Xu et al. (2021) used by meta-CRPO, do not hold any of these two benefits, and therefore (1) cannot be replaced by these

**algorithms.** Moreover, although some prior works (Zhang et al., 2020b; Liu et al., 2022) also derive closed-form solutions of safe policy optimization, safety cannot be guaranteed in each step. Instead, safety is only guaranteed for the final convergent policy, where the trust region size $\epsilon$ is reduced to 0.

## 3.2 SAFE META-POLICY TRAINING

We obtain the optimal meta-policy $\pi_{\phi^*}$ by solving the following optimization problem:

$$\max_{\phi} \mathbb{E}_{\tau \sim \mathbb{P}(\Gamma)}[J_{\tau}(\mathcal{A}^s(\pi_\phi, \Lambda, \Delta, \tau))], \text{ s.t. } J_{c_i,\tau}(\pi_\phi) \leq d_{i,\tau} + \delta_{c_i}, \forall i = 1, \cdots, p \text{ and } \forall \tau \in \Gamma. \quad (2)$$

Here, $\mathbb{E}_{\tau \sim \mathbb{P}(\Gamma)}[J_{\tau}(\mathcal{A}^s(\pi_\phi, \Lambda, \Delta, \tau))]$ is the meta-objective function and is defined by the expected accumulated reward after the parameter is adapted by the policy adaptation, which evaluates the optimality of the meta-policy $\pi_\phi$. We choose the constraints $J_{c_i,\tau}(\pi_\phi) \leq d_{i,\tau} + \delta_{c_i}, \forall i = 1, \cdots, p$ for any task $\tau$ (similar to problem (1), we set $\delta_{c_i}$ as the allowable error). There are two reasons to set the constraints. First, as shown in Proposition 1, $J_{c_i,\tau}(\pi_\phi) \leq d_{i,\tau} + \delta_{c_i}, \forall i = 1, \cdots, p$ is a sufficient condition for that the safe policy adaptation algorithm $\mathcal{A}^s(\pi_\phi, \Lambda, \Delta, \tau)$ has a solution, and further assure the safe meta-policy training (2) is well-defined. Second, the exploration of the CMDP by the meta-policy $\pi_\phi$ should be safe for each task $\tau$ to guarantee the initial policy of the policy adaptation is safe. As mentioned in Section 3.1, $\mathcal{A}^s(\pi_\phi, \Lambda, \Delta, \tau)$ is guaranteed to be safe for task $\tau$ when $\pi_\phi$ is safe, and iterative policy adaptation using $\mathcal{A}^s$ is guaranteed to be safe. Therefore, the anytime safety of the policy adaptation is guaranteed. Its formal statement is shown in Section 5.

## 4 ALGORITHM

This section introduces the efficient algorithmic solutions to solve problems (1) and (2), respectively.

### 4.1 DUAL METHOD FOR SAFE POLICY ADAPTATION

This section derives the dual method to solve problem (1) efficiently. As mentioned in Section 3.1, based on the design of problem (1), we can derive its closed-form solution under certain Lagrangian multipliers, and then solve the Lagrangian multipliers to obtain the overall solution. We first derive the closed-form solution of problem (1) and show its existence in the following proposition.

**Proposition 1.** *Suppose that the softmax policy $\pi_\phi$ satisfies $J_{c_i,\tau}(\pi_\phi) \leq d_{i,\tau} + \delta_{c_i}, \forall i = 1, \cdots, p$, the solution $\pi^\tau$ of the optimization problem (1) exists. Under certain mild constraint qualifications, there exists Lagrangian multipliers $\{u^*_{c_i,\tau}\}_{i=1}^p$ with $0 \leq u^*_{c_i,\tau} < \infty$, such that*

$$\pi^\tau(\cdot \mid s) \propto \exp(f_\phi(s, \cdot) + \eta^{-1}(A^{\pi_\phi}_\tau(s, \cdot) - \sum_{i=1}^p u^*_{c_i,\tau} A^{\pi_\phi}_{c_i,\tau}(s, \cdot))), \quad (3)$$

*for any $s \in \mathcal{S}$, where $\eta \triangleq \lambda + (1 - \gamma) \sum_{i=1}^p u^*_{c_i,\tau} \lambda_{c_i}$.*

The complete statement of Proposition 1 that includes the sufficient condition for the existence of $\{u^*_{c_i,\tau}\}_{i=1}^p$, as well as the proof of the proposition are shown in Appendix F.2.1. Proposition 1 shows that, when the meta-policy $\pi_\phi$ is softmax, the closed-form solution of the policy adaptation (1) is also softmax. The approximate function $f_\phi$ for the meta-policy $\pi_\phi$ is adapted to $f_\phi + \eta^{-1}(A^{\pi_\phi}_\tau - \sum_{i=1}^p u^*_{c_i,\tau} A^{\pi_\phi}_{c_i,\tau})$ of $\pi^\tau$. With this computation, the approximate function of $\pi^\tau$ can be directly obtained, which is much simpler than solving problem (1). More importantly, it can significantly reduce the computational complexity of the meta-gradient, which will be discussed in Section 4.2.

In addition, the closed-form solution in (3) implies the safe policy adaptation (1) can be reduced to the policy adaptation for an unconstrained MDP under the penalized reward function. Specifically, when we define a comprehensive reward function $\bar{r}_\tau \triangleq r_\tau - \sum_{i=1}^p u^*_{c_i,\tau} c_{i,\tau}$, then the term $A^{\pi_\phi}_\tau - \sum_{i=1}^p u^*_{c_i,\tau} A^{\pi_\phi}_{c_i,\tau}$ is the advantage function of $\pi_\phi$ for $\bar{r}_\tau$. This implies that problem (1) is equivalent to an unconstrained policy optimization problem, where the reward $r_\tau$ is penalized by the negative costs $-c_{i,\tau}$ and the weights of the cost penalty are given by the Lagrangian multiplier $u^*_{c_i,\tau}$ of (1).

**Proposition 2.** *Suppose the assumption in Proposition 1 holds. Let $\pi^u$ ($u \triangleq [u_1, \cdots, u_p]$) be the policy with $\pi^u(\cdot|s) \propto \exp(f_\phi(s, \cdot) + (\lambda + (1 - \gamma) \sum_{i=1}^p u_i \lambda_{c_i})^{-1}(A^{\pi_\phi}_\tau(s, \cdot) - \sum_{i=1}^p u_i A^{\pi_\phi}_{c_i,\tau}(s, \cdot)))$. Then, the Lagrangian multipliers $\{u^*_{c_i,\tau}\}_{i=1}^p$ in (3) is the solution of the dual problem of (1), i.e.,*

$$\min_{u \in \mathbb{R}^p_{\geq 0}} \mathbb{E}_{\substack{s \sim \nu^{\pi_\phi}_\tau \\ a \sim \pi^u}}[(A^{\pi_\phi}_\tau - \sum_{i=1}^p u_i A^{\pi_\phi}_{c_i,\tau})(s, a) - D_{KL}(\pi^u(\cdot|s) \| \pi_\phi(\cdot|s))] + \sum_{i=1}^p u_i d'_{i,\tau}, \quad (4)$$

*where $\eta^u \triangleq \lambda + (1-\gamma)\sum_{i=1}^p u_i\lambda_{c_i}$ and $d'_{i,\tau} \triangleq (1-\gamma)(d_{i,\tau} + \delta_{c_i} - J_{c_i,\tau}(\pi_\phi))$.*

Proposition 2 shows the derivation of the Lagrangian multiplier $u^*_{c_i,\tau}$. Its proof is shown in Appendix F.2.2. With $u^*_{c_i,\tau}$, the solution of safe policy adaptation (1) can be obtained immediately by Proposition 1. Note that problem (4) is the dual problem of (1) with the closed-form $\pi^u$ for any dual variable $u$, which enables us to use the dual method to solve problem (4). Next, we provide the algorithm of solving problem (4) and its computational complexity analysis.

Algorithm 1 states the algorithm for the safe policy adaptation. We apply the projected gradient descent (PGD) to solve the optimization problem (4) to obtain the Lagrangian multipliers $\{u^*_{c_i,\tau}\}_{i=1}^p$, then the closed-form solution of problem (1) is immediately obtained. The gradient of the objective function $\bar{L}(u)$ of problem (4) w.r.t $u$ (used in line 4 of Algorithm 1) can be stated as

$$\nabla_{u_i}\bar{L}(u) = -\mathbb{E}_{s\sim\nu_\tau^{\pi_\phi}}[\mathbb{E}_{a\sim\pi^u(\cdot|s)}[A^{\pi_\phi}_{c_i,\tau}(s,a)] + (1-\gamma)\lambda_{c_i}D_{KL}(\pi^u(\cdot|s)\|\pi_\phi(\cdot|s))] + d'_{i,\tau}, \quad (5)$$

where $\pi^u$ and $d'_{i,\tau}$ are defined in Proposition 2, and then the gradient step is projected to $\mathbb{R}^p_{\geq 0}$. The computation in (5) is derived based on the dual method shown in Proposition 6.1.1 in (Bertsekas, 1997), which is simplified compared with direct computation by the chain rule. The derivation is shown in Appendix F.2.3. As the optimization problem (4) is the dual problem of (1) and is always convex, the PGD method in Algorithm 1 can guarantee convergence to the global optimum (Iusem, 2003). Due to the low dimensionality of the decision variables of problem (4) (the dimension of the Lagrangian multipliers $\{u^*_{c_i,\tau}\}_{i=1}^p$ is the constraint number $p$) and the simplicity of gradient computation, the computational complexity of Algorithm 1 is much lower than directly solving problem (1). Other Lagrangian-based policy optimization algorithms, such as RCPO Tessler et al. (2018b) and PPO-Lagrangian Ray et al. (2019), have been used to solve safe RL. However, they are not suitable for this safe meta-RL problem. More discussion and the comparisons between the proposed dual method in (4), (5), and Algorithm 1 and the existing Lagrangian-based algorithms are shown in Appendix C.

---

**Algorithm 1** Dual-method-based safe policy adaptation

---

**Require:** Meta-policy $\pi_\phi$; Advantage functions $A^{\pi_\phi}_\tau$ and $A^{\pi_\phi}_{c_i,\tau}$; step size $\beta$.
1: $u_i = 0$ for all $i \in 1,\cdots,p$
2: **for** $n = 1,\cdots,N$ **do**
3:    Compute $\pi^u(\cdot|s) \propto \exp(f_\phi(s,\cdot) + (\lambda + (1-\gamma)\sum_{i=1}^p u_i\lambda_{c_i})^{-1}(A^{\pi_\phi}_\tau(s,\cdot) - \sum_{i=1}^p u_i A^{\pi_\phi}_{c_i,\tau}(s,\cdot)))$
4:    $u_i \leftarrow \max\{0, u_i - \beta\nabla_{u_i}\bar{L}(u)\}$ for each $i = 1,\cdots,p$, where $\nabla_{u_i}L(u)$ is shown in (5)
5: **end for**
6: $u^*_{c_i,\tau} = u_i$ for all $i = 1,\cdots,p$
7: $\pi^\tau(\cdot|s) \propto \exp(f_\phi(s,\cdot) + (\lambda + (1-\gamma)\sum_{i=1}^p u^*_{c_i,\tau}\lambda_{c_i})^{-1}(A^{\pi_\phi}_\tau(s,\cdot) - \sum_{i=1}^p u^*_{c_i,\tau}A^{\pi_\phi}_{c_i,\tau}(s,\cdot)))$
8: **return** $\{u^*_{c_i,\tau}\}_{i=1}^p, \pi^\tau$

---

## 4.2 SAFE META-POLICY TRAINING ALGORITHM

To solve the optimization problem (2) for meta-training, we first consider the computation of the meta-gradient, i.e., $\nabla_\phi\mathbb{E}_{\tau\sim\mathbb{P}(\Gamma)}[J_\tau(\mathcal{A}^s(\pi_\phi,\Lambda,\Delta,\tau))]$. The following proposition provides the computation of $\nabla_\phi J_\tau(\mathcal{A}^s(\pi_\phi,\Lambda,\Delta,\tau))$. Notice that the Lagrangian multipliers $\{u^*_{c_i,\tau}\}_{i=1}^p$ in Propositions 1 and 2 are solved by problem (4), and thus depend on the meta-policy $\pi_\phi$. We denote the solved Lagrangian multipliers with $\pi_\phi$ as $u^*_{c_i,\tau}(\pi_\phi)$ in the following sections.

**Proposition 3.** *Suppose the assumption in Proposition 1 holds. Let $\pi^\tau = \mathcal{A}^s(\pi_\phi,\Lambda,\Delta,\tau)$. Under certain conditions, we have that $\nabla_\phi J_\tau(\pi^\tau)$ exists and $\nabla_\phi J_\tau(\pi^\tau) = \frac{1}{1-\gamma}\mathbb{E}_{s\sim\nu_\tau^{\pi^\tau},a\sim\pi^\tau(\cdot|s)}[(\nabla_\phi\eta(\pi_\phi)^{-1}\bar{Q}^{\pi_\phi}_\tau(s,a) + \eta(\pi_\phi)^{-1}\nabla_\phi\bar{Q}^{\pi_\phi}_\tau(s,a) + \nabla_\phi f_\phi(s,a))Q^{\pi^\tau}_\tau(s,a)]$, where $\eta(\pi_\phi) \triangleq \lambda + (1-\gamma)\sum_{i=1}^p u^*_{c_i,\tau}(\pi_\phi)\lambda_{c_i}$, and $\bar{Q}^{\pi_\phi}_\tau \triangleq Q^{\pi_\phi}_\tau - \sum_{i=1}^p u^*_{c_i,\tau}(\pi_\phi)Q^{\pi_\phi}_{c_i,\tau}$.*

The computations of $\nabla_\phi Q^{\pi_\phi}_\tau(\cdot)$, $\nabla_\phi Q^{\pi_\phi}_{c_i,\tau}(\cdot)$ and $\nabla_\phi u^*_{c_i,\tau}(\pi_\phi)$ are shown in Appendices F.3.2 and F.3.3. The complete statement of Proposition 3 that includes the sufficient condition of the existence of $\nabla_\phi J_\tau(\pi^\tau)$, as well as the proof of the proposition are shown in Appendix F.3.1. In Proposition 3, the gradient $\nabla_\phi u^*_{c_i,\tau}(\pi_\phi)$ w.r.t $\phi$, is the gradient of the solved Lagrangian multipliers, i.e. the optimal solution of problem (4). We apply the implicit gradient theorem for constrained optimization in (Giorgi & Zuccotti, 2018; Xu & Zhu, 2023a) to show the existence and the computation of

$\nabla_\phi u^*_{c_i,\tau}(\pi_\phi)$, which is shown in Appendix F.3.3. In practice, we simplify the computation of the meta-gradient of $\nabla_\phi J_\tau(\pi^\tau)$ as

$$\mathbb{E}_{s\sim\nu^{\pi^\tau}_\tau, a\sim\pi^\tau(\cdot|s)}[(\nabla_\phi f_\phi(s,a) + \eta(\pi_\phi)^{-1}\tilde{\nabla}_\phi \bar{Q}^{\pi_\phi}_\tau(s,a))Q^{\pi^\tau}_\tau(s,a)], \qquad (6)$$

where $\eta(\pi_\phi) \triangleq \lambda + (1-\gamma)\sum_{i=1}^p u^*_{c_i,\tau}(\pi_\phi)\lambda_{c_i}$ and $\tilde{\nabla}_\phi \bar{Q}^{\pi_\phi}_\tau = \nabla_\phi Q^{\pi_\phi}_\tau - \sum_{i=1}^p u^*_{c_i,\tau}(\pi_\phi)\nabla_\phi Q^{\pi_\phi}_{c_i,\tau}$. In (6), we take $\nabla_\phi u^*_{c_i,\tau}(\pi_\phi) = 0$ in Proposition 3 approximately. On one hand, the computation complexity of $\nabla_\phi u^*_{c_i,\tau}(\pi_\phi)$ is high, as shown in Appendix F.3.3. On the other hand, under this approximation, we only omit the small change of the Lagrangian multiplier $u^*_{c_i,\tau}(\pi_\phi)$ around the meta-policy $\pi_\phi$, i.e., we keep the penalty to constraint violation but treat the weight of the penalty to constraint violation unchanged over a small neighbor of $\pi_\phi$. Therefore, the omitted term is a higher-order term with a smaller impact on the meta-gradient. Note that, the meta-gradients in many meta-learning approaches include the Hessian computation, such as supervised meta-learning approaches, like MAML and iMAML (Finn et al., 2017; Rajeswaran et al., 2019; Xu & Zhu, 2023b), meta-RL (Finn et al., 2017; Liu et al., 2019b) and safe meta-RL approach meta-CPO (Cho & Sun, 2024). In contrast, thanks to the closed-form solution (shown in Proposition 1) of the policy adaptation problem (1), the meta-gradient in (6) does not include the computations of Hessian and inverse of Hessian w.r.t. $\phi$, which holds a comparable computational complexity as the policy gradient, and therefore is more computationally efficient than the above meta-learning approaches.

---

**Algorithm 2** Safe meta-policy training algorithm

---

**Require:** Initial meta-policy $\pi_{\phi_0}$; allowable constraint violation $\delta_{c_i}$ defined in Problems (1) and (2).
1: **for** $n = 0, \cdots, N-1$ **do**
2:     Sample a task $\tau$ with the CMDP $\mathcal{M}_\tau$ from the task distribution $\mathbb{P}(\Gamma)$
3:     Evaluate $J_{c_i,\tau}(\pi_{\phi_n})$, $A^{\pi_{\phi_n}}_\tau(\cdot,\cdot)$ and $A^{\pi_{\phi_n}}_{c_i,\tau}(\cdot,\cdot)$ by sampling data using the meta-policy $\pi_{\phi_n}$ on task $\tau$
4:     **if** $J_{c_i,\tau}(\pi_{\phi_n}) \leq d_{i,\tau} + \delta_{c_i}, \forall i = 1, \cdots, p$ **then**
5:         Obtain the task-specific policy $\pi^\tau$ and the Lagrangian multipliers $u^*_{c_i,\tau}(\pi_{\phi_n})$ by Algorithm 1 with the meta-policy $\pi_{\phi_n}$
6:         Evaluate $Q^{\pi^\tau}_\tau(\cdot,\cdot)$ by sampling data using the task-specific policy $\pi^\tau$ on task $\tau$
7:         Compute the meta-gradient $\nabla_\phi J_\tau(\pi^\tau)$ by (6)
8:         Take a step of TRPO (Schulman et al., 2015a) with using $\nabla_\phi J_\tau(\pi^\tau)$ towards maximize $J_\tau(\pi^\tau)$ to obtain $\phi_{n+1}$
9:     **else**
10:        Choose any $i_n \in \{1, \cdots, p\}$ such that $J_{C_{i_n}}(\pi_{\phi_n}) > d_{i_n,\tau} + \delta_{c_{i_n}}$
11:        Compute the policy gradient $\nabla_\phi J_{C_{i_n},\tau}(\pi_{\phi_n}) \propto \mathbb{E}_{s\sim\nu^{\pi_{\phi_n}}_\tau, a\sim\pi_{\phi_n}(\cdot|s)}[\nabla_\phi f_{\phi_n}(s,a)A^{\pi_{\phi_n}}_{C_{i_n},\tau}(s,a)]$.
12:        Take a step of TRPO with using $\nabla_\phi J_{C_{i_n},\tau}(\pi_{\phi_n})$ towards minimize $J_{C_{i_n},\tau}(\pi_\phi)$ to obtain $\phi_{n+1}$
13:     **end if**
14: **end for**
15: **return** $\pi_{\phi_N}$

---

The safe meta-policy training algorithm aims to solve the optimization problem in (2) and is stated in Algorithm 2. To deal with the constraint imposed on the meta-policy $\pi_\phi$ in problem (2), we use the idea similar to CRPO (Xu et al., 2021). Specifically, we first check the constraint violation in line 4. If the constraints are not violated, we maximize the meta-objective; otherwise, we minimize the constraint functions. Under this procedure, we always have $J_{c_i,\tau}(\pi_{\phi_n}) \leq d_{i,\tau} + \delta_{c_i}, \forall i = 1, \cdots, p$ when computing the task-specific policy $\pi^\tau = \mathcal{A}^s(\pi_{\phi_n}, \Lambda, \Delta, \tau)$, and therefore the solution of $\pi^\tau$ always exists, according to Proposition 1. To stabilize the training, we use the TRPO for the policy update in lines 8 and 12, which only needs the gradient information.

## 5 THEORETICAL RESULTS

In this section, we introduce the theoretical results of the safe meta-RL framework. Note that problem (2) is a constrained bilevel optimization problem, and the convergence and optimality analysis of solving the problem and obtaining $\pi_{\phi^*}$ are widely studied in (Xu & Zhu, 2023a; Bertrand et al., 2022; Liu et al., 2021). So we analyze the performance of the solved meta-policy $\pi_{\phi^*}$ in our theoretical results. In particular, we introduce the necessary assumptions and notations, derive the performance guarantee for safe policy adaptation $\mathcal{A}^s$ in Section 5.1, and then derive the optimality and safety guarantee of the safe meta-RL framework in Section 5.2.

We introduce several necessary assumptions and notations used in the theoretical results.

**Assumption 1** (Non-empty feasible set). *The feasible set of problem (2) is not empty.*

**Assumption 2** (Sufficient visit in safe states). *There exists a set of states $\mathcal{S}^v \subseteq \mathcal{S}$ and a constant $\eta > 0$ such that, for any task $\tau \in \Gamma$ and any safe policy $\pi^s \in \{\pi \in \Pi : J_{c_i,\tau}(\pi) \leq d_i + \delta_{max}, \forall i = 1, \cdots, p\}$, $\nu_\tau^{\pi^s}(s) \geq \eta$ for all $s \in \mathcal{S}^v$.*

Assumption 1 supposes that problem (2) is well defined and its optimal meta-parameter $\phi^*$ exists. Assumption 2 supposes that there exists a set of states $\mathcal{S}^v$ such that the safe policy can take sufficient visitation in the set $\mathcal{S}^v$. We denote $\alpha \in (0, 1]$ as the lower bound of the visitation probability of safe policies to $\mathcal{S}^v$, i.e., $\sum_{s \in \mathcal{S}^v} \nu_\tau^{\pi^s}(s) \geq \alpha$ or $\int_{\mathcal{S}^v} \nu_\tau^{\pi^s}(s) ds \geq \alpha$ for any $\pi^s$.

Since the reward $r_\tau \leq r^{max}$ and $c_{i,\tau} \leq c_i^{max}$, then $|A_\tau^\pi(s,a)| \leq r^{max}/(1-\gamma)$ and $|A_{c_i,\tau}^\pi(s,a)| \leq c_i^{max}/(1-\gamma)$ are upper bounded. We define the upper bounds as $A^{max} \triangleq \max_{\tau \in \Gamma, \pi \in \Pi} |A_\tau^\pi(s,a)|$ and $A_{c_i}^{max} \triangleq \max_{\tau \in \Gamma, \pi \in \Pi} |A_\tau^\pi(s,a)|$ for each $i = 1, \cdots, p$.

## 5.1 MONOTONIC IMPROVEMENT AND ANYTIME SAFETY FOR POLICY ADAPTATION

We first introduce an intermediate lemma. Its proof is shown in Appendix F.4.1.

**Lemma 1.** *Suppose that Assumption 2 holds. For any task $\tau$, and any safe policies $\pi$, $\pi' \in \{\pi \in \Pi : J_{c_i,\tau}(\pi) \leq d_i + \delta_{max}, \forall i = 1, \cdots, p\}$, we have $J_\tau(\pi') \leq J_\tau(\pi) + \mathbb{E}_{s \sim \nu_\tau^\pi, a \sim \pi'(\cdot|s)} \left[ \frac{A_\tau^\pi(s,a)}{1-\gamma} \right] + \frac{4\gamma A^{max}}{\eta\alpha(1-\gamma)^2} \mathbb{E}_{s \sim \nu_\tau^\pi} [D_{KL}(\pi'(\cdot|s)||\pi(\cdot|s))]$ and $J_\tau(\pi') \geq J_\tau(\pi) + \mathbb{E}_{s \sim \nu_\tau^\pi, a \sim \pi'(\cdot|s)} \left[ \frac{A_\tau^\pi(s,a)}{1-\gamma} \right] - \frac{4\gamma A^{max}}{\eta\alpha(1-\gamma)^2} \mathbb{E}_{s \sim \nu_\tau^\pi} [D_{KL}(\pi'(\cdot|s)||\pi(\cdot|s))]$. The inequalities also holds for each $i = 1, \cdots, p$, when $A_\tau^\pi$ and $A_\tau^{\pi'}$ are replaced by $A_{c_i,\tau}^\pi$ and $A_{c_i,\tau}^{\pi'}$, $A^{max}$ is replaced by $A_{c_i}^{max}$, and $J_\tau$ is replaced by $J_{c_i,\tau}$.*

The right-hand side of the inequalities corresponds to the objective function and constraint functions of $\mathcal{A}^s$ in problem (1), which has a closed-form solution, as shown in Section 3.1. In specific, when using the first inequality in Lemma 1 on the accumulated cost $J_{c_i,\tau}(\pi')$, the right-hand side is the upper bound of $J_{c_i,\tau}(\pi')$. Therefore, the constraint functions in problem (1) limit the upper bound of $J_{c_i,\tau}(\pi')$ to be below the specified constraint requirement, which also applies to $J_{c_i,\tau}(\pi')$ itself. When using the second inequality in Lemma 1 on the accumulated reward $J_\tau(\pi')$, the right-hand side is the lower bound of $J_\tau(\pi')$. Then, $\mathcal{A}^s$ in problem (1) is to maximize the lower bound of $J_\tau(\pi')$, which guarantees monotonic improvement. This idea is also used in (Schulman et al., 2015a; Achiam et al., 2017). We state the results in Proposition 4 and show its proof in Appendix F.4.2.

**Proposition 4.** *Suppose that Assumption 2 holds. Suppose $\pi_\phi$ satisfies $J_{c_i,\tau}(\pi_\phi) \leq d_{i,\tau} + \delta_{c_i}, \forall i = 1, \cdots, p$. Let $\pi^\tau = \mathcal{A}^s(\pi_\phi, \Lambda, \Delta, \tau)$ with $\lambda \geq \frac{4\gamma A^{max}}{\eta\alpha(1-\gamma)}$ and $\lambda_{c_i} \geq \frac{4\gamma A_{c_i}^{max}}{\eta\alpha(1-\gamma)^2}$ for each $i$. Then, $J_{c_i,\tau}(\pi^\tau) \leq d_{i,\tau} + \delta_{c_i}$ for each $i$, and $J_\tau(\pi^\tau) \geq J_\tau(\pi_\phi)$.*

With this proposition, we can derive the properties of monotonic improvement and anytime safety guarantee for the policy adaptation, which is stated in Corollary 1.

**Corollary 1.** *Suppose that Assumptions 1 and 2 hold. Let $\lambda \geq \frac{4\gamma A^{max}}{\eta\alpha(1-\gamma)}$ and $\lambda_{c_i} \geq \frac{4\gamma A_{c_i}^{max}}{\eta\alpha(1-\gamma)^2}$ for $i = 1, \cdots, p$. Let $\pi_{[k+1]}^\tau = \mathcal{A}^s(\pi_{[k]}^\tau, \Lambda, \Delta, \tau)$ with $\delta_{c_i} = 0$ for $k \in \mathbb{N}$, where $\pi_{[0]}^\tau = \pi_{\phi^*}$ being the solution of problem (2). Then, for all $k \in \mathbb{N}$, $J_{c_i,\tau}(\pi_{[k]}^\tau) \leq d_{i,\tau}$ for each $i$ and $J_\tau(\pi_{[k+1]}^\tau) \geq J_\tau(\pi_{[k]}^\tau)$.*

When a new task $\tau \in \Gamma$ is given, we start from the meta-policy $\pi_{\phi^*}$, iteratively implement $\mathcal{A}^s$, and generate a policy sequence $\{\pi_{[k]}^\tau\}_{k=0}^N$. As indicated in Corollary 1, the constraints are satisfied for each policy in the policy sequence, which shows the anytime safety of the policy adaptations.

## 5.2 NEAR-OPTIMALITY AND SAFETY GUARANTEE FOR META-POLICY TRAINING

In Section 5.1, we show the policy is monotonically improved from $\pi_{\phi^*}$ during policy adaptation. On the other hand, $\pi_{\phi^*}$ is learned from the task distribution $\mathbb{P}(\Gamma)$, which should be a good initial policy for the task sampled from $\mathbb{P}(\Gamma)$. In this section, we compare the policy adapted from $\pi_{\phi^*}$ with the task-specific optimal policy and verify the near-optimality of the proposed safe meta-RL framework. We start with the definition of the optimal task-specific policies and the task variance.

**Definitions.** Define the optimal policy $\pi_*^\tau$ for task $\tau$ as $\pi_*^\tau \triangleq \arg\max_{\pi \in \Pi} J_\tau(\pi)$ s.t. $J_{c_i,\tau}(\pi) \leq d_{i,\tau}$. Define the $\epsilon$-conservatively optimal policy $\pi_{*,[\epsilon]}^\tau$, which is optimal for $\tau$ under conservative safety constraints, i.e., $\pi_{*,[\epsilon]}^\tau \triangleq \arg\max_{\pi \in \Pi} J_\tau(\pi)$ s.t. $J_{c_i,\tau}(\pi) \leq d_{i,\tau} - \epsilon$, where the conservative constant $\epsilon \geq 0$, and $\pi_*^\tau = \pi_{*,[0]}^\tau$. We define the variance of a task distribution $\mathbb{P}(\Gamma)$ as $\mathcal{V}ar(\mathbb{P}(\Gamma)) \triangleq \min_\phi \mathbb{E}_{\tau \sim \mathbb{P}(\Gamma)} \mathbb{E}_{s \sim \nu_\tau^{\pi_\phi}}[D_{KL}(\pi_*^\tau(\cdot|s)||\pi_\phi(\cdot|s))]$, which the minimal mean square of the distances among the optimal task-specific policies $\pi_*^\tau$, and the minimal point is denoted as $\hat{\phi}$. Similarly, the task variance under the conservative safety constraints is defined as $\mathcal{V}ar^\epsilon(\mathbb{P}(\Gamma)) \triangleq \min_\phi \mathbb{E}_{\tau \sim \mathbb{P}(\Gamma)} \mathbb{E}_{s \sim \nu_\tau^{\pi_\phi}}[D_{KL}(\pi_{*,[\epsilon]}^\tau(\cdot|s)||\pi_\phi(\cdot|s))]$, and the minimal point is denoted as $\hat{\phi}^{[\epsilon]}$. The radius of $\mathbb{P}(\Gamma)$ is defined as $R(\mathbb{P}(\Gamma)) \triangleq \max_{\tau \in \Gamma, \epsilon \in E} \mathbb{E}_{s \sim \nu_\tau^{\pi_{\hat{\phi}[\epsilon]}}}[D_{KL}(\pi_{*,[\epsilon]}^\tau(\cdot|s)||\pi_{\hat{\phi}^{[\epsilon]}}(\cdot|s))]$, where the set $E$ is defined by $E \triangleq \{\epsilon \geq 0 : \pi_{*,[\epsilon]}^\tau$ exists for all $\tau \in \Gamma\}$. Note that the task variance $\mathcal{V}ar^{[\epsilon]}$ and the radius $R$ is the inherent property of $\mathbb{P}(\Gamma)$, which measures the similarity of tasks sampled from $\mathbb{P}(\Gamma)$. For example, if the reward function $r$ and cost $c_i$ among tasks are similar, optimal policies $\pi_{*,[\epsilon]}^\tau$ are close, then $\mathcal{V}ar^{[\epsilon]}$ and $R$ are close to 0. With the definitions, the near-optimality and safety guarantee of the safe meta-RL is shown in Theorem 1.

**Theorem 1.** *Suppose that Assumptions 1 and 2 hold. Let* $\lambda = \frac{4\gamma A^{max}}{\eta\alpha(1-\gamma)}$, $\lambda_{c_i} = \frac{4\gamma A_{c_i}^{max}}{\eta\alpha(1-\gamma)^2}$ *and* $\delta_{c_i} = \frac{8\gamma A_{c_i}^{max}}{\eta\alpha(1-\gamma)^2} R(\mathbb{P}(\Gamma)) - \epsilon$ *for all* $i = 1, \cdots, p$, *where* $\epsilon$ *is chosen from* $\left[0, \frac{8\gamma A_{c_i}^{max}}{\eta\alpha(1-\gamma)^2} R(\mathbb{P}(\Gamma))\right]$. *Let* $\phi^*$ *be the solution of problem (2). The solution of* $\mathcal{A}^s(\pi_{\phi^*}, \Lambda, \Delta, \tau)$ *exists, and we have*

$$\mathbb{E}_{\tau \sim \mathbb{P}(\Gamma)}[J_\tau(\mathcal{A}^s(\pi_{\phi^*}, \Lambda, \Delta, \tau))] \geq \mathbb{E}_{\tau \sim \mathbb{P}(\Gamma)}[J_\tau(\pi_{*,[\epsilon]}^\tau)] - \frac{8\gamma A^{max}}{\eta\alpha(1-\gamma)^2} \mathcal{V}ar^\epsilon(\mathbb{P}(\Gamma)), \qquad (7)$$

$$J_{c_i,\tau}(\mathcal{A}^s(\pi_{\phi^*}, \Lambda, \Delta, \tau)) - d_{c_i,\tau} \leq \frac{8\gamma A_{c_i}^{max}}{\eta\alpha(1-\gamma)^2} R(\mathbb{P}(\Gamma)) - \epsilon, \text{ for any } \tau \in \Gamma. \qquad (8)$$

The proof of Theorem 1 is shown in Appendix F.4.3. The theorem derives (i) the lower bound of the expected accumulated reward of the policy $\pi^\tau$ adapted by one time of $\mathcal{A}^s$ from the meta-parameter $\pi_{\phi^*}$ with the comparison to the task-specific (conservatively) optimal policy $\pi_{*,[\epsilon]}^\tau$. It also derives (ii) the upper bound of the constraint violation for each task $\tau$.

**Case 1** (Safety guarantee). *When* $\delta_{c_i} = 0$, *the safe constraint is strictly satisfied, i.e.,* $J_{c_i,\tau}(\pi^\tau) - d_{c_i,\tau} \leq 0$ *for any* $\tau$, *but the optimality comparator* $J_\tau(\pi_{*,[\epsilon]}^\tau)$ *with* $\epsilon = \frac{8\gamma A_{c_i}^{max}}{\eta\alpha(1-\gamma)^2} R(\mathbb{P}(\Gamma))$ *in (7) is suboptimal ($\epsilon$-conservatively optimal).*

**Case 2** (Near-optimality). *When* $\delta_{c_i} = \frac{8\gamma A_{c_i}^{max}}{\eta\alpha(1-\gamma)^2} R(\mathbb{P}(\Gamma))$, *the optimality comparator* $J_\tau(\pi_{*,[0]}^\tau) = J_\tau(\pi_*^\tau)$ *in (7) is all-task optimum, but the constraint is violated at most* $\frac{8\gamma A_{c_i}^{max}}{\eta\alpha(1-\gamma)^2} R(\mathbb{P}(\Gamma))$.

As shown in Cases 1 and 2, there is a trade-off between the optimality of accumulated reward and the safety constraint satisfaction when the allowable constraint violation thresholds $\delta_{c_i}$ vary. In particular, when $\delta_{c_i}$ is increased, the optimality is improved while the constraint violation increases. As indicated by the optimality-safety trade-off, in the implementation of the proposed algorithm, we choose a large $\delta_{c_i}$ when the constraint satisfaction is not required to be strict, and a small $\delta_{c_i} \approx 0$ when the constraint satisfaction is prioritized. The reason for the trade-off is that the constraint function in problem (1) approximate the true constraints $J_{c_i,\tau}(\pi) - d_{c_i,\tau} \leq 0$ for any $\pi$ by only knowing the information (the advantage functions $A_{c_i,\tau}^{\pi_\phi}$) at a single policy $\pi_\phi$, and therefore are more conservative than the true constraints, which leads to loss of optimality. To the best of our knowledge, as anytime safety cannot be guaranteed in the existing framework (Khattar et al., 2023; Cho & Sun, 2024), it is the first time to show the trade-off between optimality and safety, and is also the first to provide an optimality bound with the anytime safe guarantee. Moreover, Theorem 1 is reduced to the optimality analysis in (Xu & Zhu, 2024) when choosing $\epsilon = 0$ for the unconstrained meta-RL.

Next, we delve into the optimality bound. Consider fixing $\delta_{c_i}$ and $\epsilon$ and then fixing the upper bound of the constraint violation $J_{c_i,\tau}(\pi^\tau)$. Theorem 1 shows that, the performance of meta-RL is improved when the variance of the task distribution $\mathcal{V}ar^\epsilon(\mathbb{P}(\Gamma))$ is reduced, as $\pi^\tau$ approach the task-specific optimal policy $\pi_{*,[\epsilon]}^\tau$. It corresponds to the intuition of meta-learning, which is that, when the variance of a task distribution is smaller, the tasks are more similar, and then the experience learned from the task distribution works better for new tasks sampled from the task distribution.

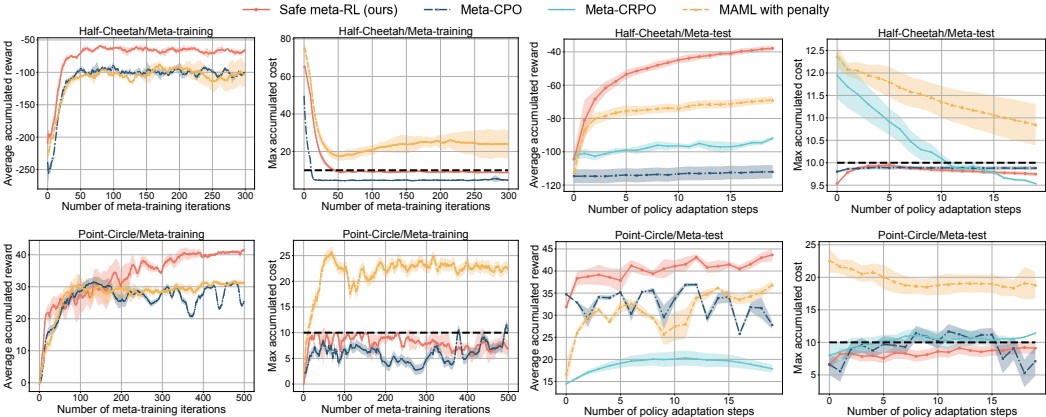

Figure 1: Average accumulated reward (**columns 1 and 3, higher is better**) and maximal accumulated cost (**columns 2 and 4, higher is worse**) across all validation/test tasks during the meta-training (**columns 1 and 2**) and the meta-test (**columns 3 and 4**) in Half-Cheetah (**row 1**) and Point-Circle (**row 2**). The accumulated reward and cost during meta-training are computed on the policy adapted one step from the meta-policy. The black dashed line is the constraint of the accumulated cost (below the line means satisfaction).

## 6 EXPERIMENTS

Our experiments aim to validate three claimed benefits of the proposed algorithms for safe meta-RL: (i) superior optimality, i,e, the accumulated rewards of the proposed algorithms can exceed those of baselines; (ii) anytime safety, i,e, all the learned meta-policy and the adapted policies should satisfy the safety constraint; (iii) high computational efficiency for both the meta-training and meta-test.

We conduct experiments on four high-dimensional locomotion scenarios, including Half-Cheetah, Humanoid, Hopper, Swimmer, and three navigation scenarios with collision avoidance, including Point-Circle, Car-Circle-Hazard, and Point-Button in Gym and Safety-Gymnasium libraries (Brockman et al., 2016; Ji et al., 2023). We compare the proposed method with three benchmarks: (a) MAML (Finn et al., 2017) with constraint penalty; (b) meta-CPO (Cho & Sun, 2024); (c) meta-CRPO (Khattar et al., 2023). In (a), we add a weighted penalty term for constraint violation to the loss function of the MAML. Note that (c) is originally designed for online safe meta-RL, where tasks are revealed sequentially during the meta-training. So, we use (3) with all training tasks provided before the meta-training and it does not have the meta-training stage (Figures 1 and 2 do not have meta-training for meta-CRPO). For the fairness of the comparison, all the methods have the same data requirements and task settings. More details about the settings of the tasks, algorithm implementation, and hyper-parameters are shown in Appendices D.1 and D.2.

Figures 1 and 2 show the experiment results in Half-Cheetah and Point-Circle. Due to the page limit, the results on the other four scenarios are shown in Appendix D.3. Figure 1 shows that the proposed safe meta-RL algorithm significantly outperforms all the baseline methods regarding the optimality, i.e. about 50% improvement over the best baselines in terms of the accumulated rewards during both the meta-training and

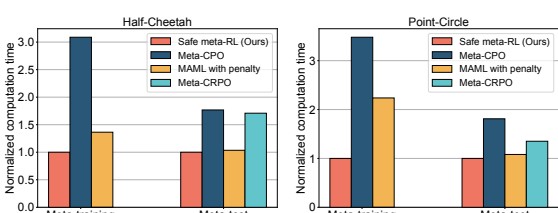

Figure 2: Normalized computation time of the meta-training (per iteration) and meta-test in Half-Cheetah and Point-Circle.

the meta-test in Half-Cheetah and Point-Circle. Moreover, as shown in the fourth column of Figure 1, the proposed algorithms achieve anytime safety during the meta-test, i.e., the maximal accumulated costs always satisfy the constraints, while the baselines cannot achieve it. Figure 2 shows that our algorithm is much more efficient than the baselines, saving about 70% of the computation time for meta-training and 50% for meta-testing compared to meta-CPO.

## 7 CONCLUSION

In this paper, we study a safe meta-RL problem with the requirement of anytime safety. We present an algorithm with three key advantages, including superior optimality, anytime safety guarantee, and high computational efficiency. We provide a theoretical analysis regarding the near-optimality and safety guarantees and empirically demonstrate the advantages of the proposed algorithms.

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

# Appendix for "Safe Meta-Reinforcement Learning via Dual-Method-Based Policy Adaptation: Near-Optimality and Anytime Safety Guarantee"

## A    RELATED WORKS

**Safety metrics in safe RL.** Safe RL aims to handle the safety requirements in the practical applications of RL. Safe RL typically applies two categories of safety metrics. The first metric is used in CMDP (Altman, 2021) and is applied in (Tessler et al., 2018a; Chow et al., 2018; Ding et al., 2021; Chen et al., 2021; Achiam et al., 2017; Yang et al., 2019; Polosky et al., 2022). It introduces costs associated with state-action pairs based on MDP, and the agent is defined as safe when the expected accumulated costs satisfy given safety constraints. The second metric is remaining in the safety region (Wachi & Sui, 2020; Yu et al., 2022; Paternain et al., 2022), which is stricter than the first metric. Specifically, the agent is safe when it remains in a desired safe set for any sampled trajectory. In this paper, we consider the anytime safety during policy adaptation, where each policy is required during the exploration of an unknown MDP. It is naturally infeasible to guarantee anytime safety under the second safety metric, as the action to remain in the safety region is unknown before the exploration. In contrast, the agent could be safe under the first safety metric even if it visits some undesired states. As a result, we consider the first safety metric.

**Solutions of CMDPs.** The solutions of the CMDPs can be categorized into (i) penalty function (Guan et al., 2022), (ii) primal-dual approaches (Tessler et al., 2018a; Chow et al., 2018; Yu et al., 2019; Ding et al., 2021; Chen et al., 2021), (iii) trust-region approaches (Achiam et al., 2017; Yang et al., 2019; Zhang et al., 2020b; Liu et al., 2022). Existing works theoretically establish the safety guarantee for both primal-dual approaches (Chow et al., 2018; Yu et al., 2019; Ding et al., 2021) and trust-region approaches (Achiam et al., 2017). The primal-dual approaches update the dual variables and the policy simultaneously. Therefore, they gradually reduce the total cost below the required threshold by multiple policy optimization steps and can only establish the safety guarantee for the final convergent policy and cannot guarantee anytime safety during policy optimization. Therefore, they cannot meet the anytime safety requirement during policy adaptation in the safe meta-RL problems, i.e., the safety constraints are satisfied during each step of policy adaptation. In contrast, trust-region approaches constrain the policy within a safe policy set, potentially ensuring safety for every policy during the policy optimization process. However, the computational complexity of existing trust-region approaches is high, especially when applied to the safe meta-RL problem. The safety policy adaptation in this paper belongs to the category of trust-region approaches. On the other hand, we propose a novel safe policy adaptation method and derive a dual method to address the computational inefficiency issue.

**Cautious adaptation and safe meta-RL.** Cautious adaptation (Zhang et al., 2020a) and safe meta-RL both consider to learn prior knowledge to improve the safety level of the adaptations in new environments. On the other hand, cautious adaptation considers the out-of-distribution exploration with the prior learned safety knowledge. The safe meta-RL focuses on in-distribution few-shot learning with safety constraints. Therefore, the safe meta-RL requires less exploration data during adaptation than cautious adaptation, but is limited to in-distribution tasks and less generalizable than cautious adaptation.

**Safe meta-RL v.s. multitask/multi-objective safe RL methods.** Safe meta-RL, multi-task safe RL Kim et al. (2023), and multi-objective safe RL Huang et al. (2022) all consider the multiple tasks in the safe RL setting. However, the biggest difference between meta-safe RL and multi-task/multi-objective safe RL is that the agent in meta-safe RL is required to adapt to a new and unknown

environment under few-shot data collection. Therefore, the policy adaptation algorithm is the most important part of meta-safe RL. This paper designs a novel policy adaptation algorithm that holds several benefits for the few-shot policy adaptation that the existing methods do not hold. In contrast, the multi-task/multi-objective safe RL learns the policies for multiple tasks during the training stage, where the policy adaptation is not required. Therefore, the multi-task/multi-objective can borrow the existing policy optimization methods and do not need to design a new one.

## B    DISCUSSION OF THE RELATIONS BETWEEN CPO (ACHIAM ET AL., 2017) AND THE SAFE POLICY ADAPTATION BY PROBLEM (1)

The safe policy adaptation $\mathcal{A}^s$ in (1) is inspired by the derivation of CPO, the first optimization problem in Section 5.3 of (Achiam et al., 2017), and replaces the term $\sqrt{D_{KL}\left(\pi(\cdot|s)\|\pi_\phi(\cdot|s)\right)}$ in the objective and the constraint functions of the optimization problem by $D_{KL}\left(\pi(\cdot|s)\|\pi_\phi(\cdot|s)\right)$. Similarly, we derive the inequalities in Lemma 1 replace the term $\max_s D_{KL}(\pi'(\cdot|s)\|\pi(\cdot|s))$ in Theorem 1 in (Schulman et al., 2015a) and replace the term $\sqrt{\mathbb{E}_{s\sim\nu_\tau^\pi}\left[D_{KL}(\pi'(\cdot|s)\|\pi(\cdot|s))\right]}$ in Corollary 3 in (Achiam et al., 2017) by $\mathbb{E}_{s\sim\nu_\tau^\pi}\left[D_{KL}(\pi'(\cdot|s)\|\pi(\cdot|s))\right]$ in the right-hand side of the inequalities.

The modification from (Achiam et al., 2017) to the safe policy adaptation $\mathcal{A}^s$ holds two benefits: (i) performance guarantee and (ii) computational efficiency. First, as Corollary 3 in (Achiam et al., 2017) enables the feasibility, the monotonic improvement, and the constraint satisfaction to hold for the solution of the first optimization problem in Section 5.3 of (Achiam et al., 2017), Lemma 1 enables the feasibility, the monotonic improvement, and the constraint satisfaction to hold for the safe policy adaptation $\mathcal{A}^s$. Second, the modification to the safe policy adaptation $\mathcal{A}^s$ enables us to derive its closed-form solution and $\mathcal{A}^s$ can be solved by the dual method, which significantly reduces the computational complexity of the meta-safe RL algorithm, as mentioned in Section 4.1. On the other hand, one cannot use the dual method for the first optimization problem in Section 5.3 of (Achiam et al., 2017), and the computational complexity is high. Paper (Achiam et al., 2017) solves an approximate problem to mitigate the issue, but the computational complexity is still high, meanwhile, the safety constraint violation cannot be avoided in theory and also usually appears in practice.

## C    COMPARISONS BETWEEN THE PROPOSED DUAL METHOD AND EXISTING LAGRANGIAN-BASED SAFE RL ALGORITHMS

The Lagrangian-based policy optimization algorithm, such as RCPO Tessler et al. (2018b), PPO-Lagrangian Ray et al. (2019) and CRPO Xu et al. (2021) used in meta-CRPO Khattar et al. (2023), has been widely used to solve safe RL. However, although both the proposed dual method in Section 4.1 and the primal-dual method in RCPO, PPO-Lagrangian, and CRPO, are Lagrangian-based safe policy optimization algorithms, they are different. The primal-dual method is much worse than the proposed method and is not suitable for this safe meta-RL problem.

The dual-method in Section 4.1, including (4) and (5), is to solve the safe policy adaptation problem in (1). As mentioned in Section 3.1, the safe policy adaptation (1) holds several benefits similar to CPO, including the safety guarantee for a single policy optimization step (using data collected on a single policy) and the monotonic improvement. Moreover, we derive the closed-form solution under certain Lagrangian multipliers for the optimization problem (1). Based on the derived closed-form solution of (1) (shown in (3) ), we can use the dual method shown in (4) and (5) to solve the safe policy adaptation problem in (1), which significantly reduces the computational complexity during the meta-training.

In contrast, RCPO and PPO-Lagrangian do not hold any of the benefits shown in CPO and the proposed algorithm. First, RCPO and PPO-Lagrangian use the gradient ascent steps on the Lagrangian, which do not have the safety guarantee and the monotonic improvement in each policy optimization step, and therefore cannot guarantee anytime safety in the meta-test stage. Moreover, there is no closed-form solution for the policy optimization step in RCPO and PPO-Lagrangian, and therefore cannot be solved by the dual method, which leads the high computational complexity during the meta-training.

# D EXPERIMENTAL SUPPLEMENTS

All experiments are executed on a computer with a 5.20 GHz Intel Core i12 CPU.

## D.1 TASK SETTINGS

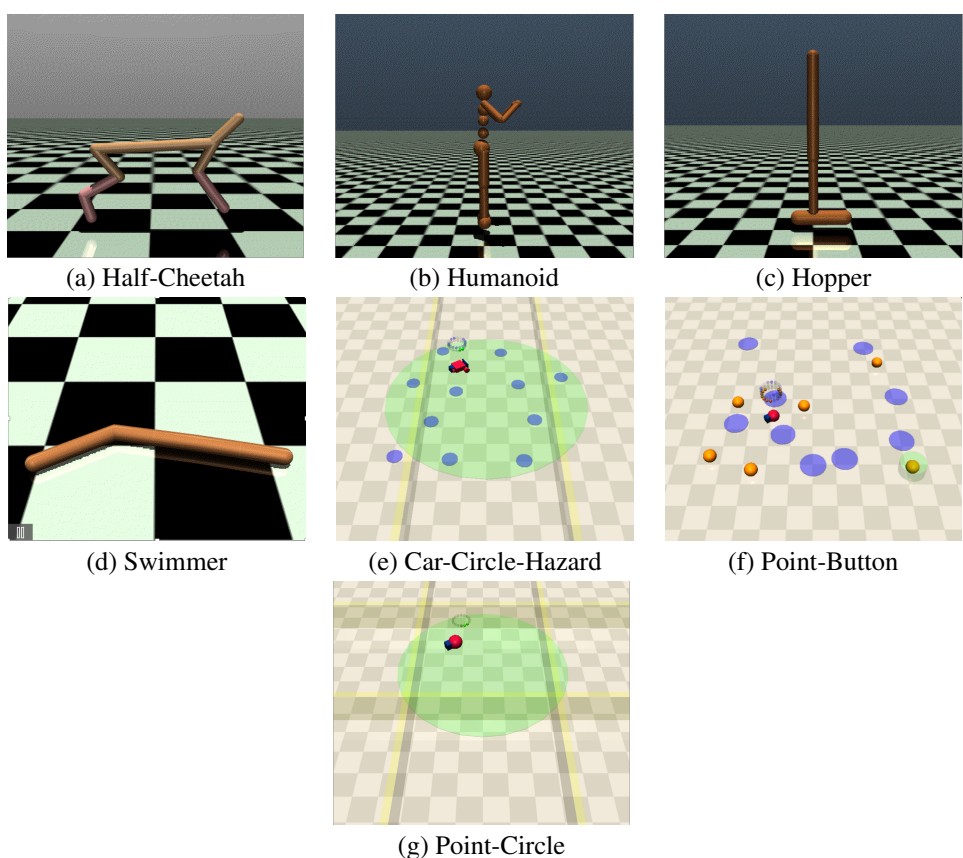

(a) Half-Cheetah      (b) Humanoid      (c) Hopper

(d) Swimmer      (e) Car-Circle-Hazard      (f) Point-Button

(g) Point-Circle

Figure 3: High-dimensional locomotion tasks and navigation tasks with collision avoidance.

We conduct experiments on totally seven scenarios, which include four high-dimensional locomotion scenarios (Half-Cheetah, Humanoid, Hopper, and Swimmer) in Gym library (Brockman et al., 2016), and three navigation scenarios with collision avoidance (Point-Circle, Car-Circle-Hazard, and Point-Button) in Safety-Gymnasium library (Ji et al., 2023). The scenarios are visually illustrated in Figure 3. We use the task setups similar to those used in previous works on meta-RL and safe meta-RL (Cho & Sun, 2024; Finn et al., 2017; Khattar et al., 2023). We provide the details of the task setups as follows.

**Half-Cheetah.** Half-Cheetah (Figure 3.a) has a 17-dimensional state space and a 6-dimensional action space. In the experiment of Half-Cheetah, the reward is the negative absolute value between the agent's current velocity and a goal velocity, where the goal velocity characterizes the task. The task distribution is defined by the distribution of the goal velocity, which is a uniform distribution from 0.0 to 2.0. The cost is defined by $h_{\text{cheetah}} - h_0 \leq d_\tau$, i.e. the cost is positive when its head is higher than $h_0$.

**Humanoid.** Humanoid (Figure 3.b) has a 376-dimensional observation space and a 17-dimensional action space. In the experiment of Humanoid, the reward is set as $v_y \sin \theta + v_x \cos \theta$, where $v_x$ and $v_y$ are the velocities along the $x$-axis and $y$-axis, and $\theta$ is the walking direction of the humanoid. So the reward is the velocity along the direction $\theta$. The task is characterized by the walking direction $\theta$, which is sampled uniformly from 0 to $\pi/2$. The cost is defined by the control cost of the humanoid robot, i.e., $\sum_i c_i^2$, where $c_i$ is the torque imposed on each component.

Table 2: Hyper-parameter setting in $\mathcal{A}^s$

| scenario | $\lambda, \lambda_{c_1}$ | $d_\tau$ | $\delta_{c_1}$ |
|---|---|---|---|
| Half-Cheetah | 1.0 | 10.0 | 0.0 |
| Humanoid | 5.0 | 20.0 | 0.0 |
| Hopper | 1.0 | 5.0 | 0.0 |
| Swimmer | 0.2 | 5.0 | 0.0 |
| Point-Circle | 0.5 | 10.0 | 0.0 |
| Car-Circle-Hazard | 0.5 | 10.0 | 0.0 |
| Point-Button | 0.5 | 10.0 | 0.0 |

**Hopper.** Hopper (Figure 3.c) has a 12-dimensional state space and a 3-dimensional action space. In the experiment of Hopper, the reward is the negative absolute value between the agent's current velocity and a goal velocity, where the goal velocity characterizes the task. The task distribution is defined by the distribution of the goal velocity, which is a uniform distribution from 0.0 to 1.0. The cost is defined by the control cost of the robot.

**Swimmer.** Swimmer (Figure 3.d) has a 8-dimensional state space and a 2-dimensional action space. In the experiment of Swimmer, for different tasks, we add a Gaussian noise to the state transition, and the variance is uniformly sampled from 0.0 to 0.5 for different tasks; we use the reward defined as the negative absolute value between the agent's current velocity and a goal velocity, which is a uniform distribution from 0.0 to 1.0, we used the cost defined by the control cost of the swimmer robot, i.e., $w \sum_i c_i^2$, where $c_i$ is the torque imposed on each component and the weight $w$ is sampled uniformly from 0.5 to 1.

**Point-Circle.** Point-Circle (Figure 3.e) has a 28-dimensional state space and a 2-dimensional action space. In the experiment of Point-Circle, a positive reward is given when the agent runs in a circle, and a positive cost is given when the agent does not stay within the safe region. The setting of the safe region characterizes the task. The task distribution is defined by the distribution of the circle radius and the wall distance. The circle radius is a uniform distribution from 1.0 to 1.5 and the wall distance is a uniform distribution from 0.55 to 0.75.

**Car-Circle-Hazard.** Car-Circle-Hazard (Figure 3.f) has a 60-dimensional state space and a 2-dimensional action space. In the experiment of Car-Circle-Hazard, a positive reward is given when the agent runs in a circle, and a positive cost is given when the agent does not stay within the safe region or collides with Hazards. The setting of the safe region and the hazards characterize the task. The task distribution is defined by the distribution of the circle radius, the distribution of the positions, and the distribution of the number of hazards. The circle radius is a uniform distribution from 0.7 to 1.0 and the number of hazards is a uniform distribution from 3 to 7. the distribution of the position of the hazard is a uniform distribution over the safety space.

**Point-Button.** Point-Button (Figure 3.g) has a 56-dimensional state space and a 2-dimensional action space. In the experiment of Point-Button, a positive reward is given when the agent touches a goal button, and a positive cost is given when it does not stay within the safe region and touches any no-goal button or hazards. The setting of the buttons and the hazards characterize the task. The task distribution is defined by the distribution of the number and the positions of buttons and the number and the positions of hazards. Both the number of buttons and the number of hazards is a uniform distribution from 6 to 10, and the distributions of positions of buttons and hazards are uniform distributions over the safety space.

### D.2 ALGORITHM SETTINGS

We apply Algorithm 4. We consider the policy as a Gaussian distribution, where the neural network produces the means and variances of the actions. The neural network policy has two hidden layers of size 64, with tanh nonlinearities. The horizon is 200, with 40 rollouts per policy adaptation step for all problems in the high-dimensional locomotion scenarios. The horizon is 500, with 10 rollouts per policy adaptation step for all problems in the navigation scenarios. The discount factor $\gamma = 0.99$. The models are trained for up to 300 meta-iterations in the meta-training. In each iteration, we sample 10 tasks from the task distribution. The meta-policy is tested on 20 tasks and is adapted by 20 iterations for each task in the meta-test. For the TRPO in meta-parameter optimization, we use the

KL-divergence constraint as $\delta = 1e - 3$. We set $\lambda = \lambda_{c_1}$ in the safe policy adaptation $\mathcal{A}^s$ in problem (1). Table 2 shows the setting of $\lambda$ and $d_\tau$ in $\mathcal{A}^s$ for each scenario.

We compare the proposed method with three benchmarks: (a) MAML (Finn et al., 2017) with constraint penalty, (b) meta-CPO (Cho & Sun, 2024), and meta-CRPO (Khattar et al., 2023). For all methods, we run each algorithm 5 times, including meta-training and meta-test, and show the mean and standard deviation of the evaluation quantities.

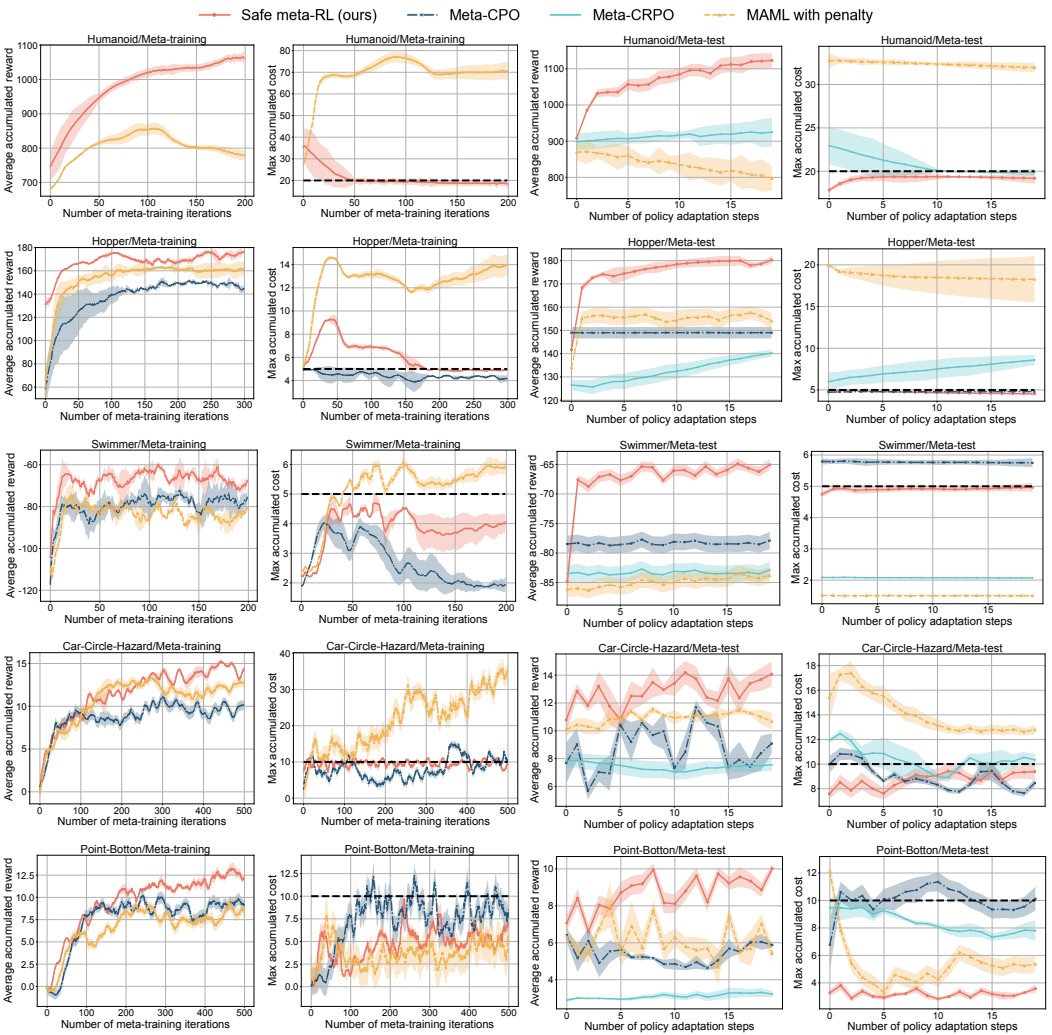

Figure 4: Average accumulated reward (**columns 1 and 3, higher is better**) and maximal accumulated cost (**columns 2 and 4, higher is worse**) across all validation/test tasks during the meta-training (**columns 1 and 2**) and the meta-test (**columns 3 and 4**) in Humanoid (**row 1**), Hopper (**row 2**), Swimmer (**row 3**), Car-Circle-Hazard (**row 4**), Point-Botton (**row 5**). The accumulated reward and cost during meta-training are computed on the policy adapted one step from the meta-policy. The black dashed line is the constraint of the accumulated cost (below the line means satisfaction).

### D.3 SUPPLEMENTAL RESULTS

Figures 4 and 5 show the experimental results in Humanoid, Hopper, Car-Circle-Hazard, and Point-Button. Note that meta-CRPO is not designed for offline optimization of meta-policy, and then there is no meta-training result for the approach. Due to the high dimension of the Humanoid tasks, the meta-training of meta-CPO is too slow (10 times slower than the proposed method) in Humanoid tasks. It is extremely time-consuming to run the meta-training of meta-CPO multiple times on humanoid tasks and draw its figure. So the result of meta-CPO is not shown in Fig 4.

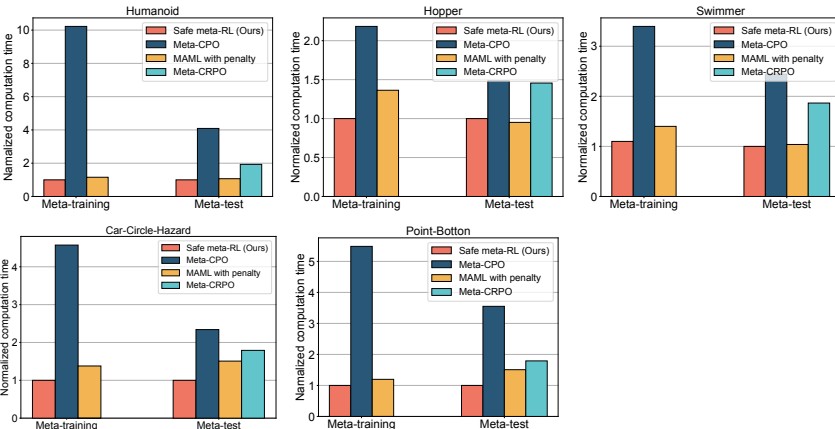

Figure 5: Normalized computation time of the meta-training and the meta-test in Humanoid, Hopper, Swimmer, Car-Circle-Hazard, and Point-Botton.

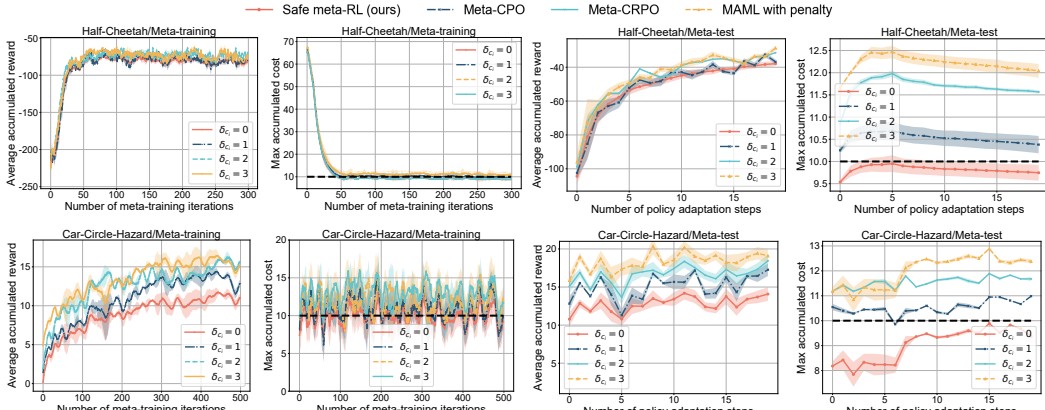

Figure 6: Average accumulated reward (**columns 1 and 3, higher is better**) and maximal accumulated cost (**columns 2 and 4, higher is worse**) across all validation/test tasks during the meta-training (**columns 1 and 2**) and the meta-test (**columns 3 and 4**) in Half-Cheetah (**row 1**) and Car-Circle-Hazard (**row 2**). The accumulated reward and cost during meta-training are computed on the policy adapted one step from the meta-policy. The black dashed line is the constraint of the accumulated cost (below the line means satisfaction).

Figure 4 shows that the proposed safe meta-RL algorithm significantly outperforms all the baseline methods regarding the optimality, i.e. the accumulated reward during both the meta-training and the meta-test in all the scenarios. Moreover, it shows that the proposed algorithms achieve anytime safety during the meta-test, i.e., the maximal accumulated costs always satisfy the constraints, while the baselines cannot achieve it. Figure 5 shows that our algorithm is much more efficient than the baselines in meta-training and meta-test.

### D.4 SELECTION OF HYPER-PARAMETER

To investigate the influence of the hyper-parameter, the allowable constraint violation constant $\delta_{c_i}$, in experiments, we conduct the experiments with $\delta_{c_i} = 0.0, 1.0, 2.0$ and $3.0$, on two environments, including Half-cheetah and Car-Circle-Hazard. The results are shown in Figure 6.

As stated in Section 5.2, the theoretical result shows a trade-off between the optimality and the safety constraint satisfaction when the allowable constraint violation thresholds $\delta_{c_i}$ vary. In particular, when $\delta_{c_i}$ is increased, the optimality is improved while the constraint violation increases. This statement is verified by Figure 6. Specifically, especially in Car-Circle-Hazard, when the allowable constraint violation threshold $\delta_{c_i}$ varies from 0.0 to 3.0, the performance is improved but the constraint violation is increased in both the meta-training and the meta-test. Therefore, as indicated in both theoretical results in Section 5.2 and the experimental results in Figure 6, we choose a large $\delta_{c_i}$ when the

constraint satisfaction is not required to be strict, and a small $\delta_{c_i} \to 0$ when the constraint satisfaction is prioritized.

For the hyper-parameter $\lambda, \lambda_{c_i}$, we set $\lambda = \lambda_{c_i}$ and tune them such that, the KL divergence of initial policy $\pi$ and the adapted policy $\pi'$ solved from the safe policy adaptation problem (1) is close to 0.03. If the KL divergence is too large, the objective and constraint functions of problem (1) are not good approximations to the accumulated reward/cost functions, as indicated by Lemma 1. If the KL divergence is too small, the policy adaptation step of problem (1) is too small.

# E ALGORITHM SUPPLEMENT

## E.1 AN ALTERNATIVE ALGORITHM IMPLEMENTATION

When the proposed algorithms are applied to high-dimensional continuous state and action spaces, we provide Algorithms 3 and 4, an alternative algorithm implementation of Algorithms 2 and 1. Compared with Algorithms 2 and 1, Algorithms 3 and 4 avoid approximating $A_\tau^{\pi_{\phi_n}}$ and $A_{c_i,\tau}^{\pi_{\phi_n}}$ during the meta-training, since it is costly to approximate the value functions $V_\tau^{\pi_{\phi_n}}$ and $V_{c_i,\tau}^{\pi_{\phi_n}}$ by neural networks and use GAE (Schulman et al., 2015b) to estimate the advantage functions $A_\tau^{\pi_{\phi_n}}$ and $A_{c_i,\tau}^{\pi_{\phi_n}}$ for each sampled task. Instead, Algorithms 3 and 4 only require to approximate $Q_\tau^{\pi_{\phi_n}}$ and $Q_{c_i,\tau}^{\pi_{\phi_n}}$, which can be estimated by Monte-Carlo sampling.

More specifically, in line 3 of Algorithm 3 replace

$$\pi^u(\cdot|s) \propto \exp(f_\phi(s,\cdot) + (\lambda + (1-\gamma)\sum_{i=1}^{p} u_i \lambda_{c_i})^{-1}(A_\tau^{\pi_\phi}(s,\cdot) - \sum_{i=1}^{p} u_i A_{c_i,\tau}^{\pi_\phi}(s,\cdot)))$$

in line 3 of Algorithm 1 by

$$\pi^u(\cdot|s) \propto \exp(f_\phi(s,\cdot) + (\lambda + (1-\gamma)\sum_{i=1}^{p} u_i \lambda_{c_i})^{-1}(Q_\tau^{\pi_\phi}(s,\cdot) - \sum_{i=1}^{p} u_i Q_{c_i,\tau}^{\pi_\phi}(s,\cdot))). \quad (9)$$

These two equations are equivalent, where the $Q$ function replaces the $A$ function. Similarly, line 10 of Algorithm 3 is also equivalent to line 7 of Algorithm 1.

Line 11 in Algorithm 4 is equivalent to line 11 of Algorithm 2, where the $Q$ function also replaces the $A$ function. The left problem is how to solve the optimization problem (4) and obtain the the Lagrangian multipliers $u_{c_i,\tau}^*(\pi_{\phi_n})$ only using the $Q$ functions.

We show the solution next. The gradient of the objective function $\bar{L}(u)$ in problem (4) w.r.t $u$ is

$$\nabla_{u_i}\bar{L}(u) = -\mathbb{E}_{s \sim \nu_\tau^{\pi_\phi}}[\mathbb{E}_{a \sim \pi^u(\cdot|s)}[A_{c_i,\tau}^{\pi_\phi}(s,a)] + (1-\gamma)\lambda_{c_i} D_{KL}(\pi^u(\cdot|s)\|\pi_\phi(\cdot|s))] + d'_{i,\tau},$$

as shown in (5). Notice that the value of $\nabla_{u_i}\bar{L}(u)$ is the constraint function in the optimization problem (1),

$$-\mathbb{E}_{s \sim \nu_\tau^{\pi_\phi}}[\mathbb{E}_{a \sim \pi(\cdot|s)}[A_{c_i,\tau}^{\pi_\phi}(s,a)] + (1-\gamma)\lambda_{c_i} D_{KL}(\pi(\cdot|s)\|\pi_\phi(\cdot|s))] + d'_{i,\tau},$$

when $\pi = \pi^u$. Moreover, the constraint function in problem (1) is already designed as a replacement of $-J_{c_i,\tau}(\pi) + d_{i,\tau} + \delta_{c_i}$ and it is cheaper to compute than $-J_{c_i,\tau}(\pi) + d_{i,\tau} + \delta_{c_i}$ for arbitrary $\pi$ in problem (1). However, in the problem of approximating $\nabla_{u_i}\bar{L}(u)$, thanks to the derived closed-form $\pi^\tau$ as $\pi^u$ shown in (9), using the original one $-J_{c_i,\tau}(\pi^u) + d_{i,\tau} + \delta_{c_i}$ becomes cheaper. So, we directly use $-J_{c_i,\tau}(\pi^u) + d_{i,\tau} + \delta_{c_i}$. Therefore, we have

$$\nabla_{u_i}\bar{L}(u) \approx (1-\gamma)(-J_{c_i,\tau}(\pi^u) + d_{i,\tau} + \delta_{c_i}). \quad (10)$$

Next, we use the first-order approximation to approximate $-J_{c_i,\tau}(\pi^u) + d_{i,\tau} + \delta_{c_i}$. Assume the policy $\pi^u$ is parameterized by $\pi_{\theta_u}$, then

$$\frac{1}{1-\gamma}\nabla_{u_i}\bar{L}(u) \approx -J_{c_i,\tau}(\pi^u) + d_{i,\tau} + \delta_{c_i}$$

$$\approx -\left(\nabla_\phi^\top J_{c_i,\tau}(\pi_\phi)(\theta_\phi - \phi) + J_{c_i,\tau}(\pi_\phi)\right) + d_{i,\tau} + \delta_{c_i}$$

$$= -\frac{1}{1-\gamma}\mathbb{E}_{s \sim \nu_\tau^{\pi_\phi}, a \sim \pi_\phi(\cdot|s)}[\nabla_u^\top \ln \pi_\phi(a|s) Q_{c_i,\tau}^{\pi_\phi}(s,a)](\theta_u - \phi) - J_{c_i,\tau}(\pi_\phi) + d_{i,\tau} + \delta_{c_i}$$

---

**Algorithm 3** Safe policy adaptation algorithm with the first-order approximation

---

**Require:** Meta-policy $\pi_\phi$; Advantage functions $Q_\tau^{\pi_\phi}$ and $Q_{c_i,\tau}^{\pi_\phi}$; step size $\beta$.
1: $u_i = 0$ for all $i \in 1, \cdots, p$
2: **for** $n = 1, \cdots, N$ **do**
3:     Compute $\pi^u(\cdot|s) \propto \exp(f_\phi(s,\cdot) + (\lambda + (1-\gamma)\sum_{i=1}^p u_i \lambda_{c_i})^{-1}(Q_\tau^{\pi_\phi}(s,\cdot) - \sum_{i=1}^p u_i Q_{c_i,\tau}^{\pi_\phi}(s,\cdot)))$
4:     **for** $i = 1, \cdots, p$ **do**
5:         $u_i \leftarrow \max\{0, u_i - \beta \nabla_{u_i} \bar{L}(u)\}$ where $\nabla_{u_i} L(u)$ is shown in (10)
6:     **end for**
7: **end for**
8: $u_{c_i,\tau}^* = u_i$ for all $i = 1, \cdots, p$
9: $\pi^\tau(\cdot|s) \propto \exp(f_\phi(s,\cdot) + (\lambda + (1-\gamma)\sum_{i=1}^p u_{c_i,\tau}^* \lambda_{c_i})^{-1}(Q_\tau^{\pi_\phi}(s,\cdot) - \sum_{i=1}^p u_{c_i,\tau}^* Q_{c_i,\tau}^{\pi_\phi}(s,\cdot)))$
10: **return** $\{u_{c_i,\tau}^*\}_{i=1}^p, \pi^\tau$

---

**Algorithm 4** An alternative algorithm of meta-training

---

**Require:** Initial meta-policy $\pi_{\phi_0}$;
1: **for** $n = 0, \cdots, N$ **do**
2:     Sample a task $\tau$ with the CMDP $\mathcal{M}_\tau$ from the task distribution $\mathbb{P}(\Gamma)$
3:     Evaluate $J_{c_i,\tau}(\pi_{\phi_n})$, $Q_\tau^{\pi_{\phi_n}}(\cdot,\cdot)$ and $Q_{c_i,\tau}^{\pi_{\phi_n}}(\cdot,\cdot)$ for the current meta-policy $\pi_{\phi_n}$ on task $\tau$
4:     **if** $J_{c_i,\tau}(\pi_{\phi_n}) \leq d_{i,\tau} + \delta_{c_i}, \forall i = 1, \cdots, p$ **then**
5:         Obtain the task-specific policy $\pi^\tau$ and the Lagrangian multipliers $u_{c_i,\tau}^*(\pi_{\phi_n})$ by Algorithm 3 with the meta-policy $\pi_{\phi_n}$
6:         Evaluate $Q_\tau^{\pi^\tau}(\cdot,\cdot)$ for the task-specific policy $\pi^\tau$ on task $\tau$
7:         Compute the meta-gradient $\nabla_\phi J_\tau(\pi^\tau) = \frac{1}{1-\gamma}\mathbb{E}_{s\sim\nu_\tau^{\pi^\tau},a\sim\pi^\tau(\cdot|s)}[\nabla_\phi f_{\phi_n}(s,a)Q_\tau^{\pi^\tau}(s,a)]$
8:         Take a step of TRPO (Schulman et al., 2015a) with using $\nabla_\phi J_\tau(\pi^\tau)$ towards maximize $J_\tau(\pi^\tau)$ to obtain $\phi_{n+1}$
9:     **else**
10:        Choose any $i_n \in \{1, \cdots, p\}$ such that $J_{C_{i_n}}(\pi_{\phi_n}) > d_{i_n,\tau} + \delta_{c_{i_n}}$
11:        Compute the policy gradient $\nabla_\phi J_{C_{i_n},\tau}(\pi_{\phi_n}) \propto \mathbb{E}_{s\sim\nu_\tau^{\pi_{\phi_n}},a\sim\pi_{\phi_n}(\cdot|s)}[\nabla_\phi f_{\phi_n}(s,a)Q_{C_{i_n},\tau}^{\pi_{\phi_n}}(s,a)]$.
12:        Take a step of TRPO (Schulman et al., 2015a) with using $\nabla_\phi J_{C_{i_n},\tau}(\pi_{\phi_n})$ towards minimize $J_{C_{i_n},\tau}(\pi_\phi)$ to obtain $\phi_{n+1}$
13:     **end if**
14: **end for**
15: **return**

---

Then,

$$\nabla_{u_i}\bar{L}(u) \approx -\mathbb{E}_{s\sim\nu_\tau^{\pi_\phi},a\sim\pi_\phi(\cdot|s)}[\nabla_u^\top \ln \pi_\phi(a|s)Q_{c_i,\tau}^{\pi_\phi}(s,a)](\theta_u - \phi) + (1-\gamma)(J_{c_i,\tau}(\pi_\phi) - d_{i,\tau} - \delta_{c_i}),$$
(11)

In this way, we replace all the estimations of the $A$ function with the estimations of the $Q$ functions, without the requirement of extra data collection.

### E.2   ACTION SAMPLING IN ALGORITHM IMPLEMENTATION

In Algorithms 1 and 2, we need to sample actions from

$$\pi^u(\cdot|s) \propto \exp(f_\phi(s,\cdot) + \eta^{-1}(Q_\tau^{\pi_\phi}(s,\cdot) - \sum_{i=1}^p u_i Q_{c_i,\tau}^{\pi_\phi}(s,\cdot))).$$
(12)

When the action space is discrete (no matter whether the state space is discrete or continuous), it is trivial to do the sampling. When the action space is high-dimensional and continuous, it is not easy to do the sampling. Here, we show two solutions. In the implementation of Algorithms 1 and 2, we apply the second solution.

### E.2.1   THE FIRST SOLUTION

Similar to many widely used RL algorithm implementations, such as (Schulman et al., 2015a), we also consider the policy parameterized by a Gaussian distribution, i.e.,

$$\pi_\phi(a|s) = \frac{\exp(f_\phi(s,a))}{\int_{a'} \exp(f_\phi(s,a'))\, da'} = A_1 \exp\left(-\frac{(a - g_\phi(s))^2}{2\delta_\phi^2}\right),$$

where $f_\phi = -\frac{(a-g_\phi(s))^2}{2\delta_\phi^2}$ and $g_\phi$ is a neural network with the input $s$. So the policy is a softmax policy.

For the policy in (12), we have

$$\pi^u(a|s) = A_2 \exp\left(-\frac{(a-g_\phi(s))^2}{2\delta_\phi^2} - \eta^{-1}\frac{(a-g_Q(s))^2}{2\delta_Q^2}\right).$$

Here, $Q_\tau^{\pi_\phi}(s,a) - \sum_{i=1}^p u_i Q_{c_i,\tau}^{\pi_\phi}(s,a)$ is approximated by $-\frac{(a-g_Q(s))^2}{2\delta_Q^2} + C(s)$ where $g_Q(s)$ and $C(s)$ are neural networks with the input $s$.

Then,

$$\pi^u(a|s) = A_3 \exp\left(-\frac{\left(a - \left(\frac{\eta\delta_Q^2}{\eta\delta_\phi^2+\delta_Q^2}g_\phi(s) + \frac{\delta_\phi^2}{\eta\delta_\phi^2+\delta_Q^2}g_Q(s)\right)\right)^2}{2\frac{\delta_\phi^2\delta_Q^2}{\eta\delta_\phi^2+\delta_Q^2}}\right), \tag{13}$$

i.e., the $\pi^u(a|s)$ is Gaussian with the mean is $\frac{\eta\delta_Q^2}{\eta\delta_\phi^2+\delta_Q^2}g_\phi(s) + \frac{\delta_\phi^2}{\eta\delta_\phi^2+\delta_Q^2}g_Q(s)$ and the standard deviation is $\sqrt{\frac{\delta_\phi^2\delta_Q^2}{\eta\delta_\phi^2+\delta_Q^2}}$. This can be sampled by many code libraries directly.

We can also treat the approximate function $-\frac{(a-g_Q(s))^2}{2\delta_Q^2}$ as $A_\tau^{\pi_\phi}(s,a) - \sum_{i=1}^p u_i A_{c_i,\tau}^{\pi_\phi}(s,a)$ and used in Algorithms (1) and (2), which take $\pi^u(\cdot|s) \propto \exp(f_\phi(s,\cdot) + \eta^{-1}(A_\tau^{\pi_\phi}(s,\cdot) - \sum_{i=1}^p u_i A_{c_i,\tau}^{\pi_\phi}(s,\cdot)))$.

### E.2.2 THE SECOND SOLUTION

In the second solution, we also consider the policy parameterized by a Gaussian distribution, i.e.,

$$\pi_\phi(a|s) = \frac{\exp(f_\phi(s,a))}{\int_{a'}\exp(f_\phi(s,a'))\,da'} = A_1 \exp\left(-\frac{(a-g_\phi(s))^2}{2\delta_\phi^2}\right),$$

where $f_\phi = -\frac{(a-g_\phi(s))^2}{2\delta_\phi^2}$ and $g_\phi$ is a neural network with the input $s$.

We use the policy parameterized by $\theta$ to approximate the policy $\pi^u(\cdot|s) \propto \exp(f_\phi(s,\cdot) + \eta^{-1}(Q_\tau^{\pi_\phi}(s,\cdot) - \sum_{i=1}^p u_i Q_{c_i,\tau}^{\pi_\phi}(s,\cdot)))$, by minimizing the expected KL-divergence, i.e.,

$$\min_\theta loss(\theta) = \mathbb{E}_{s\sim\nu_\tau^{\pi_\phi}}\left[D_{KL}\left(\pi_\theta(\cdot|s) \,\Big\|\, \frac{\exp(f_\phi(s,\cdot) + \eta^{-1}(Q_\tau^{\pi_\phi}(s,\cdot) - \sum_{i=1}^p u_i Q_{c_i,\tau}^{\pi_\phi}(s,\cdot)))}{Z_\phi(s)}\right)\right].$$

As shown in (Haarnoja et al., 2018), the problem is equivalent to $\min_\theta loss(\theta) = $

$$\mathbb{E}_{s\sim\nu_\tau^{\pi_\phi},a\sim\pi_\theta(\cdot|s)}\left[\ln\pi_\theta(a|s) - \left(f_\phi(s,a) + \eta^{-1}(Q_\tau^{\pi_\phi}(s,a) - \sum_{i=1}^p u_i Q_{c_i,\tau}^{\pi_\phi}(s,a))\right)\right].$$

This optimization problem can be restated as

$$\min_\theta \mathbb{E}_{s\sim\nu_\tau^{\pi_\phi},a\sim\pi_\phi(\cdot|s)}\left[\frac{\pi_\theta(\cdot|s)}{\pi_\phi(\cdot|s)}\left(\ln\pi_\theta(a|s) - \left(f_\phi(s,a) + \eta^{-1}(Q_\tau^{\pi_\phi}(s,a) - \sum_{i=1}^p u_i Q_{c_i,\tau}^{\pi_\phi}(s,a))\right)\right)\right].$$

Therefore, we do not need more data to approximate the expectation $\mathbb{E}_{s\sim\nu_\tau^{\pi_\phi},a\sim\pi_\phi(\cdot|s)}$. Similarly, we can also use $\pi_\theta$ to approximate $\pi^u(\cdot|s) \propto \exp(f_\phi(s,\cdot) + (\lambda + (1-\gamma)\sum_{i=1}^p u_i\lambda_{c_i})^{-1}(A_\tau^{\pi_\phi}(s,\cdot) - \sum_{i=1}^p u_i A_{c_i,\tau}^{\pi_\phi}(s,\cdot)))$.

## F ANALYSIS AND PROOF

### F.1 AUXILIARY RESULTS

**Lemma 2** (Policy gradient (Sutton & Barto, 2018; Agarwal et al., 2021)). *Let $\pi_\theta$ be the parameterized policy with the parameter $\theta$. It holds that*

$$\nabla_\theta J_\tau(\pi_\theta) = \frac{1}{1-\gamma} \mathbb{E}_{s \sim \nu_\tau^{\pi_\theta}, a \sim \pi_\theta(\cdot|s)} \left[ \nabla_\theta \ln \pi_\theta(a|s) Q_\tau^{\pi_\theta}(s,a) \right]$$

$$= \frac{1}{1-\gamma} \mathbb{E}_{s \sim \nu_\tau^{\pi_\theta}, a \sim \pi_\theta(\cdot|s)} \left[ \nabla_\theta \ln \pi_\theta(a|s) A_\tau^{\pi_\theta}(s,a) \right].$$

**Lemma 3** (Policy gradient of the softmax policy). *For the softmax policy $\pi_\theta$ as $\pi_\theta(a|s) = \frac{\exp(f_\theta(s,a))}{\sum_{a' \in \mathcal{A}} \exp(f_\theta(s,a'))}$ (in discrete action space $\mathcal{A}$) or $\pi_\theta(a|s) \triangleq \frac{\exp(f_\theta(s,a))}{\int_{\mathcal{A}} \exp(f_\theta(s,a'))da'}$ (in continuous action space $\mathcal{A}$), $\forall (s,a) \in \mathcal{S} \times \mathcal{A}$. It holds that*

$$\nabla_\theta J_\tau(\pi_\theta) = \frac{1}{1-\gamma} \mathbb{E}_{s \sim \nu_\tau^{\pi_\theta}, a \sim \pi_\theta(\cdot|s)} \left[ \nabla_\theta f_\theta(s,a) A_\tau^{\pi_\theta}(s,a) \right]. \tag{14}$$

*Proof.* We prove it under the discrete action space $\mathcal{A}$. The proof under the continuous action space $\mathcal{A}$ is similar.

From Lemma 2, we have

$$\nabla_\theta J_\tau(\pi_\theta) = \frac{1}{1-\gamma} \mathbb{E}_{s \sim \nu_\tau^{\pi_\theta}, a \sim \pi_\theta(\cdot|s)} \left[ \nabla_\theta \ln \pi_\theta(a|s) A_\tau^{\pi_\theta}(s,a) \right]$$

$$= \frac{1}{1-\gamma} \mathbb{E}_{s \sim \nu_\tau^{\pi_\theta}, a \sim \pi_\theta(\cdot|s)} \left[ \nabla_\theta \ln \left( \frac{\exp(f_\theta(s,a))}{\sum_{a' \in \mathcal{A}} \exp(f_\theta(s,a'))} \right) A_\tau^{\pi_\theta}(s,a) \right]$$

$$= \frac{1}{1-\gamma} \mathbb{E}_{s \sim \nu_\tau^{\pi_\theta}, a \sim \pi_\theta(\cdot|s)} \left[ \nabla_\theta f_\theta(s,a) - \nabla_\theta \ln \left( \sum_{a' \in \mathcal{A}} \exp(f_\theta(s,a')) \right) A_\tau^{\pi_\theta}(s,a) \right]$$

Here, $\nabla_\theta \ln \left( \sum_{a' \in \mathcal{A}} \exp(f_\theta(s,a')) \right)$ is independent with $a$, then $\nabla_\theta J_\tau(\pi_\theta)$

$$= \frac{1}{1-\gamma} \mathbb{E}_{s \sim \nu_\tau^{\pi_\theta}, a \sim \pi_\theta(\cdot|s)} \left[ \nabla_\theta f_\theta(s,a) - \nabla_\theta \ln \left( \sum_{a' \in \mathcal{A}} \exp(f_\theta(s,a')) \right) A_\tau^{\pi_\theta}(s,a) \right]$$

$$= \frac{1}{1-\gamma} \mathbb{E}_{s \sim \nu_\tau^{\pi_\theta}, a \sim \pi_\theta(\cdot|s)} \left[ \nabla_\theta f_\theta(s,a) A_\tau^{\pi_\theta}(s,a) \right] -$$

$$\frac{1}{1-\gamma} \mathbb{E}_{s \sim \nu_\tau^{\pi_\theta}} \left[ \nabla_\theta \ln \left( \sum_{a' \in \mathcal{A}} \exp(f_\theta(s,a')) \right) \mathbb{E}_{a \sim \pi_\theta(\cdot|s)} A_\tau^{\pi_\theta}(s,a) \right].$$

Since $\mathbb{E}_{a \sim \pi_\theta(\cdot|s)} A_\tau^{\pi_\theta}(s,a) = \mathbb{E}_{a \sim \pi_\theta(\cdot|s)}[Q_\tau^{\pi_\theta}(s,a)] - V_\tau^{\pi_\theta}(s) = 0$. Then,

$$\nabla_\theta J_\tau(\pi_\theta) = \frac{1}{1-\gamma} \mathbb{E}_{s \sim \nu_\tau^{\pi_\theta}, a \sim \pi_\theta(\cdot|s)} \left[ \nabla_\theta f_\theta(s,a) A_\tau^{\pi_\theta}(s,a) \right].$$

$\square$

### F.2 PROOFS OF CLOSED-FORM SOLUTION OF SAFE POLICY ADAPTATION

#### F.2.1 PROOF OF PROPOSITION 1

We provide the complete statement of Proposition 1 as the following Proposition 5.

**Proposition 5.** *When the softmax policy $\pi_\phi$ satisfies $J_{c_i,\tau}(\pi_\phi) \leq d_{i,\tau} + \delta_{c_i}, \forall i = 1, \cdots, p$, the solution $\pi^\tau$ of the optimization problem (1) exists. Suppose an appropriate constraint qualification (to be stipulated) holds at $\pi^\tau$, there exists $\{u_{c_i,\tau}^*\}_{i=1}^p$ with $u_{c_i,\tau}^* \geq 0$, such that*

$$\pi^\tau(\cdot \mid s) \propto \exp\left( f_\phi(s,\cdot) + \eta^{-1}(A_\tau^{\pi_\phi}(s,\cdot) - \sum_{i=1}^p u_{c_i,\tau}^* A_{c_i,\tau}^{\pi_\phi}(s,\cdot)) \right), \; \forall s \in \mathcal{S},$$

*i.e.,*

$$\pi^{\tau}(a|s) = \frac{\exp\left(f_{\phi}(s,a) + \left(\lambda + \sum_{i=1}^{p} u_{c_i,\tau}^* \lambda_{c_i}\right)^{-1} \left(A_{\tau}^{\pi_{\phi}}(s,a) - \sum_{i=1}^{p} u_{c_i,\tau}^* A_{c_i,\tau}^{\pi_{\phi}}(s,a)\right)\right)}{\sum_{a \in \mathcal{A}} \exp\left(f_{\phi}(s,a') + \eta^{-1}\left(A_{\tau}^{\pi_{\phi}}(s,a') - \sum_{i=1}^{p} u_{c_i,\tau}^* A_{c_i,\tau}^{\pi_{\phi}}(s,a')\right)\right)},$$

*in discrete action space $\mathcal{A}$, or*

$$\pi^{\tau}(a|s) = \frac{\exp\left(f_{\phi}(s,a) + \left(\lambda + \sum_{i=1}^{p} u_{c_i,\tau}^* \lambda_{c_i}\right)^{-1} \left(A_{\tau}^{\pi_{\phi}}(s,a) - \sum_{i=1}^{p} u_{c_i,\tau}^* A_{c_i,\tau}^{\pi_{\phi}}(s,a)\right)\right)}{\int_{a'} \exp\left(f_{\phi}(s,a') + \eta^{-1}\left(A_{\tau}^{\pi_{\phi}}(s,a') - \sum_{i=1}^{p} u_{c_i,\tau}^* A_{c_i,\tau}^{\pi_{\phi}}(s,a')\right)\right) da'},$$

*in continuous action space $\mathcal{A}$, where $\eta = (1-\gamma)\lambda + \sum_{i=1}^{p} u_{c_i,\tau}^* \lambda_{c_i}$.*

There are many constraint qualifications where each of them assures the validity of the proposition, including but not limited to Mangasarian-Fromovitz constraint qualification (MFCQ), linear independence constraint qualification (LICQ), and Slater's condition (SC) (Giorgi & Zuccotti, 2018). Refer to (Peterson, 1973) for more validated constraint qualifications.

The assumption that one constraint qualification holds at $\pi^{\tau}$ is mild. For example, if there exists a policy $\pi$ such that $\forall i$

$$J_{c_i,\tau}(\pi_{\phi}) + \mathbb{E}_{\substack{s \sim \nu_{\tau}^{\pi_{\phi}} \\ a \sim \pi(\cdot|s)}}\left[\frac{A_{c_i,\tau}^{\pi_{\phi}}(s,a)}{1-\gamma}\right] + \lambda_{c_i} \mathbb{E}_{s \sim \nu_{\tau}^{\pi_{\phi}}}\left[D_{KL}\left(\pi(\cdot|s)\|\pi_{\phi}(\cdot|s)\right)\right] < d_{i,\tau} + \delta_{c_i}, \quad (15)$$

then the Slater's condition holds. Note that when $\pi = \pi_{\phi}$, we have $J_{c_i,\tau}(\pi_{\phi}) + \mathbb{E}_{\substack{s \sim \nu_{\tau}^{\pi_{\phi}} \\ a \sim \pi(\cdot|s)}}\left[\frac{A_{c_i,\tau}^{\pi_{\phi}}(s,a)}{1-\gamma}\right] +$

$\lambda_{c_i} \mathbb{E}_{s \sim \nu_{\tau}^{\pi_{\phi}}}\left[D_{KL}\left(\pi(\cdot|s)\|\pi_{\phi}(\cdot|s)\right)\right] \leq d_{i,\tau} + \delta_{c_i}$. It usually exists a $\pi$ near $\pi_{\phi}$ such that (15) holds or the $\pi_{\phi}$ itself can assure (15) holds. Next, we prove the proposition.

*Proofs of Proposition 5.* The optimization problem (1) can be restated as

$$\operatorname*{argmin}_{\pi \in \Pi} - \mathbb{E}_{\substack{s \sim \nu_{\tau}^{\pi_{\phi}} \\ a \sim \pi(\cdot|s)}}\left[A_{\tau}^{\pi_{\phi}}(s,a)\right] + \lambda \mathbb{E}_{s \sim \nu_{\tau}^{\pi_{\phi}}}\left[D_{KL}\left(\pi(\cdot|s)\|\pi_{\phi}(\cdot|s)\right)\right],$$

$$\text{s.t.} \quad \mathbb{E}_{\substack{s \sim \nu_{\tau}^{\pi_{\phi}} \\ a \sim \pi(\cdot|s)}}\left[A_{c_i,\tau}^{\pi_{\phi}}(s,a)\right] + \lambda_{c_i}' \mathbb{E}_{s \sim \nu_{\tau}^{\pi_{\phi}}}\left[D_{KL}\left(\pi(\cdot|s)\|\pi_{\phi}(\cdot|s)\right)\right] \leq d_{i,\tau}', \ i = 1, \cdots, p,$$

where the constants $\lambda_{c_i}' \triangleq (1-\gamma)\lambda_{c_i}$, and $d_{i,\tau}' \triangleq (1-\gamma)(d_{i,\tau} + \delta_{c_i} - J_{c_i,\tau}(\pi_{\phi}))$.

First, we consider the discrete state-action space $\mathcal{S} \times \mathcal{A}$. Considering the probability at each state-action pair $\pi(a|s)$ as the decision variable, the minimization is taken over the probability simplex $\left\{\pi(\cdot|s) : 0 \leq \pi(a|s) \leq 1, \sum_{a \in \mathcal{A}} \pi(a|s) = 1\right\}$. Then the optimization problem is formally stated as

$$\operatorname*{argmin}_{\pi} \mathbb{E}_{s \sim \nu_{\tau}^{\pi_{\phi}}}\left[\sum_{a \in \mathcal{A}} -\pi(a|s)A_{\tau}^{\pi_{\phi}}(s,a) + \lambda D_{KL}\left(\pi(\cdot|s)\|\pi_{\phi}(\cdot|s)\right)\right],$$

$$\text{s.t.} \ \mathbb{E}_{s \sim \nu_{\tau}^{\pi_{\phi}}}\left[\sum_{a \in \mathcal{A}} \pi(a|s)A_{c_i,\tau}^{\pi_{\phi}}(s,a) + \lambda_{c_i}' D_{KL}\left(\pi(\cdot|s)\|\pi_{\phi}(\cdot|s)\right)\right] \leq d_{i,\tau}', \ i = 1, \cdots, p,$$

$$\sum_{a \in \mathcal{A}} \pi(a|s) = 1 \text{ for any } s \in \mathcal{S},$$

$$\pi(a|s) \leq 1 \text{ for any } a \in \mathcal{A}, s \in \mathcal{S},$$

$$- \pi(a|s) \leq 0 \text{ for any } a \in \mathcal{A}, s \in \mathcal{S}.$$

(16)

Since $\pi_{\phi} \in \Pi_{\tau}^{C}$, we have $d_{i,\tau}' = (1-\gamma)(d_{i,\tau} + \delta_{c_i} - J_{c_i,\tau}(\pi_{\phi})) \geq 0$, the solution of (16) exists.

According to Theorem 1 in (Giorgi & Zuccotti, 2018) and theorems in (Bertsekas, 1997; Boyd & Vandenberghe, 2004), since the constraint qualification holds, the Karush-Kuhn-Tucker (KKT)

conditions hold at $\pi^\tau$, i.e., there exists Lagrangian multipliers $\{u^*_{c_i,\tau}\}_{i=1}^p$, $u^*_0(s)$ for all $s \in \mathcal{S}$, $u^*_1(s,a)$ and $u^*_2(s,a)$ for all $(s,a) \in \mathcal{S} \times \mathcal{A}$, such that

$$u^*_{c_i,\tau} \geq 0, \forall i = 1, \cdots, p,$$

$$u^*_1(s,a) \geq 0, u^*_2(s,a) \geq 0, \ \forall (s,a) \in \mathcal{S} \times \mathcal{A}, \tag{17}$$

$$\mathbb{E}_{s \sim \nu^{\pi_\phi}_\tau} \left[ \sum_{a \in \mathcal{A}} \pi^\tau(a|s) A^{\pi_\phi}_{c_i,\tau}(s,a) + \lambda'_{c_i} D_{KL}\left(\pi^\tau(\cdot|s)\|\pi_\phi(\cdot|s)\right) \right] - d'_{i,\tau} \leq 0, \ \forall i = 1, \cdots, p,$$

$$\pi^\tau(s,a) \geq 0, \pi^\tau(s,a) \leq 1, \forall(s,a) \in \mathcal{S} \times \mathcal{A}, \tag{18}$$

$$\sum_{a \in \mathcal{A}} \pi^\tau(a|s) = 1, \ \forall s \in \mathcal{S}, \tag{19}$$

$$u^*_{c_i,\tau} \left( \mathbb{E}_{s \sim \nu^{\pi_\phi}_\tau} \left[ \sum_{a \in \mathcal{A}} \pi^\tau(a|s) A^{\pi_\phi}_{c_i,\tau}(s,a) + \lambda'_{c_i} D_{KL}\left(\pi^\tau(\cdot|s)\|\pi_\phi(\cdot|s)\right) \right] - d'_{i,\tau} \right) = 0,$$

$$u^*_1(s,a)(\pi^\tau(s,a) - 1) = 0, \forall(s,a) \in \mathcal{S} \times \mathcal{A}, \tag{20}$$

$$-u^*_2(s,a)\pi^\tau(s,a) = 0, \forall(s,a) \in \mathcal{S} \times \mathcal{A}, \tag{21}$$

$$\nabla_\pi L(\pi^\tau, \{u^*_{c_i,\tau}\}_{i=1}^p, u^*_0, u^*_1, u^*_2) = 0, \tag{22}$$

where

$$L(\pi, \{u^*_{c_i,\tau}\}_{i=1}^p, u^*_0, u^*_1, u^*_2)) \triangleq \mathbb{E}_{s \sim \nu^{\pi_\phi}_\tau} \left[ \sum_{a \in \mathcal{A}} -\pi(a|s) A^{\pi_\phi}_\tau(s,a) + \lambda D_{KL}\left(\pi(\cdot|s)\|\pi_\phi(\cdot|s)\right) \right]$$

$$+ \sum_{i=1}^p u^*_{c_i,\tau} \left( \mathbb{E}_{s \sim \nu^{\pi_\phi}_\tau} \left[ \sum_{a \in \mathcal{A}} \pi(a|s) A^{\pi_\phi}_{c_i,\tau}(s,a) + \lambda'_{c_i} D_{KL}\left(\pi(\cdot|s)\|\pi_\phi(\cdot|s)\right) \right] - d'_{i,\tau} \right)$$

$$+ \sum_{s \in \mathcal{S}} u^*_0(s) \left( \sum_{a \in \mathcal{A}} \pi(a|s) - 1 \right) + \sum_{s \in \mathcal{S}} \sum_{s \in \mathcal{S}} u^*_1(s,a)(\pi(s,a) - 1) - u^*_2(s,a)\pi(s,a). \tag{23}$$

Note that (17) (18) (19) (20) (21)(22) constitute the KKT condition for the following optimization problem:

$$\operatorname*{argmin}_\pi \mathbb{E}_{s \sim \nu^{\pi_\phi}_\tau} \left[ \sum_{a \in \mathcal{A}} \pi(a|s) \left( -A^{\pi_\phi}_\tau(s,a) + \sum_{i=1}^p u^*_{c_i,\tau} A^{\pi_\phi}_{c_i,\tau}(s,a) \right) \right.$$

$$\left. + \left( \lambda + \sum_{i=1}^p u^*_{c_i,\tau} \lambda'_{c_i} \right) D_{KL}\left(\pi(\cdot|s)\|\pi_\phi(\cdot|s)\right) \right] - \sum_{i=1}^p u^*_{c_i,\tau} d'_{i,\tau} \tag{24}$$

$$\text{s.t. } \sum_{a \in \mathcal{A}} \pi(a|s) = 1 \text{ for any } s \in \mathcal{S},$$

$$\pi(a|s) \leq 1 \text{ for any } a \in \mathcal{A}, s \in \mathcal{S},$$

$$-\pi(a|s) \leq 0 \text{ for any } a \in \mathcal{A}, s \in \mathcal{S}.$$

i.e., the KKT condition for the optimization problem (24) holds at $\pi^\tau$ with Lagrangian multipliers $u^*_0(s)$, $u^*_1(s,a)$ and $u^*_2(s,a)$. Here, $\{u^*_{c_i,\tau}\}_{i=1}^p$ are constants for the problem.

Since the terms $-\mathbb{E}_{s \sim \nu^{\pi_\phi}_\tau}\left[\sum_{a \in \mathcal{A}} \pi(a|s) A^{\pi_\phi}_\tau(s,a)\right]$ and $\mathbb{E}_{s \sim \nu^{\pi_\phi}_\tau}\left[\sum_{a \in \mathcal{A}} \pi(a|s) A^{\pi_\phi}_{c_i,\tau}(s,a)\right]$ are linear; the term $\mathbb{E}_{s \sim \nu^{\pi_\phi}_\tau}\left[D_{KL}\left(\pi(\cdot|s)\|\pi_\phi(\cdot|s)\right)\right]$ is convex, the optimization problem (24) is convex. Moreover, since all the constraint functions are affine, the Slater's condition holds naturally for the optimization problem (24), as shown in (Boyd & Vandenberghe, 2004). Therefore, the strong duality holds. Then, $\pi^\tau$ is the optimal solution for (24).

In (24), we can omit the term $-\sum_{i=1}^p u^*_{c_i,\tau} d'_{i,\tau}$ and keep the solution unchanged. Next, we borrow the conclusion of Proposition 3.1 in (Liu et al., 2019a), we have $\pi^\tau(a|s) =$

$$\frac{\exp\left( f_\phi(s,a) + \left(\lambda + \sum_{i=1}^p u^*_{c_i,\tau} \lambda'_{c_i}\right)^{-1} \left(A^{\pi_\phi}_\tau(s,a) - \sum_{i=1}^p u^*_{c_i,\tau} A^{\pi_\phi}_{c_i,\tau}(s,a)\right) \right)}{\sum_{a \in \mathcal{A}} \exp\left( f_\phi(s,a') + \left(\lambda + \sum_{i=1}^p u^*_{c_i,\tau} \lambda'_{c_i}\right)^{-1} \left(A^{\pi_\phi}_\tau(s,a') - \sum_{i=1}^p u^*_{c_i,\tau} A^{\pi_\phi}_{c_i,\tau}(s,a')\right) \right)},$$

i.e.,

$$\pi^{\tau}(\cdot \mid s) \propto \exp\left(f_{\phi}(s, \cdot) + (\lambda + \sum_{i=1}^{p} u_{c_i,\tau}^* \lambda_{c_i}')^{-1}(A_{\tau}^{\pi_{\phi}}(s, \cdot) - \sum_{i=1}^{p} u_{c_i,\tau}^* A_{c_i,\tau}^{\pi_{\phi}}(s, \cdot))\right),$$

for all $s \in \mathcal{S}$. Since $\lambda_{c_i}' = (1 - \gamma)\lambda_{c_i}$, the proof is done.

$\square$

### F.2.2 PROOF OF PROPOSITION 2

*Proof of Proposition 2.* For the Lagrangian multiplier variables $u$, $u_0$, $u_1$, $u_2$, we denote the solution of $\min_{\pi} L(\pi, u, u_0, u_1, u_2)$ as $\pi^{\{u, u_0, u_1, u_2\}}$ ($L$ is shown in (23)), i.e.,

$$\pi^{\{u, u_0, u_1, u_2\}} = \arg \min_{\pi} L(\pi, u, u_0, u_1, u_2).$$

From the proof of Proposition 1, we have the strong duality for the optimization problem (16) holds. Then, we have $\{u^*, u_0^*, u_1^*, u_2^*\} =$

$$\arg \max_{\{u, u_0, u_1, u_2\}} L(\pi^{\{u, u_0, u_1, u_2\}}, u, u_0, u_1, u_2), \text{ s.t. } u \geq 0, u_1 \geq 0, u_2 \geq 0. \quad (25)$$

Next, from the above optimization problem, we set $u_0$, $u_1$, $u_2$ as $u_0^*(u)$, $u_1^*(u)$, $u_2^*(u)$ in (25), where $u_0^*(u)$, $u_1^*(u)$, $u_2^*(u)$ are the solution of dual variable (Lagrangian multiplier solution) of the following problem:

$$\arg\min_{\pi} \mathbb{E}_{s \sim \nu_{\tau}^{\pi_{\phi}}} \left[\sum_{a \in \mathcal{A}} \pi(a|s)\left(-A_{\tau}^{\pi_{\phi}}(s, a) + \sum_{i=1}^{p} u_i A_{c_i,\tau}^{\pi_{\phi}}(s, a)\right)\right.$$

$$\left. + \left(\lambda + (1 - \gamma)\sum_{i=1}^{p} u_i \lambda_{c_i}\right) D_{KL}\left(\pi(\cdot|s) \| \pi_{\phi}(\cdot|s)\right)\right] - \sum_{i=1}^{p} u_i d_{i,\tau}' \quad (26)$$

$$\text{s.t. } \sum_{a \in \mathcal{A}} \pi(a|s) = 1 \text{ for any } s \in \mathcal{S},$$

$$\pi(a|s) \leq 1 \text{ for any } a \in \mathcal{A}, s \in \mathcal{S},$$

$$-\pi(a|s) \leq 0 \text{ for any } a \in \mathcal{A}, s \in \mathcal{S}.$$

We have

$$u^* = \arg\max_u L(\pi^{\{u, u_0^*(u), u_1^*(u), u_2^*(u)\}}, u, u_0^*(u), u_1^*(u), u_2^*(u)), \text{ s.t. } u \geq 0. \quad (27)$$

Similar to solution of (24), we have the solution of (26) is $\pi^u$, where $\pi^u(\cdot|s) \propto \exp(f_{\phi}(s, \cdot) + (\sum_{i=1}^{p} u_i \lambda_{c_i})^{-1}(A_{\tau}^{\pi_{\phi}}(s, \cdot) - \sum_{i=1}^{p} u_i A_{c_i,\tau}^{\pi_{\phi}}(s, \cdot)))$. Moreover, from the strong duality of the optimization problem (26) (linear inequality constraints), we have

$$\pi^{\{u, u_0^*(u), u_1^*(u), u_2^*(u)\}} = \arg\min_{\pi} L(\pi, u, u_0^*(u), u_1^*(u), u_2^*(u)) = \pi^u. \quad (28)$$

Therefore,

$$u^* = \arg\max_u L(\pi^u, u, u_0^*(u), u_1^*(u), u_2^*(u)), \text{ s.t. } u \geq 0.$$

Moreover, we know

$$\sum_{s \in \mathcal{S}} u_0^*(u)(s)\left(\sum_{a \in \mathcal{A}} \pi^u(a|s) - 1\right) + \sum_{s \in \mathcal{S}}\sum_{s \in \mathcal{S}} u_1^*(u)(s, a)(\pi^u(s, a) - 1) - u_2^*(u)(s, a)\pi^u(s, a) = 0.$$

Form (27) and (23), we have

$$u^* = \max_u \mathbb{E}_{s \sim \nu_{\tau}^{\pi_{\phi}}, a \sim \pi^u(\cdot|s)}[-A_{\tau}^{\pi_{\phi}}(s, a) + \sum_{i=1}^{p} u_i A_{c_i,\tau}^{\pi_{\phi}}(s, a)] + (\lambda + \sum_{i=1}^{p} u_i \lambda_{c_i}')$$

$$\mathbb{E}_{s \sim \nu_{\tau}^{\pi_{\phi}}}\left[D_{KL}\left(\pi^u(\cdot|s) \| \pi_{\phi}(\cdot|s)\right)\right] - \sum_{i=1}^{p} u_i(1 - \gamma)(d_{i,\tau} + \delta_{c_i} - J_{c_i,\tau}(\pi_{\phi}))$$

$$\text{s.t. } u_i \geq 0, \forall i = 1, \cdots, p.$$

Then, the proof is done.

$\square$

### F.2.3 DEVIATION OF GRADIENT W.R.T. THE DUAL VARIABLES

We derive the gradient of $\bar{L}$ w.r.t. the dual variables $u$ for (5). Let

$$\hat{L}(u, \pi^u) \triangleq \mathbb{E}_{\substack{s \sim \nu_\tau^{\pi_\phi} \\ a \sim \pi^u}} [A_\tau^{\pi_\phi}(s, a) - \sum_{i=1}^{p} u_i A_{c_i, \tau}^{\pi_\phi}(s, a)]$$

$$- (\lambda + (1 - \gamma) \sum_{i=1}^{p} u_i \lambda_{c_i}) \mathbb{E}_{s \sim \nu_\tau^{\pi_\phi}} [D_{KL}(\pi^u(\cdot|s) \| \pi_\phi(\cdot|s))] + \sum_{i=1}^{p} u_i d'_{i, \tau}$$

where $d'_{i, \tau} \triangleq (1 - \gamma)(d_{i, \tau} + \delta_{c_i} - J_{c_i, \tau}(\pi_\phi))$. Then,

$$\nabla_u \bar{L}(u) = \nabla_1 \hat{L}(u, \pi^u) + \nabla_u \pi^u \nabla_2 \hat{L}(u, \pi^u)$$

Consider $\nabla_2 \hat{L}(u, \pi^u)$. From (28), we have

$$\pi^{\{u, u_0^*(u), u_1^*(u), u_2^*(u)\}} = \arg\min_\pi L(\pi, u, u_0^*(u), u_1^*(u), u_2^*(u)) = \pi^u$$

where $L$ is shown in (23) and $u_0^*(u), u_1^*(u), u_2^*(u)$ are the solution of dual variable of (26). Then

$$\nabla_1 L(\pi^u, u, u_0^*(u), u_1^*(u), u_2^*(u)) = 0.$$

Moreover, we know

$$\sum_{s \in \mathcal{S}} u_0^*(u)(s) \left( \sum_{a \in \mathcal{A}} \pi^u(a|s) - 1 \right) + \sum_{s \in \mathcal{S}} \sum_{s \in \mathcal{S}} u_1^*(u)(s, a)(\pi^u(s, a) - 1) - u_2^*(u)(s, a)\pi^u(s, a) = 0.$$

Thus,

$$\nabla_2 \hat{L}(u, \pi^u) = \nabla_1 L(\pi^u, u, u_0^*(u), u_1^*(u), u_2^*(u)) = 0.$$

Then, we have

$$\nabla_u \bar{L}(u) = \nabla_1 \hat{L}(u, \pi^u).$$

Therefore,

$$\nabla_{u_i} \bar{L}(u) = -\mathbb{E}_{s \sim \nu_\tau^{\pi_\phi}} [\mathbb{E}_{a \sim \pi^u(\cdot|s)} [A_{c_i, \tau}^{\pi_\phi}(s, a)] + (1 - \gamma)\lambda_{c_i} D_{KL}(\pi^u(\cdot|s) \| \pi_\phi(\cdot|s))] + d'_{i, \tau}.$$

### F.3 META-GRADIENT

#### F.3.1 COMPUTATION OF META-GRADIENT

**Proposition 6.** *Let $\pi^\tau = \mathcal{A}^s(\pi_\phi, \Lambda, \Delta, \tau)$. Suppose all the assumptions in Proposition (5) hold. Suppose the LICQ and the strict complementary slackness condition (SCSC) (Giorgi & Zuccotti, 2018; Xu & Zhu, 2023a) for the optimization problem (3.1) holds at $\pi^\tau$. Then, $\nabla_\phi J_\tau(\pi^\tau)$ exists and*

$$\nabla_\phi J_\tau(\pi^\tau) = \frac{1}{1 - \gamma} \mathbb{E}_{s \sim \nu_\tau^{\pi^\tau}, a \sim \pi^\tau(\cdot|s)} [(\nabla_\phi \eta(\pi_\phi)^{-1} \bar{Q}_\tau^{\pi_\phi}(s, a)$$
$$+ \eta(\pi_\phi)^{-1} \nabla_\phi \bar{Q}_\tau^{\pi_\phi}(s, a) + \nabla_\phi f_\phi(s, a)) Q_\tau^{\pi^\tau}(s, a)],$$

*where $\eta(\pi_\phi) \triangleq \lambda + (1 - \gamma) \sum_{i=1}^{p} u_{c_i, \tau}^*(\pi_\phi)\lambda_{c_i}$, and $\bar{Q}_\tau^{\pi_\phi} \triangleq Q_\tau^{\pi_\phi} - \sum_{i=1}^{p} u_{c_i, \tau}^*(\pi_\phi)Q_{c_i, \tau}^{\pi_\phi}$.*

*Proof.* For any meta-policy $\pi_\phi$, the objective function of the optimization problem (3.1) is strongly concave and the constraint function is convex. The LICQ and the SCSC hold at $\pi^\tau$. According to Theorem 2 in (Xu & Zhu, 2023a), $\nabla_\phi J_\tau(\pi^\tau)$ exists.

We have

$$\pi^\tau(\cdot|s) \propto \exp(f_\phi(s, \cdot) + \eta(\pi_\phi)^{-1}(A_\tau^{\pi_\phi}(s, \cdot) - \sum_{i=1}^{p} u_{c_i, \tau}^* A_{c_i, \tau}^{\pi_\phi}(s, \cdot)))$$

is equivalent to

$$\pi^\tau(\cdot|s) \propto \exp(f_\phi(s, \cdot) + \eta(\pi_\phi)^{-1}(Q_\tau^{\pi_\phi}(s, \cdot) - \sum_{i=1}^{p} u_{c_i, \tau}^* Q_{c_i, \tau}^{\pi_\phi}(s, \cdot))).$$

From Lemma 3, we have

$$\nabla_\phi J_\tau(\pi^\tau) = \frac{1}{1-\gamma} \mathbb{E}_{s \sim \nu_\tau^{\pi^\tau}, a \sim \pi^\tau(\cdot|s)}[\nabla_\phi \left(\eta(\pi_\phi)^{-1} \bar{Q}_\tau^{\pi_\phi}(s,a) + f_\phi(s,a)\right) Q_\tau^{\pi^\tau}(s,a)]$$

$$= \frac{1}{1-\gamma} \mathbb{E}_{s \sim \nu_\tau^{\pi^\tau}, a \sim \pi^\tau(\cdot|s)}[\left(\nabla_\phi \eta(\pi_\phi)^{-1} \bar{Q}_\tau^{\pi_\phi}(s,a)\right.$$

$$\left. + \eta(\pi_\phi)^{-1} \nabla_\phi \bar{Q}_\tau^{\pi_\phi}(s,a) + \nabla_\phi f_\phi(s,a)\right) Q_\tau^{\pi^\tau}(s,a)].$$

$\square$

### F.3.2 COMPUTATION OF $\nabla_\phi Q_\tau^{\pi_\phi}(s,a)$

We have

$$\nabla_\phi Q_\tau^{\pi_\phi}(s,a) = \frac{\gamma}{1-\gamma} \cdot \mathbb{E}_{(s',a') \sim \sigma_{\tau,\pi_\phi}^{(s,a)}} \left[\nabla_\phi f_\phi(s',a') Q_\tau^{\pi_\phi}(s',a')\right]. \quad (29)$$

where the state-action visitation probability $\sigma_{\tau,\pi_\theta}^{(s,a)}$ initialized at $(s,a) \in \mathcal{S} \times \mathcal{A}$ is defined by

$$\sigma_{\tau,\pi_\phi}^{(s,a)}(s',a') = (1-\gamma) \sum_{t=0}^\infty \gamma^t \mathbb{P}(s_t = s', a_t = a'|\pi_\phi, s_0 \sim P_\tau(\cdot|s,a)).$$

*Proof.* As shown in (Wang et al., 2020),

$$\nabla_\phi Q_\tau^{\pi_\phi}(s,a) = \nabla_\phi \left((1-\gamma) \cdot r_\tau(s,a) + \gamma \cdot \mathbb{E}_{s' \sim P_\tau(\cdot|s,a)} \left[V_\tau^{\pi_\phi}(s')\right]\right)$$

$$= \frac{\gamma}{1-\gamma} \cdot \mathbb{E}_{(s',a') \sim \sigma_{\tau,\pi_\phi}^{(s,a)}} \left[\nabla_\phi \ln \pi_\phi(a'|s') \cdot Q_\tau^{\pi_\phi}(s',a')\right].$$

By Lemma 3, from (14), we can obtain (29). $\square$

### F.3.3 GRADIENT OF LAGRANGIAN MULTIPLIERS

We show the existence and the computation of $\nabla_\phi u_{c_i,\tau}^*(\pi_\phi)$ in the following proposition.

**Proposition 7.** *Let $\pi^\tau = \mathcal{A}^s(\pi_\phi, \Lambda, \Delta, \tau)$. Suppose all the assumptions in Proposition (5) hold. Suppose the LICQ and the strict complementary slackness condition (SCSC) (Giorgi & Zuccotti, 2018; Xu & Zhu, 2023a) for the optimization problem (3.1) holds at $\pi^\tau$. Then, the Lagrangian multipliers $u_{c_i,\tau}^*(\pi_\phi)$ is unique for any given $\pi_\phi$, $\nabla_\phi u_{c_i,\tau}^*(\pi_\phi)$ exists. For $i \in \{1, \cdots, p\}$, if $u_{c_i,\tau}^*(\pi_\phi) = 0$, then $\nabla_\phi u_{c_i,\tau}^*(\pi_\phi) = 0$. Let $\bar{u}_{c_i,\tau}^*(\pi_\phi)$ be the vector includes all all $i \in \{1, \cdots, p\}$ with $u_{c_i,\tau}^*(\pi_\phi) > 0$,*

$$\nabla_\phi u_{c_i,\tau}^*(\pi_\phi) = -\nabla_\phi \nabla_{\bar{u}} \hat{L}(\bar{u}, \phi) \nabla_{\bar{u}}^2 \hat{L}(\bar{u}, \phi)^{-1}$$

*where $\hat{L}(\bar{u}, \phi) = \mathbb{E}[A_\tau^{\pi_\phi}(s,a) - \sum_{i=1}^p u_i A_{c_i,\tau}^{\pi_\phi}(s,a)] - \eta^u \mathbb{E}_{s \sim \nu_\tau^{\pi_\phi}} [D_{KL}(\pi^u(\cdot|s)\|\pi_\phi(\cdot|s))] + \sum_{i=1}^p u_i(d_{i,\tau} + \delta_{c_i} - J_{c_i,\tau}(\pi_\phi)).$*

*Proof.* For any meta-policy $\pi_\phi$, the objective function of the optimization problem (3.1) is strongly concave and the constraint function is convex. The LICQ and the SCSC hold at $\pi^\tau$. According to Theorem 2 in (Xu & Zhu, 2023a), the Lagrangian multipliers $u_{c_i,\tau}^*(\pi_\phi)$ is unique for any given $\pi_\phi$ and $\nabla_\phi u_{c_i,\tau}^*(\pi_\phi)$ exists. The computation is shown in (Xu & Zhu, 2023a). For all $i \in \{1, \cdots, p\}$ with $u_{c_i,\tau}^*(\pi_\phi) = 0$, we have $\nabla_\phi u_{c_i,\tau}^*(\pi_\phi) = 0$.

$\square$

### F.4 OPTIMALITY AND CONSTRAINT SATISFACTION ANALYSIS

#### F.4.1 LEMMAS FOR OPTIMALITY AND SAFE ANALYSIS

**Lemma 4.** *Suppose that Assumption 2 holds. For any task $\tau$, and any safe policies $\pi$ and $\pi' \in \{\pi \in \Pi : J_{c_i,\tau}(\pi) \leq d_i + \delta_{max}, \forall i = 1, \cdots, p\}$, the following bound holds:*

$$\frac{1}{1-\gamma} \mathbb{E}_{\substack{s \sim \nu_\tau^\pi \\ a \sim \pi'(\cdot|s)}} [A_\tau^\pi(s,a)] - C_\tau^\pi(\pi') \leq J_\tau(\pi') - J_\tau(\pi) \leq \frac{1}{1-\gamma} \mathbb{E}_{\substack{s \sim \nu_\tau^\pi \\ a \sim \pi'(\cdot|s)}} [A_\tau^\pi(s,a)] + C_\tau^\pi(\pi') \quad (30)$$

*where*

$$C_\tau^\pi(\pi') = \frac{8\gamma \max_{s,a} A_\tau^\pi(s,a)}{\alpha(1-\gamma)^2} D_{TV}^{max}(\pi||\pi')\mathbb{E}_{s\sim\nu_\tau^\pi, s\in\mathcal{S}^v}\left[D_{TV}(\pi(\cdot|s)||\pi'(\cdot|s))\right].$$

*Here, we define* $D_{TV}(\pi(\cdot|s)||\pi'(\cdot|s)) \triangleq \frac{1}{2}\sum_{a\in\mathcal{A}}|\pi(a|s) - \pi'(a|s)|$ *and* $D_{TV}^{max}(\pi||\pi') \triangleq \max_{s\in\mathcal{S}^v} D_{TV}(\pi(\cdot|s)||\pi'(\cdot|s))$.

*The inequalities (30) also holds for each* $i = 1, \cdots, p$, *when* $A_\tau^\pi$ *and* $A_\tau^{\pi'}$ *are replaced by* $A_{c_i,\tau}^\pi$ *and* $A_{c_i,\tau}^{\pi'}$, $\max_{s,a} A_\tau^\pi(s,a)$ *is replaced by* $\max_{s,a} A_{c_i,\tau}^\pi(s,a)$, $J_\tau$ *is replaced by* $J_{c_i,\tau}$.

*Proof.* The proof follows similar lines of Theorem 1 in (Schulman et al., 2015a) and Corollary 1 and 2 in (Achiam et al., 2017). For the sake of self-containedness, we provide the complete proof.

Let $P_\tau^\pi$ is a matrix where $P_\tau^\pi(i,j) = \mathbb{E}_{a\sim\pi(\cdot|s_i)}P_\tau(s_j|s_i,a)$ and $P_\tau^{\pi'}$ is a matrix where $P_\tau^{\pi'}(i,j) = \mathbb{E}_{a\sim\pi'(\cdot|s_i)}P_\tau(s_j|s_i,a)$. Let $G = (1 + \gamma P_\tau^\pi + (\gamma P_\tau^\pi)^2 + \ldots) = (1 - \gamma P_\tau^\pi)^{-1}$, and similarly $\tilde{G} = (1 + \gamma P^{\pi'_\tau} + (\gamma P^{\pi'_\tau})^2 + \ldots) = (1 - \gamma P^{\pi'_\tau})^{-1}$. Let $\rho$ be a density vector on state space and $r_\tau$ is a reward function vector on state space, thus $r_\tau^\top \rho$ is a scalar meaning the expected reward under density $\rho$. Note that $J_\tau(\pi) = r_\tau^\top G\rho_\tau$, and $J_\tau(\pi') = r_\tau^\top \tilde{G}\rho_\tau$. Here, $\rho_\tau$ is the initial state distribution for task $\tau$. Let $\Delta = P_\tau^{\pi'} - P_\tau^\pi$.

Follow the proof in Appendix B in (Schulman et al., 2015a), we have

$$G^{-1} - \tilde{G}^{-1} = (1 - \gamma P_\pi) - (1 - \gamma P_{\tilde{\pi}}) = \gamma\Delta.$$

Left multiply by $\tilde{G}$ and right multiply by $G$,

$$\tilde{G} = \gamma\tilde{G}\Delta G + G. \tag{31}$$

Left multiply by $G$ and right multiply by $\tilde{G}$,

$$\tilde{G} = \gamma G\Delta\tilde{G} + G. \tag{32}$$

Substituting the right-hand side in (31) into $\tilde{G}$ in (32), then

$$\tilde{G} = G + \gamma G\Delta G + \gamma^2 G\Delta\tilde{G}\Delta G.$$

So we have

$$J_\tau(\pi') - J_\tau(\pi) = r_\tau^\top(\tilde{G} - G)\rho_\tau = \gamma r_\tau^\top G\Delta G\rho_\tau + \gamma^2 r_\tau^\top G\Delta\tilde{G}\Delta G\rho_\tau. \tag{33}$$

Note that $r_\tau^\top G = v_\tau^{\pi\top}$, where $v$ is the value function on the state space. We also have $G\rho_\tau = \frac{1}{1-\gamma}\nu_\tau^\pi$, where $\nu_\tau^\pi$ is the state visitation distribution vector. So,

$$J_\tau(\tilde{\pi}) - J_\tau(\pi) = r_\tau^\top(\tilde{G} - G)\rho_\tau = \frac{\gamma}{1-\gamma}v_\tau^{\pi\top}\Delta\nu_\tau^\pi + \frac{\gamma^2}{1-\gamma}v_\tau^{\pi\top}\Delta\tilde{G}\Delta\nu_\tau^\pi.$$

Consider the first term $\frac{\gamma}{1-\gamma}v_\tau^{\pi\top}\Delta\nu_\tau^\pi$, similar to Equation (50) in (Schulman et al., 2015a), we have

$$\begin{aligned}
\gamma v_\tau^{\pi\top}\Delta\nu_\tau^\pi &= v_\tau^{\pi\top}(P_\tau^{\pi'} - P_\tau^\pi)\nu_\tau^\pi \\
&= \sum_s \nu_\tau^\pi(s)\sum_{s'}\sum_a(\pi'(a|s) - \pi(a|s))P_\tau(s'|s,a)\gamma v_\tau^\pi(s') \\
&= \sum_s \nu_\tau^\pi(s)\sum_a(\pi'(a|s) - \pi(a|s))\left[r(s) + \sum_{s'}P_\tau(s'|s,a)\gamma v_\tau^\pi(s') - v(s)\right] \\
&= \sum_s \nu_\tau^\pi(s)\sum_a(\pi'(a|s) - \pi(a|s))A_\tau^\pi(s,a)
\end{aligned} \tag{34}$$

Since we have $\sum_a \pi(a|s)A_\tau^\pi(s,a) = 0$, we have

$$\gamma v_\tau^{\pi\top}\Delta\nu_\tau^\pi = \sum_s \nu_\tau^\pi(s)\sum_a\pi'(a|s)A_\tau^\pi(s,a) = \mathop{\mathbb{E}}_{\substack{s\sim\nu_\tau^\pi \\ a\sim\pi'(\cdot|s)}}\left[A_\tau^\pi(s,a)\right].$$

Combine (33) and the above equation, we have the following for the second term:

$$\frac{\gamma^2}{1-\gamma} v_\tau^{\pi\top} \Delta \tilde{G} \Delta \nu_\tau^\pi = J_\tau(\pi') - J_\tau(\pi) - \frac{1}{1-\gamma} \mathop{\mathbb{E}}_{\substack{s \sim \nu_\tau^\pi \\ a \sim \pi'(\cdot|s)}} [A_\tau^\pi(s,a)].$$

Then we need to show

$$\left| \frac{\gamma^2}{1-\gamma} v_\tau^{\pi\top} \Delta \tilde{G} \Delta \nu_\tau^\pi \right| \le C_\tau^\pi(\pi').$$

First,

$$\left| \frac{\gamma^2}{1-\gamma} v_\tau^{\pi\top} \Delta \tilde{G} \Delta \nu_\tau^\pi \right|$$

$$\le \left| \frac{\gamma^2}{1-\gamma} \left( v_\tau^{\pi\top} \Delta \right)_{\mathcal{S}^v} \left( \tilde{G} \Delta \nu_\tau^\pi \right)_{\mathcal{S}^v} \right| + \left| \frac{\gamma^2}{1-\gamma} \left( v_\tau^{\pi\top} \Delta \right)_{\mathcal{S}/\mathcal{S}^v} \left( \tilde{G} \Delta \nu_\tau^\pi \right)_{\mathcal{S}/\mathcal{S}^v} \right|$$

By Hölder's inequality,

$$\left| \frac{\gamma^2}{1-\gamma} v_\tau^{\pi\top} \Delta \tilde{G} \Delta \nu_\tau^\pi \right| \le \frac{\gamma}{1-\gamma} \|\gamma v_\tau^{\pi\top} \Delta\|_\infty \|\tilde{G} \Delta \nu_\tau^\pi\|_1.$$

Similar to (34), each element in the vector $\gamma v_\tau^{\pi\top} \Delta$ is $\sum_a (\pi'(a|s) - \pi(a|s)) A_\tau^\pi(s,a)$, then we have

$$\left\| \gamma \left( v_\tau^{\pi\top} \Delta \right)_{\mathcal{S}^v} \right\|_\infty \le \max_{s \in \mathcal{S}^v} \sum_a |\pi'(a|s) - \pi(a|s)| A_\tau^\pi(s,a) \le 2 \max_{s,a} A_\tau^\pi(s,a) D_{TV}^{max}(\pi||\pi').$$

$$\left\| \gamma \left( v_\tau^{\pi\top} \Delta \right)_{\mathcal{S}/\mathcal{S}^v} \right\|_\infty \le \max_{s \in \mathcal{S}/\mathcal{S}^v} \sum_a |\pi'(a|s) - \pi(a|s)| A_\tau^\pi(s,a) \le 4 \max_{s,a} A_\tau^\pi(s,a).$$

From the Lemma 3 of (Achiam et al., 2017), we have

$$\|\tilde{G} \Delta \nu_\tau^\pi\|_1 \le \frac{2}{1-\gamma} \mathbb{E}_{s \sim \nu_\tau^\pi} [D_{TV}(\pi(\cdot|s)||\pi'(\cdot|s))].$$

Therefore, we have

$$\left| \frac{\gamma^2}{1-\gamma} v_\tau^{\pi\top} \Delta \tilde{G} \Delta \nu_\tau^\pi \right|$$

$$\le \frac{4\gamma \max_{s,a} A_\tau^\pi(s,a)}{(1-\gamma)^2} \left( D_{TV}^{max} \left( \pi||\pi' \right) \mathbb{E}_{s \sim \nu_\tau^\pi, s \in \mathcal{S}^v} [D_{TV}(\pi(\cdot|s)||\pi'(\cdot|s))] \right.$$

$$\left. + \frac{2(1-\alpha)}{\alpha} D_{TV}^{max}(\pi||\pi') \mathbb{E}_{s \sim \nu_\tau^\pi, s \in \mathcal{S}^v} [D_{TV}(\pi(\cdot|s)||\pi'(\cdot|s))] \right)$$

$$\le \frac{8\gamma \max_{s,a} A_\tau^\pi(s,a)}{\alpha(1-\gamma)^2} D_{TV}^{max}(\pi||\pi') \mathbb{E}_{s \sim \nu_\tau^\pi, s \in \mathcal{S}^v} [D_{TV}(\pi(\cdot|s)||\pi'(\cdot|s))]$$

Then the bounds hold.

$\square$

**Lemma 5** (Restatement of Lemma 1). *Suppose that Assumption 2 holds. For any task $\tau$, and any safe policies $\pi, \pi' \in \{\pi \in \Pi : J_{c_i,\tau}(\pi) \le d_i + \delta_{max}, \ \forall i = 1, \cdots, p\}$, the following bound holds:*

$$J_\tau(\pi') - J_\tau(\pi) \le \frac{1}{1-\gamma} \mathop{\mathbb{E}}_{\substack{s \sim \nu_\tau^\pi \\ a \sim \pi'(\cdot|s)}} [A_\tau^\pi(s,a)] + \frac{4\gamma A^{max}}{\eta\alpha(1-\gamma)^2} \mathbb{E}_{s \sim \nu_\tau^\pi} [D_{KL}(\pi'(\cdot|s)||\pi(\cdot|s))]$$

*and*

$$J_\tau(\pi') - J_\tau(\pi) \ge \frac{1}{1-\gamma} \mathop{\mathbb{E}}_{\substack{s \sim \nu_\tau^\pi \\ a \sim \pi'(\cdot|s)}} [A_\tau^\pi(s,a)] - \frac{4\gamma A^{max}}{\eta\alpha(1-\gamma)^2} \mathbb{E}_{s \sim \nu_\tau^\pi} [D_{KL}(\pi'(\cdot|s)||\pi(\cdot|s))].$$

*These two inequalities also holds for each $i = 1, \cdots, p$, when $A_\tau^\pi$ and $A_\tau^{\pi'}$ are replaced by $A_{c_i,\tau}^\pi$ and $A_{c_i,\tau}^{\pi'}$, $A^{max}$ is replaced by $A_{c_i}^{max}$, $J_\tau$ is replaced by $J_{c_i,\tau}$.*

Lemma 5 is a variant of Theorem 1 in (Schulman et al., 2015a) and Corollary 1 and 2 in (Achiam et al., 2017). The difference is that, under Assumption 2, the inequalities in Lemma 5 replace the term $\max_s D_{KL}(\pi'(\cdot|s)||\pi(\cdot|s)$ in Theorem 1 in (Schulman et al., 2015a) and replace the term $\sqrt{\mathbb{E}_{s\sim\nu_\tau^\pi}\left[D_{KL}(\pi'(\cdot|s)||\pi(\cdot|s))\right]}$ in Corollary 1 and 2 in (Achiam et al., 2017) by $\mathbb{E}_{s\sim\nu_\tau^\pi}\left[D_{KL}(\pi'(\cdot|s)||\pi(\cdot|s))\right]$ in the right-hand side of the inequalities.

*Proof.* We show the first inequality. The second inequality follows a similar way. From Lemma 4,

$$J_\tau(\pi') - J_\tau(\pi) - \frac{1}{1-\gamma}\mathop{\mathbb{E}}_{\substack{s\sim\nu_\tau^\pi \\ a\sim\pi'(\cdot|s)}}\left[A_\tau^\pi(s,a)\right]$$

$$\leq \frac{8\gamma\max_{s,a}A_\tau^\pi(s,a)}{\alpha(1-\gamma)^2}D_{TV}^{max}(\pi||\pi')\mathbb{E}_{s\sim\nu_\tau^\pi,s\in\mathcal{S}^v}\left[D_{TV}(\pi(\cdot|s)||\pi'(\cdot|s))\right].$$

From Assumption 2, for any safe policy $\pi$, we have $\nu_\tau^\pi(s) \geq \eta$ for all $s \in \mathcal{S}^v$, then we have $\eta D_{TV}^{max}(\pi||\pi') \leq \mathbb{E}_{s\sim\nu_\tau^\pi}\left[D_{TV}(\pi(\cdot|s)||\pi'(\cdot|s))\right]$, i.e.,

$$D_{TV}^{max}(\pi||\pi') \leq \frac{1}{\eta}\mathbb{E}_{s\sim\nu_\tau^\pi,s\in\mathcal{S}^v}\left[D_{TV}(\pi(\cdot|s)||\pi'(\cdot|s))\right].$$

Then, we have

$$\mathbb{E}_{s\sim\nu_\tau^\pi,s\in\mathcal{S}^v}\left[D_{TV}(\pi(\cdot|s)||\pi'(\cdot|s))\right]^2 \leq \mathbb{E}_{s\sim\nu_\tau^\pi,s\in\mathcal{S}^v}\left[D_{TV}^2(\pi(\cdot|s)||\pi'(\cdot|s))\right]$$

$$\leq \mathbb{E}_{s\sim\nu_\tau^\pi,s\in\mathcal{S}^v}\left[D_{TV}^2(\pi(\cdot|s)||\pi'(\cdot|s))\right].$$

From the above three inequalities, we have

$$J_\tau(\pi') - J_\tau(\pi) - \frac{1}{1-\gamma}\mathop{\mathbb{E}}_{\substack{s\sim\nu_\tau^\pi \\ a\sim\pi'(\cdot|s)}}\left[A_\tau^\pi(s,a)\right] \leq \frac{8\gamma A^{max}}{\eta\alpha(1-\gamma)^2}\mathbb{E}_{s\sim\nu_\tau^\pi}\left[D_{TV}^2(\pi(\cdot|s)||\pi'(\cdot|s))\right]. \quad (35)$$

From (Csiszár & Körner, 2011), we have

$$D_{TV}^2(\pi(\cdot|s)||\pi'(\cdot|s)) \leq \frac{1}{2}D_{KL}(\pi'(\cdot|s)||\pi(\cdot|s)).$$

Therefore,

$$J_\tau(\pi') - J_\tau(\pi) \leq \frac{1}{1-\gamma}\mathop{\mathbb{E}}_{\substack{s\sim\nu_\tau^\pi \\ a\sim\pi'(\cdot|s)}}\left[A_\tau^\pi(s,a)\right] + \frac{4\gamma A^{max}}{\eta\alpha(1-\gamma)^2}\mathbb{E}_{s\sim\nu_\tau^\pi}\left[D_{KL}(\pi'(\cdot|s)||\pi(\cdot|s))\right]$$

$\square$

### F.4.2 PROOF OF PROPOSTION 4

*Proof of Propostion 4.* From Lemma 1, we have

$$J_\tau(\pi) \leq J_\tau(\pi_\phi) + \mathbb{E}_{s\sim\nu_\tau^\pi,a\sim\pi(\cdot|s)}\left[\frac{A_\tau^{\pi_\phi}(s,a)}{1-\gamma}\right] + \frac{4\gamma A^{max}}{\eta\alpha(1-\gamma)^2}\mathbb{E}_{s\sim\nu_\tau^{\pi_\phi}}\left[D_{KL}(\pi(\cdot|s)||\pi_\phi(\cdot|s))\right]$$

Since $\lambda_{c_i} \geq \frac{4\gamma A_{c_i}^{max}}{\eta\alpha(1-\gamma)^2}$, we have

$$J_{c_i,\tau}(\pi^\tau)$$

$$\leq J_{c_i,\tau}(\pi_\phi) + \mathop{\mathbb{E}}_{\substack{s\sim\nu_\tau^{\pi_\phi} \\ a\sim\pi^\tau(\cdot|s)}}\left[\frac{A_{c_i,\tau}^{\pi_\phi}(s,a)}{1-\gamma}\right] + \frac{4\gamma A_{c_i}^{max}}{\eta\alpha(1-\gamma)^2}\mathbb{E}_{s\sim\nu_\tau^{\pi_\phi}}\left[D_{KL}(\pi^\tau(\cdot|s)||\pi_\phi(\cdot|s))\right]$$

$$\leq J_{c_i,\tau}(\pi_\phi) + \mathop{\mathbb{E}}_{\substack{s\sim\nu_\tau^{\pi_\phi} \\ a\sim\pi^\tau(\cdot|s)}}\left[\frac{A_{c_i,\tau}^{\pi_\phi}(s,a)}{1-\gamma}\right] + \lambda_{c_i}\mathbb{E}_{s\sim\nu_\tau^{\pi_\phi}}\left[D_{KL}(\pi^\tau(\cdot|s)||\pi_\phi(\cdot|s))\right]$$

$$\leq d_{i,\tau} + \delta_{c_i}.$$

Also, we have

$$J_\tau(\pi) \geq J_\tau(\pi_\phi) + \mathbb{E}_{s\sim\nu_\tau^\pi, a\sim\pi(\cdot|s)}\left[\frac{A_\tau^{\pi_\phi}(s,a)}{1-\gamma}\right] - \frac{4\gamma A^{max}}{\eta\alpha(1-\gamma)^2}\mathbb{E}_{s\sim\nu_\tau^{\pi_\phi}}\left[D_{KL}(\pi(\cdot|s)||\pi_\phi(\cdot|s))\right]$$

Since $\lambda \geq \frac{4\gamma A^{max}}{\eta\alpha(1-\gamma)}$, we have

$$J_\tau(\pi) \geq J_\tau(\pi_\phi) + \mathbb{E}_{s\sim\nu_\tau^\pi, a\sim\pi(\cdot|s)}\left[\frac{A_\tau^{\pi_\phi}(s,a)}{1-\gamma}\right] - \frac{\lambda}{1-\gamma}\mathbb{E}_{s\sim\nu_\tau^{\pi_\phi}}\left[D_{KL}(\pi(\cdot|s)||\pi_\phi(\cdot|s))\right].$$

For the solution $\pi^\tau$ of problem (1), we have

$$J_\tau(\pi^\tau) \geq \mathop{\mathbb{E}}_{\substack{s\sim\nu_\tau^{\pi_\phi}\\a\sim\pi^\tau(\cdot|s)}}\left[\frac{A_\tau^{\pi_\phi}(s,a)}{1-\gamma}\right] - \frac{\lambda}{1-\gamma}\mathbb{E}_{s\sim\nu_\tau^{\pi_\phi}}\left[D_{KL}\left(\pi^\tau(\cdot|s)||\pi_\phi(\cdot|s)\right)\right] + J_\tau(\pi_\phi)$$

$$= \max_{\pi\in\Pi_\tau^C}\mathop{\mathbb{E}}_{\substack{s\sim\nu_\tau^{\pi_\phi}\\a\sim\pi(\cdot|s)}}\left[\frac{A_\tau^{\pi_\phi}(s,a)}{1-\gamma}\right] - \frac{\lambda}{1-\gamma}\mathbb{E}_{s\sim\nu_\tau^{\pi_\phi}}\left[D_{KL}\left(\pi(\cdot|s)||\pi_\phi(\cdot|s)\right)\right] + J_\tau(\pi_\phi)$$

$$\geq \mathop{\mathbb{E}}_{\substack{s\sim\nu_\tau^{\pi_\phi}\\a\sim\pi_\phi(\cdot|s)}}\left[\frac{A_\tau^{\pi_\phi}(s,a)}{1-\gamma}\right] - \frac{\lambda}{1-\gamma}\mathbb{E}_{s\sim\nu_\tau^{\pi_\phi}}\left[D_{KL}\left(\pi_\phi(\cdot|s)||\pi_\phi(\cdot|s)\right)\right] + J_\tau(\pi_\phi) = J_\tau(\pi_\phi).$$

where $\Pi_\tau^C$ is the feasible set of problem (1). The last inequality comes from $\pi^\phi \in \Pi_\tau^C$. $\qquad\square$

### F.4.3 PROOF OF THEOREM 1

Recall the notations defined in Section 5.2 and used in this section: the optimal task-specific policy $\pi_*^\tau$ for task $\tau$ as
$$\pi_*^\tau \triangleq \operatorname{argmax}_{\pi\in\Pi} J_\tau(\pi) \text{ s.t. } J_{c_i,\tau}(\pi) \leq d_{i,\tau};$$
the conservative task-specific optimal policy $\pi_{*,[\epsilon]}^\tau$, which is optimal for $\tau$ under conservative safety constraints, i.e.,
$$\pi_{*,[\epsilon]}^\tau \triangleq \operatorname{argmax}_{\pi\in\Pi} J_\tau(\pi) \text{ s.t. } J_{c_i,\tau}(\pi) \leq d_{i,\tau} - \epsilon,$$
where the conservative constant $\epsilon \geq 0$; the task variance
$$\mathcal{V}ar(\mathbb{P}(\Gamma)) \triangleq \min_\phi \mathbb{E}_{\tau\sim\mathbb{P}(\Gamma)}\mathbb{E}_{s\sim\nu_\tau^{\pi_\phi}}[D_{KL}(\pi_*^\tau(\cdot|s)||\pi_\phi(\cdot|s))];$$
the task variance under the conservative safety constraints
$$\mathcal{V}ar^\epsilon(\mathbb{P}(\Gamma)) \triangleq \min_\phi \mathbb{E}_{\tau\sim\mathbb{P}(\Gamma)}\mathbb{E}_{s\sim\nu_\tau^{\pi_\phi}}[D_{KL}(\pi_{*,[\epsilon]}^\tau(\cdot|s)||\pi_\phi(\cdot|s))],$$
and its minimal point
$$\hat{\phi}^{[\epsilon]} \triangleq \arg\min_\phi \mathbb{E}_{\tau\sim\mathbb{P}(\Gamma)}\mathbb{E}_{s\sim\nu_\tau^{\pi_\phi}}[D_{KL}(\pi_{*,[\epsilon]}^\tau(\cdot|s)||\pi_\phi(\cdot|s))],$$
the radius of the task distribution $\mathbb{P}(\Gamma)$
$$R(\mathbb{P}(\Gamma)) \triangleq \max_{\tau\in\Gamma,\epsilon\in E} \mathbb{E}_{s\sim\nu_\tau^{\pi_{\hat{\phi}^{[\epsilon]}}}}[D_{KL}(\pi_{*,[\epsilon]}^\tau(\cdot|s)||\pi_{\hat{\phi}^{[\epsilon]}}(\cdot|s))],$$
where the set $E$ is defined by $\{\epsilon \geq 0 : \pi_{*,[\epsilon]}^\tau \text{ exists for all } \tau \in \Gamma\}$.

We also define
$$R^{[\epsilon]}(\mathbb{P}(\Gamma)) \triangleq \max_{\tau\in\Gamma} \mathbb{E}_{s\sim\nu_\tau^{\pi_{\hat{\phi}^{[\epsilon]}}}}[D_{KL}(\pi_{*,[\epsilon]}^\tau(\cdot|s)||\pi_{\hat{\phi}^{[\epsilon]}}(\cdot|s))].$$

We first show some lemmas for the proof of Theorem 1.

**Lemma 6.** *Suppose that Assumption 2 holds. For any $\epsilon$, The policy $\pi_{*,[\epsilon]}^\tau$ belongs to the set* $\left\{\pi\in\Pi: J_{c_i,\tau}(\pi_{\hat{\phi}^{[\epsilon]}}) + \mathop{\mathbb{E}}_{\substack{s\sim\nu_\tau^{\pi_{\hat{\phi}^{[\epsilon]}}}\\a\sim\pi(\cdot|s)}}\left[\frac{A_{c_i,\tau}^{\pi_{\hat{\phi}^{[\epsilon]}}}(s,a)}{1-\gamma}\right] + \frac{4\gamma A_{c_i}^{max}}{\eta\alpha(1-\gamma)^2}\mathbb{E}_{s\sim\nu_\tau^{\pi_{\hat{\phi}^{[\epsilon]}}}}[D_{KL}(\pi(\cdot|s)||\pi_{\hat{\phi}^{[\epsilon]}}(\cdot|s))]\right.$

$\left. \leq d_{c_i,\tau} - \epsilon + \frac{8\gamma A_{c_i}^{max}}{\eta\alpha(1-\gamma)^2}R(\mathbb{P}(\Gamma)) \text{ for all } i = 1,\cdots,p\right\}$

*Proof.* From the second inequality in Lemma 1, $J_{c_i,\tau}(\pi_{*,[\epsilon]}^{\tau}) \geq$

$$J_{c_i,\tau}(\pi_{\hat{\phi}[\epsilon]}) + \frac{1}{1-\gamma} \mathop{\mathbb{E}}_{\substack{s \sim \nu_\tau^{\pi_{\hat{\phi}[\epsilon]}} \\ a \sim \pi_{*,[\epsilon]}^{\tau}(\cdot|s)}} \left[ A_{c_i,\tau}^{\pi_{\hat{\phi}[\epsilon]}}(s,a) \right] - \frac{4\gamma A_{c_i}^{max}}{\eta\alpha(1-\gamma)^2} \mathop{\mathbb{E}}_{s \sim \nu_\tau^{\pi_{\hat{\phi}[\epsilon]}}} \left[ D_{KL}(\pi_{*,[\epsilon]}^{\tau}(\cdot|s) || \pi_{\hat{\phi}[\epsilon]}(\cdot|s)) \right].$$

Since $J_{c_i,\tau}(\pi_{*,[\epsilon]}^{\tau}) \leq d_{c_i,\tau} - \epsilon$, we have

$$J_{c_i,\tau}(\pi_{\hat{\phi}[\epsilon]}) + \frac{1}{1-\gamma} \mathop{\mathbb{E}}_{\substack{s \sim \nu_\tau^{\pi_{\hat{\phi}[\epsilon]}} \\ a \sim \pi_{*,[\epsilon]}^{\tau}(\cdot|s)}} \left[ A_{c_i,\tau}^{\pi_{\hat{\phi}[\epsilon]}}(s,a) \right]$$

$$- \frac{4\gamma A_{c_i}^{max}}{\eta\alpha(1-\gamma)^2} \mathop{\mathbb{E}}_{s \sim \nu_\tau^{\pi_{\hat{\phi}[\epsilon]}}} \left[ D_{KL}(\pi_{*,[\epsilon]}^{\tau}(\cdot|s) || \pi_{\hat{\phi}[\epsilon]}(\cdot|s)) \right] \leq d_{c_i,\tau} - \epsilon.$$

Then,

$$J_{c_i,\tau}(\pi_{\hat{\phi}[\epsilon]}) + \mathop{\mathbb{E}}_{\substack{s \sim \nu_\tau^{\pi_{\hat{\phi}[\epsilon]}} \\ a \sim \pi_{*,[\epsilon]}^{\tau}(\cdot|s)}} \left[ \frac{A_{c_i,\tau}^{\pi_{\hat{\phi}[\epsilon]}}(s,a)}{1-\gamma} \right] + \frac{4\gamma A_{c_i}^{max}}{\eta\alpha(1-\gamma)^2} \mathop{\mathbb{E}}_{s \sim \nu_\tau^{\pi_{\hat{\phi}[\epsilon]}}} \left[ D_{KL}(\pi_{*,[\epsilon]}^{\tau}(\cdot|s) || \pi_{\hat{\phi}[\epsilon]}(\cdot|s)) \right]$$

$$\leq d_{c_i,\tau} - \epsilon + \frac{8\gamma A_{c_i}^{max}}{\eta\alpha(1-\gamma)^2} \mathop{\mathbb{E}}_{s \sim \nu_\tau^{\pi_{\hat{\phi}[\epsilon]}}} \left[ D_{KL}(\pi_{*,[\epsilon]}^{\tau}(\cdot|s) || \pi_{\hat{\phi}[\epsilon]}(\cdot|s)) \right]$$

$$\leq d_{c_i,\tau} - \epsilon + \frac{8\gamma A_{c_i}^{max}}{\eta\alpha(1-\gamma)^2} R^{[\epsilon]}(\mathbb{P}(\Gamma))$$

$$\leq d_{c_i,\tau} - \epsilon + \frac{8\gamma A_{c_i}^{max}}{\eta\alpha(1-\gamma)^2} R(\mathbb{P}(\Gamma)).$$

$\square$

**Lemma 7.** *Suppose that Assumption 2 holds. We have*

$$\pi_{\hat{\phi}[\epsilon]} \in \left\{ \pi \in \Pi : J_{c_i,\tau}(\pi) \leq d_{c_i,\tau} - \epsilon + \frac{8\gamma A_{c_i}^{max}}{\eta\alpha(1-\gamma)^2} R(\mathbb{P}(\Gamma)) \text{ for all } i = 1, \cdots, p \text{ and } \tau \in \Gamma \right\}.$$

*Proof.* From the second inequality in Lemma 1, $J_{c_i,\tau}(\pi_{*,[\epsilon]}^{\tau}) \geq$

$$J_{c_i,\tau}(\pi_{\hat{\phi}[\epsilon]}) + \frac{1}{1-\gamma} \mathop{\mathbb{E}}_{\substack{s \sim \nu_\tau^{\pi_{\hat{\phi}[\epsilon]}} \\ a \sim \pi_{*,[\epsilon]}^{\tau}(\cdot|s)}} \left[ A_{c_i,\tau}^{\pi_{\hat{\phi}[\epsilon]}}(s,a) \right] - \frac{4\gamma A_{c_i}^{max}}{\eta\alpha(1-\gamma)^2} \mathop{\mathbb{E}}_{s \sim \nu_\tau^{\pi_{\hat{\phi}[\epsilon]}}} \left[ D_{KL}(\pi_{*,[\epsilon]}^{\tau}(\cdot|s) || \pi_{\hat{\phi}[\epsilon]}(\cdot|s)) \right].$$

Since $J_{c_i,\tau}(\pi_{*,[\epsilon]}^{\tau}) \leq d_{c_i,\tau} - \epsilon$, we have

$$J_{c_i,\tau}(\pi_{\hat{\phi}[\epsilon]}) + \frac{1}{1-\gamma} \mathop{\mathbb{E}}_{\substack{s \sim \nu_\tau^{\pi_{\hat{\phi}[\epsilon]}} \\ a \sim \pi_{*,[\epsilon]}^{\tau}(\cdot|s)}} \left[ A_{c_i,\tau}^{\pi_{\hat{\phi}[\epsilon]}}(s,a) \right]$$

$$\leq d_{c_i,\tau} - \epsilon + \frac{4\gamma A_{c_i}^{max}}{\eta\alpha(1-\gamma)^2} \mathop{\mathbb{E}}_{s \sim \nu_\tau^{\pi_{\hat{\phi}[\epsilon]}}} \left[ D_{KL}(\pi_{*,[\epsilon]}^{\tau}(\cdot|s) || \pi_{\hat{\phi}[\epsilon]}(\cdot|s)) \right].$$

Also, from (34) and the proof of Lemma 1, we have

$$\frac{1}{1-\gamma} \mathop{\mathbb{E}}_{\substack{s \sim \nu_\tau^{\pi_{\hat{\phi}[\epsilon]}} \\ a \sim \pi_{*,[\epsilon]}^{\tau}(\cdot|s)}} \left[ A_{c_i,\tau}^{\pi_{\hat{\phi}[\epsilon]}}(s,a) \right]$$

$$= \frac{1}{1-\gamma} \sum_s \nu_\tau^{\pi_{\hat{\phi}[\epsilon]}}(s) \sum_a (\pi_{*,[\epsilon]}^{\tau}(a|s) - \pi_{\hat{\phi}[\epsilon]}(a|s)) A_{c_i,\tau}^{\pi_{\hat{\phi}[\epsilon]}}(s,a)$$

$$\leq \frac{4 A_{c_i}^{max}}{\eta\alpha(1-\gamma)} D_{TV}^{max}(\pi_{*,[\epsilon]}^{\tau}(\cdot|s) || \pi_{\hat{\phi}[\epsilon]}(\cdot|s))^2$$

$$\leq \frac{4 A_{c_i}^{max}}{\eta\alpha(1-\gamma)} \mathop{\mathbb{E}}_{s \sim \nu_\tau^{\pi_{\hat{\phi}[\epsilon]}}} \left[ D_{KL}(\pi_{*,[\epsilon]}^{\tau}(\cdot|s) || \pi_{\hat{\phi}[\epsilon]}(\cdot|s)) \right]$$

Then,

$$J_{c_i,\tau}(\pi_{\hat{\phi}^{[\epsilon]}}) + \frac{1}{1-\gamma} \mathbb{E}_{\substack{\pi_{\hat{\phi}^{[\epsilon]}} \\ s \sim \nu_\tau \\ a \sim \pi^\tau_{*,[\epsilon]}(\cdot|s)}} \left[ A^{\pi_{\hat{\phi}^{[\epsilon]}}}_{c_i,\tau}(s,a) \right]$$

$$\leq d_{c_i,\tau} - \epsilon + \frac{4A^{max}_{c_i}}{\eta\alpha(1-\gamma)^2} \mathbb{E}_{s \sim \nu_\tau^{\pi_{\hat{\phi}^{[\epsilon]}}}} \left[ D_{KL}(\pi^\tau_{*,[\epsilon]}(\cdot|s)||\pi_{\hat{\phi}^{[\epsilon]}}(\cdot|s)) \right]$$

$$\leq d_{c_i,\tau} - \epsilon + \frac{8\gamma A^{max}_{c_i}}{\eta\alpha(1-\gamma)^2} R^{[\epsilon]}(\mathbb{P}(\Gamma))$$

$$\leq d_{c_i,\tau} - \epsilon + \frac{8\gamma A^{max}_{c_i}}{\eta\alpha(1-\gamma)^2} R(\mathbb{P}(\Gamma)).$$

Here, we assume $\gamma \geq 0.5$, which is commonly used.

$\square$

**Theorem 2.** *Suppose that Assumptions 1 and 2 hold. Let $\pi^\tau(\hat{\phi}^{[\epsilon]}) = \mathcal{A}^s(\pi_{\hat{\phi}^{[\epsilon]}}, \Lambda, \Delta, \tau)$ with $\lambda = \frac{4\gamma A^{max}}{\eta\alpha(1-\gamma)}$, $\lambda_{c_i} = \frac{2\gamma 4 A^{max}_{c_i}}{\eta\alpha(1-\gamma)^2}$ and $\delta_{c_i} = \frac{4\gamma 4 A^{max}_{c_i}}{\eta\alpha(1-\gamma)^2} R(\mathbb{P}(\Gamma)) - \epsilon$. We have*

$$\mathbb{E}_{\tau \sim \mathbb{P}(\Gamma)}[J_\tau(\pi^\tau_{*,[\epsilon]}) - J_\tau(\mathcal{A}^s(\pi_{\phi^*}, \Lambda, \Delta, \tau))] \leq \frac{8\gamma A^{max}}{\eta\alpha(1-\gamma)^2} Var^\epsilon(\mathbb{P}(\Gamma)).$$

*Proof.* From Lemma 6, we have that $\pi^\tau_{*,[\epsilon]} \in \Pi^B$, where $\Pi^B \triangleq \{\pi \in \Pi : J_{c_i,\tau}(\pi_{\hat{\phi}^{[\epsilon]}}) + \frac{1}{1-\gamma} \mathbb{E}_{\substack{\pi_{\hat{\phi}^{[\epsilon]}} \\ s \sim \nu_\tau \\ a \sim \pi(\cdot|s)}} \left[ A^{\pi_{\hat{\phi}^{[\epsilon]}}}_{c_i,\tau}(s,a) \right] + \lambda_{c_i,\tau} \mathbb{E}_{s \sim \nu_\tau^{\pi_{\hat{\phi}^{[\epsilon]}}}} [D_{KL}(\pi(\cdot|s)||\pi_{\hat{\phi}^{[\epsilon]}}(\cdot|s))] \leq d_{c_i,\tau} + \delta_{c_i}, \forall i\}$.

Also, $\pi^\tau(\hat{\phi}^{[\epsilon]}) \in \Pi^B$. Therefore, from the definition of $\mathcal{A}^s$ in problem (1), we have

$$\mathbb{E}_{\substack{\pi_{\hat{\phi}^{[\epsilon]}} \\ s \sim \nu_\tau \\ a \sim \pi^\tau(\hat{\phi}^{[\epsilon]})(\cdot|s)}} \left[ A^{\pi_{\hat{\phi}^{[\epsilon]}}}_\tau(s,a) \right] - \lambda \bar{D}_{KL}(\pi^\tau(\hat{\phi}^{[\epsilon]}), \pi_{\hat{\phi}^{[\epsilon]}}) \geq \mathbb{E}_{\substack{\pi_{\hat{\phi}^{[\epsilon]}} \\ s \sim \nu_\tau \\ a \sim \pi^\tau_{*,[\epsilon]}(\cdot|s)}} \left[ A^{\pi_{\hat{\phi}^{[\epsilon]}}}_\tau(s,a) \right] - \lambda \bar{D}_{KL}(\pi^\tau_{*,[\epsilon]}, \pi_{\hat{\phi}^{[\epsilon]}}),$$

where we use $\bar{D}_{KL}(\pi_1(\cdot|s), \pi_2(\cdot|s))$ to represent $\mathbb{E}_{s \sim \nu_\tau^{\pi_2}}[D_{KL}(\pi_1(\cdot|s), \pi_2(\cdot|s))]$.

From the second inequality in Lemma 1 and the above inequality,

$$J_\tau(\pi^\tau(\hat{\phi}^{[\epsilon]})) - J_\tau(\pi_{\hat{\phi}^{[\epsilon]}}) \geq \frac{1}{1-\gamma} \mathbb{E}_{\substack{\pi_{\hat{\phi}^{[\epsilon]}} \\ s \sim \nu_\tau \\ a \sim \pi^\tau(\hat{\phi}^{[\epsilon]})(\cdot|s)}} \left[ A^{\pi_{\hat{\phi}^{[\epsilon]}}}_\tau(s,a) \right] - \frac{\lambda}{1-\gamma} \bar{D}_{KL}(\pi^\tau(\hat{\phi}^{[\epsilon]}), \pi_{\hat{\phi}^{[\epsilon]}})$$

$$\geq \frac{1}{1-\gamma} \mathbb{E}_{\substack{\pi_{\hat{\phi}^{[\epsilon]}} \\ s \sim \nu_\tau \\ a \sim \pi^\tau_{*,[\epsilon]}(\cdot|s)}} \left[ A^{\pi_{\hat{\phi}^{[\epsilon]}}}_\tau(s,a) \right] - \frac{\lambda}{1-\gamma} \bar{D}_{KL}(\pi^\tau_{*,[\epsilon]}, \pi_{\hat{\phi}^{[\epsilon]}}).$$

From the first inequality in Lemma 1,

$$J_\tau(\pi^\tau_{*,[\epsilon]}) - J_\tau(\pi_{\hat{\phi}^{[\epsilon]}}) \leq \frac{1}{1-\gamma} \mathbb{E}_{\substack{\pi_{\hat{\phi}^{[\epsilon]}} \\ s \sim \nu_\tau \\ a \sim \pi^\tau_{*,[\epsilon]}(\cdot|s)}} \left[ A^{\pi_{\hat{\phi}^{[\epsilon]}}}_\tau(s,a) \right] + \frac{4\gamma A^{max}}{\eta\alpha(1-\gamma)^2} \bar{D}_{KL}(\pi^\tau_{*,[\epsilon]}, \pi_{\hat{\phi}^{[\epsilon]}}).$$

From the last two inequalities,

$$J_\tau(\pi^\tau(\hat{\phi}^{[\epsilon]})) - J_\tau(\pi^\tau_{*,[\epsilon]}) \geq -(\frac{4\gamma A^{max}}{\eta\alpha(1-\gamma)^2} + \frac{\lambda}{1-\gamma}) \bar{D}_{KL}(\pi^\tau_{*,[\epsilon]}, \pi_{\hat{\phi}^{[\epsilon]}}),$$

i.e.,

$$J_\tau(\pi^\tau_{*,[\epsilon]}) - J_\tau(\mathcal{A}^s(\pi_{\hat{\phi}^{[\epsilon]}}, \Lambda, \Delta, \tau)) \leq (\frac{4\gamma A^{max}}{\eta\alpha(1-\gamma)^2} + \frac{\lambda}{1-\gamma}) \bar{D}_{KL}(\pi^\tau_{*,[\epsilon]}, \pi_{\hat{\phi}^{[\epsilon]}}).$$

Then,

$$\mathbb{E}_{\tau \sim \mathbb{P}(\Gamma)}[J_\tau(\pi_{*,[\epsilon]}^\tau) - J_\tau(\mathcal{A}^s(\pi_{\hat{\phi}^{[\epsilon]}}, \Lambda, \Delta, \tau))]$$

$$\leq (\frac{4\gamma A^{max}}{\eta\alpha(1-\gamma)^2} + \frac{\lambda}{1-\gamma})\mathbb{E}_{\tau \sim \mathbb{P}(\Gamma)}[\bar{D}_{KL}(\pi_{*,[\epsilon]}^\tau, \pi_{\hat{\phi}^{[\epsilon]}})]$$

$$= (\frac{2\gamma A^{max}}{\eta\alpha(1-\gamma)^2} + \frac{\lambda}{1-\gamma})\mathcal{V}ar^\epsilon(\mathbb{P}(\Gamma)).$$

Moreover, from Lemma 7,

$$\pi_{\hat{\phi}^{[\epsilon]}} \in \Pi^C \triangleq \{\pi \in \Pi : J_{c_i,\tau}(\pi) \leq d_{c_i,\tau} + \delta_{c_i} \text{ for all } i = 1, \cdots, p \text{ and } \tau \in \Gamma\}.$$

From the definition of $\phi^*$, we have

$$\mathbb{E}_{\tau \sim \mathbb{P}(\Gamma)}[J_\tau(\mathcal{A}^s(\pi_{\phi^*}, \Lambda, \Delta, \tau))] \geq \max_{\pi \in \Pi^C} \mathbb{E}_{\tau \sim \mathbb{P}(\Gamma)}[J_\tau(\mathcal{A}^s(\pi, \Lambda, \Delta, \tau))]$$

$$\geq \mathbb{E}_{\tau \sim \mathbb{P}(\Gamma)}[J_\tau(\mathcal{A}^s(\pi_{\hat{\phi}^{[\epsilon]}}, \Lambda, \Delta, \tau))]$$

Then, we have

$$\mathbb{E}_{\tau \sim \mathbb{P}(\Gamma)}[J_\tau(\pi_{*,[\epsilon]}^\tau) - J_\tau(\mathcal{A}^s(\pi_{\phi^*}, \Lambda, \Delta, \tau))]$$

$$\leq \mathbb{E}_{\tau \sim \mathbb{P}(\Gamma)}[J_\tau(\pi_{*,[\epsilon]}^\tau) - J_\tau(\mathcal{A}^s(\pi_{\hat{\phi}^{[\epsilon]}}, \Lambda, \Delta, \tau))]$$

$$\leq (\frac{\gamma A^{max}}{\eta\alpha(1-\gamma)^2} + \frac{\lambda}{1-\gamma})\mathcal{V}ar^\epsilon(\mathbb{P}(\Gamma))$$

$$\leq \frac{8\gamma A^{max}}{\eta\alpha(1-\gamma)^2}\mathcal{V}ar^\epsilon(\mathbb{P}(\Gamma)).$$

$\square$

*Proof of Theorem 1.* Theorem 1 is proven by combining Theorem 2 with Corrolary 1. $\square$

# G  LIMITATIONS AND FUTURE WORKS

In this paper, we consider the safety metric of CMDP, i.e., the expected accumulated costs satisfy the given safety threshold. This metric is generally less rigorous than the safe control research, where safety is defined as persistently satisfying certain state constraints. A future work is establishing the safe meta-RL algorithm with the rigorous safety metric. Another limitation is that we assume the solution of problem (2) exists, i.e., there exists a policy such that it is safe for all tasks as the initial policy for policy adaptation steps. A future work is to release this assumption and identify a safe task-specific meta-policy for each given task.

