# OpenReview forum: "Safe Meta-Reinforcement Learning via Dual-Method-Based Policy Adaptation: Near-Optimality and Anytime Safety Guarantee"
_ICLR.cc/2025/Conference — Submitted to ICLR 2025_

### Official Review · Reviewer_bErq · 2024-10-17

**Soundness:** 1
**Presentation:** 2
**Contribution:** 2
**Rating:** 3
**Confidence:** 4

**Summary:**

This paper proposes an algorithm for a safe meta-reinforcement learning problem. Specifically, the proposed algorithm ensures anytime safety during the meta-test, which consists of safe policy adaptation and safe meta-policy training modules. Theoretically, the authors prove the anytime safety guarantee of policy adaptation and show that the obtained policy is near-optimal. The authors' empirical experiments show that the proposed algorithm performs better than the baseline methods.

**Strengths:**

- This paper addresses an important and interesting problem. The motivation behind this problem is well-presented and easy to understand.
- The proposed algorithm is technically sound. It is easy to follow the deviation of the algorithm.
- The experiment has been well designed and the results are good. This paper sufficiently covers the necessary baseline methods including state-of-the-art methods. The benchmark tasks are Gym and Safety-Gymnasium, which are quite standard safe RL literature.

**Weaknesses:**

- I have a serious concern about the theoretical results in Section 5. Specifically, I guess the authors have some misunderstanding regarding sufficient safe visits or ergodicity.
- This paper is on safe RL, which implicitly means that there is a set of states that cannot be visited (frequently). This also means that the ergodicity assumption does not hold in safe RL tasks. If I understand correctly, the authors seem to cite Moldovan and Abbeel (2012) as a piece of evidence that CMDP is ergodic in line 376. Unfortunately, however, in Moldovan and Abbeel (2012), there are opposite statements such as

    > Almost all proposed exploration techniques presume ergodicity; authors present it as a harmless technical assumption but it rarely holds in interesting practical problems.

    > Unfortunately, many environments are not ergodic.

- Related to the above, I disagree with Remark 1 and Remark 2. If the authors address standard MDP, the remarks would be true. It is not true with CMDPs or safe RL. Safe RL studies should not assume ergodicity and thus actually consider "reachability" or "returnability" as in
    - Turchetta, Matteo, Felix Berkenkamp, and Andreas Krause. "Safe exploration in finite markov decision processes with gaussian processes." Advances in neural information processing systems 29 (2016).
    - Wachi, Akifumi, and Yanan Sui. "Safe reinforcement learning in constrained markov decision processes." International Conference on Machine Learning. PMLR, 2020.
- Even worse, the authors try to propose an algorithm with almost surely. Hence, I feel Assumption 2 is incompatible with the nature of the proposed algorithm.
- The authors may want to argue that Assumption 2 still holds by setting a large $B$, but I guess such a large $B$ will lead to useless bounds on both safety and optimality.
- The authors may also want to insist that it is ok to guarantee safety only during meta-tests. However, I do not think it is reasonable to assume sufficient coverage "for any policy and any state."

**Suggestions**
- I read through the proofs of the theorems, but a large portion is strongly built on Assumption 2. Given Section 5 is a core contribution of this paper, I think this is a serious mistake and I do not think that it can be fixed during the rebuttal period.
- For the next submission, I recommend the authors to change Assumption 2 in two points.
    - I think "for all policy $\pi$" is too strong. If I were an author, I would try to make an assumption with "for a safe policy".
    - Also, "for all state $s$" is not reasonable. I would make an assumption characterized by a (safe) subset of state space.

**Questions:**

- Q1: Please tell me your thoughts about my comments in Weakness.

- Q2: Is it possible to relax Assumption 2 while maintaining the claims in Theorems?

---

> ### Author Response · Authors · 2024-11-20
> **Response**
>
> We are grateful and indebted for the reviewer's time and effort invested to evaluate our manuscript, and for all the suggestions and reference recommendations to make our manuscript a better and stronger contribution. We answer the weaknesses and questions as follows.
>
> >**Weakness 1. Safe RL studies should not assume ergodicity and thus actually consider "reachability" or "returnability" as in (Turchetta, 2016) and (Wachi, 2020). The authors try to propose an algorithm with almost surely. Hence, I feel Assumption 2 is incompatible with the nature of the proposed algorithm.**
>
> **Answer:** Papers (Turchetta, 2016) and (Wachi, 2020) study a safe RL scenario, which aims to develop a policy such that any state $s_t$ on the trajectory is safe, i.e., the cost $c(s_t)$  is small than a constant for any timestep $t$. In these two papers, the algorithms with the almost surely property are proposed, i.e., the probability of visiting an unsafe state is smaller than $\epsilon$.
>
> This manuscript, including the algorithm design and the theoretical results (shown in Section 5 in Theorem 1), studies the constrained MDP where the safety is defined as that the expected accumulated costs $J\_{c\_i,\tau}(\pi)=\mathbb{E}\_\pi [\sum\_{t} \gamma^t c(s\_t)]$ are smaller than a threshold. Therefore, it allows for a safe policy to visit some unsafe states (the states with high costs), as long as the expectation of the accumulated costs along many trajectories is smaller than a threshold. Therefore, we are not trying to propose an algorithm with the almost surely property. As a result, this manuscript does not require some assumptions in papers (Turchetta, 2016) and (Wachi, 2020), such as reachability and the regularity of the constraint function $c(s)$. Moreover, as the proposed method is a model-free method, i.e., the knowledge about the cost $c(s)$ on state $s$ cannot be obtained until the state $s$ is visited, we require the assumption of the state visitation. Therefore, the proposed algorithm is compatible with our assumptions.
>
> >**Weakness 2. The weaknesses related to the validness of Assumption 2 and Remarks 1 and 2.**
>
> **Answer:** We agree that the ergodicity may be a strict assumption in the safe RL problem and Assumption 2 may require a large $B$. Thanks for your comments and suggestions.
>
> In the revised manuscript, we relax Assumption 2 to be compatible with the safe RL setting. Following your suggestion, the relaxed assumption only assumes the safe policy has sufficient visitation to a subset of safe space. Here is the relaxed assumption.
>
> **Relaxed assumption:**
>
> >>There exists a set of states $\mathcal{S}^{v} \subseteq \mathcal{S}$ and a constant $\eta > 0$ such that, for any task $\tau \in \Gamma$ and any safe policy $\pi^{s} \in$ { $ \pi \in \Pi: J\_{c_i,\tau}(\pi) \leq d_i + \delta_{max}, \forall i = 1, \cdots, p$ },
> $\nu^{\pi^s}_{\tau}(s) \geq \eta $ for all $s \in \mathcal{S}^{v}$.
>
> The new assumption supposes that there exists a set of states $\mathcal{S}^{v}$ such that the safe policy can take sufficient visitation in the set.
>
> Under the new assumption, we rewrite the proofs for Section 5, including Lemma 1, Proposition 4, Corollary 1, and Theorem 1.
> Overall, with the new Assumption 2, the theoretical results remain unchanged except for some constants. In the proofs for the theoretical results in Section 5, Assumption 2 and the constant $B$ in Assumption 2 (in the old manuscript) are only used when proving Proposition 4, and the proofs of all remaining theoretical results are built on Proposition 4. Therefore, in the revised manuscript, based on the new assumption, we build a new Proposition 4, which uses the constant $\eta$ and $\alpha$ (in the new assumption) to replace the constant $B$. Then, we almost keep other proofs unchanged.
>
> Thanks for the suggestion again. Please refer to Section 5 and Appendix F.4 in the revised manuscript for the details of the theoretical results and proofs.
>
> >**Question 2. Is it possible to relax Assumption 2 while maintaining the claims in Theorems?**
>
> **Answer:** Yes. As stated in the answer to Weakness 2, we relax Assumption 2 and maintain the theoretical results unchanged except for some constants.

---

> > ### Comment · Reviewer_bErq · 2024-11-24
> > **Response**
> >
> > **Ergodicity.** Since the initial review, I have fully understood that this paper focuses on expected cumulative safety constraints. However, it is important to note that ergodicity should not be assumed in the context of safe RL. While Turchetta et al. (2016) and Wachi (2020) provide accessible examples, the general principle holds: ergodicity is not a valid assumption for safe RL in this setting. Specifically, if the safety cost $c(s,a)$ exceeds the available safety budget (i.e., the safety threshold minus the cumulative safety cost), it is clear that ergodicity cannot hold.
> >
> > **Experimental Results.** The experimental results presented in the paper are not aligned with the theoretical framework. In particular, there is a mismatch between the assumptions in the theory and the experimental settings. For consistency and to strengthen the paper's argument, it is essential to adjust the experimental setup so that it aligns with the newly introduced Assumption 2.
> >
> > **Conclusion.** As I noted in my initial review, I do not believe the issues raised in this paper can be addressed within the rebuttal period. After reviewing the relevant literature on safe RL, carefully examining the theoretical assumptions and results, and evaluating the experimental design, I conclude that this paper should be evaluated in a different review cycle. Therefore, I will maintain my original rating of 3: Reject.

---

> > > ### Author Response · Authors · 2024-11-25
> > >
> > > Thanks for your reply.
> > >
> > > In terms of Assumption 2, in the revised manuscript, we no longer assume the ergodicity and only make an assumption on safe policies and a subset of the state space. We rewrite all the theorems and their corresponding proofs under the new relaxed assumption.
> > >
> > > In terms of experimental results, our experiments are aligned with the theoretical framework and the new assumption 2. Specifically, assumption 2 only serves as an indicator to select the hyperparameter $\lambda$ in Theorem 1. After Assumption 2 is replaced by the new relaxed assumption, only the constant $\lambda$ in Theorem 1 is changed and all the problem settings are not changed. In the experiments, the agent can visit any state in a subset of the state space, which matches the relaxed Assumption 2. Therefore, there is no mismatch between the new assumption and the experimental settings.

---

> ### Comment · Reviewer_bErq · 2024-11-26
> **Final response**
>
> First of all, please note that this is the **discussion** phase. I do not think it is appropriate for authors to focus on only the reviewers' mistakes while ignoring the paper's weaknesses. In the first rebuttal, it was unclear whether the authors admitted their misunderstandings on ergodicity. I believe it is better to accept mistakes and discuss constructively to improve the paper.
>
> **Ergodicity.** It was confirmed that the errors have been corrected in the revised version of the paper.
>
> **Experimental settings.** I am not talking about hyperparameter $\lambda$. I have been discussing how the dataset was collected. The current implementation is based on the old assumptions, and unsafe policies collected a large amount of unsafe trajectories with sufficient coverage. This is not consistent with the current theoretical results.
>
> **Regarding the comments by Reviewer 9TCU.** The degree of novelty may elicit different opinions from different people, but I understand the perspective of Reviewer 9TCU. The proposed algorithm can be seen as an incremental extension of CRPO.
>
> **Conclusion.** Again, this paper has suffered from initial critical errors resulting from unreasonable assumptions, which previously led to unreasonable theoretical results and now result in inconsistent empirical analyses. I do not believe this paper can be ready for publication during this discussion phase. Therefore, I will maintain my original rating of 3: Reject considering that this paper should be evaluated in a different review cycle.

---

> > ### Author Response · Authors · 2024-11-26
> > **Thanks very much for the suggestions**
> >
> > We sincerely appreciate your effort and time in reviewing our paper, as well as your invaluable suggestions. We admit that we had a misunderstanding about ergodicity in the first submitted version, and we fully acknowledge the importance of the issue you raised. The suggestions and insights about the modification (Suggestion 2) are instrumental in improving our work for future submissions. We fully understand and respect your decision based on this issue and are grateful for your constructive feedback. Thanks again for your thoughtful and thorough review.

---

### Official Review · Reviewer_TzFn · 2024-11-02

**Soundness:** 3
**Presentation:** 2
**Contribution:** 3
**Rating:** 6
**Confidence:** 3

**Summary:**

This paper considers the safe meta-reinforcement learning (meta-RL) problem, and proposes a novel meta-RL algorithm that achieves 1) superior optimality, 2) anytime safety, and 3) high computational efficiency. The proposed algorithm is shown theoretically to achieve monotonic improvement, near optimality, and anytime safety. Experimental results are provided to support the 3 claims.

**Strengths:**

The considered problem is important. The proposed algorithm has 3 key advantages including superior optimality, anytime safety, and high computational efficiency. The key advantages are justified both theoretically and empirically.

**Weaknesses:**

1. My most important concern about this paper is its writing.

    - While the proposed algorithm is strong, no intuition is provided when introducing the algorithm structure, making the flow of the paper not smooth.

    - The connection between the proposed approach and Meta-CRPO and Meta-CPO is not clear. Although this is briefly discussed in the appendix, I suggest connecting the proposed method with the previous approaches also when introducing the new algorithm. This makes the readers much easier to understand what the key difference in the algorithm is that makes the proposed method perform better.

    - There is some ambiguity about some terms introduced in the paper. Please see Questions for details.

2. The experimental results can be improved. The influence of the hyperparameters is not discussed or tested, and how to select the hyperparameters is unclear.

**Questions:**

1. Line 69 "Both meta-CRPO and meta-CPO provide positive upper bounds of the constraint violation". Do the authors mean $d_i > 0$ or meta-CRPO and meta-CPO cannot satisfy the constraint that the sum of $c_i$ less than $d_i$?

1. Line 129: There is a sum over $a'\in\mathcal A$ when defining the softmax policy. Does it mean that the paper only considers discrete action space?

2. Line 137: Should the $J_{c_i, \tau}(\pi)$ be $J_{c_i}(\pi)$?

3. Line 192: it is said that setting $\delta_{c_i} = 0$ for all $i$ is too strict, and to alleviate the issue, the paper set $\delta_{c_i}\geq 0$. However, in the hyperparameters provided in Table 2, why $\delta_c$ is set to $0$ in all environments?

4. Line 210: "Safety cannot be guaranteed in each step". What is the definition of "step" here?

5. Line 245: "The complete statement of Proposition 3 that...". Is this Proposition 1 instead? Same question for Line 246 "Proposition 3 shows that..."

6. Line 324: "Note that the meta-gradient in (6) does not include the computations of Hessian and inverse of Hessian w.r.t. $\phi$". Could the authors clarify the reason why the meta-gradient in (6) avoids the Hessian and what is the cost of not using Hessian?

7. Does assumption 2 imply that the state space considered in the paper needs to be discrete and finite?

8. Can the proposed algorithm work if the max accumulated cost constraint is set to $0$?

---

> ### Author Response · Authors · 2024-11-20
> **Response (1/3)**
>
> We are grateful and indebted for the reviewer's time and effort invested to evaluate our manuscript, and for all the suggestions and reference recommendations to make our manuscript a better and stronger contribution. We answer the weaknesses and questions as follows.
>
> >**Weakness 1.1. While the proposed algorithm is strong, no intuition is provided when introducing the algorithm structure, making the flow of the paper not smooth.**
>
> **Answer:** Thanks for the comments. In the revised manuscript, we add several explanations after the proposed algorithms are introduced. Please refer to the revised manuscript for the details and the added sentences are highlighted in red color.
>
> For example, after the safe policy optimization algorithm (1), we add the intuition of the algorithm design: "The safe policy adaptation $\mathcal{A}^{s}$ in problem (1) is inspired by the derivation of CPO, where both problem (1) and CPO aim to approximate the original safe RL problem. Specifically, the objective and constraint functions of problem (1) serve as upper bounds of the true objective and constraint functions $J_\tau(\pi)$ and $J_{c_i,\tau}(\pi)$ of the safe RL problem."
>
> >**Weakness 1.2. The connection between the proposed approach and Meta-CRPO and Meta-CPO is not clear. Although this is briefly discussed in the appendix, I suggest connecting the proposed method with the previous approaches also when introducing the new algorithm. This makes the readers much easier to understand what the key difference in the algorithm is that makes the proposed method perform better.**
>
> **Answer:** Thanks for the suggestions. In the revised manuscript, we add the discussion that connects the proposed method with Meta-CRPO and Meta-CPO, in Section 3.1 and Section 4.1. Please refer to the revised manuscript for the details and the added sentences are highlighted in red color.
>
> >**Weakness 2. The experimental results can be improved. The influence of the hyperparameters is not discussed or tested, and how to select the hyperparameters is unclear.**
>
> **Answer:** Thanks for the suggestions. In the revised manuscript (Appendix D.4 and Figure 6), we test the experimental results with different hyper-parameter settings, i.e. different allowable constraint violation constant $\delta_{c_i}$, on two environments, including Half-cheetah and Car-Circle-Hazard.
>
> Moreover, we also include guidance about how to choose the hyper-parameters, including $\delta_{c_i}$, $\lambda$, and $\lambda_{c_i}$ in Appendix D.4 of the revised manuscript. The guidance is presented as follows.
>
> Guidance of selecting $\delta_{c_i}$: As indicated in both theoretical results in Section 5.2 and the experimental results in Figure 6, we choose a large $\delta_{c_i}$ when the constraint satisfaction is not required to be strict, and a small $\delta_{c_i} \rightarrow 0$ when the constraint satisfaction is prioritized.
>
> Guidance of selecting $\lambda$ and $\lambda_{c_i}$: We set $\lambda=\lambda_{c_i}$ and tune them such that, the KL divergence of initial policy $\pi$ and the adapted policy $\pi^\prime$ solved from the safe policy adaptation problem (1) is close to $0.03$. If the KL divergence is too large, the objective and constraint functions of problem (1) are not good approximations of the accumulated reward/cost functions, as indicated by Lemma 1. If the KL divergence is too small, the policy adaptation step of problem (1) is too small.

---

> ### Author Response · Authors · 2024-11-20
> **Response (2/3)**
>
> >**Question 1. Line 69 "Both meta-CRPO and meta-CPO provide positive upper bounds of the constraint violation". Do the authors mean that meta-CRPO and meta-CPO cannot satisfy the constraint that the sum of $c_i$ is less than $d_i$?**
>
> **Answer:** Yes. In meta-CRPO and meta-CPO, the constraint violation converges to zero as the number of policy optimization steps becomes sufficiently large or when the KL divergence between the initial policy and the adapted policy is sufficiently small. Consequently, the $\sum_t c_i(t) - d_i$ tends to zero. However, there is no guarantee that it is always smaller than zero.
>
> >**Question 2. Line 129: There is a sum over $a^{\prime} \in \mathcal{A}$ when defining the softmax policy. Does it mean that the paper only considers discrete action space?**
>
> **Answer:** Thanks for your question. In this manuscript, the action space $\mathcal{A}$ could be either discrete or continuous.
> The state space $\mathcal{S}$ could be either a discrete space or a bounded continuous space. In the revised manuscript, we have clarified it and modified the definitions, theorems, and proofs that are not compatible with it, including the definition of the softmax policy.
>
> In the experiments, all the environments have continuous state space and action space.
>
> >**Question 3. Line 137: Should the $J_{c_i, \tau}(\pi)$ be $J_{c_i}(\pi)$ ?**
>
> **Answer:** Yes, thanks for pointing it out.
>
> >**Question 4. Line 192: it is said that setting $\delta_{c_i}=0$ for all $i$ is too strict, and to alleviate the issue, the paper set $\delta_{c_i}=0$. However, in the hyperparameters provided in Table 2, why $\delta_{c_i}=0$ is set to  in all environments?**
>
> **Answer:** Thanks for the question. The statement should be modified to "When the requirement of the constraint satisfaction is not strict, setting $\delta_{c_i}=0$ for all $i$ in problem (1) may overly restrict the policy update step. To enhance the algorithm’s flexibility, we set $\delta_{c_i} \geq 0$ as an allowable constraint violation in problem (1). "
>
> In the experiments, we aim to verify the anytime safe property of the proposed method. Therefore, we set $\delta_{c_i}=0$ such that $J_{c_i,\tau}(\pi) \leq d_{i,\tau}$ always holds for the adapted policies $\pi$.
>
> To compare the experimental results under different values of $\delta_{c_i}$, in the revised manuscript (Appendix D.4 and Figure 6), we add the experiments on two environments, including Half-cheetah and Car-Circle-Hazard.
>
> >**Question 5. Line 210: "Safety cannot be guaranteed in each step". What is the definition of "step" here?**
>
> **Answer:** The policy optimization algorithm, such as (1), requires the data collection by a single policy, i.e., the initial policy $\pi_\phi$, and produces the adapted policy $\mathcal{A}^{s}(\pi_\phi, \Lambda, \Delta, \tau)$. This is one step of the policy adaptation.
>
> In the manuscript, we consider the anytime property of the policy optimization algorithm, i.e., any policy used to explore the environment should be safe. This anytime property requires that each step of the policy optimization algorithm should be safe, because the output policy from each step of the policy optimization algorithm needs to be safe.
>
>  We have included the above definition of "step" in the revised manuscript (Section 3.1, lines 187-189).

---

> > ### Author Response · Authors · 2024-11-20
> > **Response (3/3)**
> >
> > **Question 6. Line 245: "The complete statement of Proposition 3 that...". Is this Proposition 1 instead? Same question for Line 246 "Proposition 3 shows that..." **
> >
> > **Answer:** Yes, thanks for pointing it out. We have corrected the statement in the revised manuscript (line 252).
> >
> > >**Question 7. Line 324: "Note that the meta-gradient in (6) does not include the computations of Hessian and inverse of Hessian w.r.t. $\phi$." Could the authors clarify the reason why the meta-gradient in (6) avoids the Hessian and what is the cost of not using Hessian?**
> >
> > **Answer:** In many existing methods, such as MAML and meta-CPO, the computation of meta-gradient includes the computations of Hessian and inverse of Hessian w.r.t. $\phi$. For example, in MAML,
> > the adapted policy $\pi_\theta$ is obtained by one-step policy gradient ascent $\theta=\phi+ \alpha \nabla_\phi J_\tau(\pi_\phi)$. The meta-gradient, i.e., the gradient of the meta-objective $\nabla_\phi J_\tau(\phi+ \alpha \nabla_\phi J_\tau(\pi_\phi))$, includes the computations of Hessian.
> >
> > In this manuscript, we derive the closed-form solution of the policy adaptation problem $\mathcal{A}^{s}$ in (1), as shown in Proposition 1. Specifically, the adapted policy $\pi_\tau$ has an analytical expression $\pi_\tau=g(\phi)$. The meta-gradient is $\nabla_\phi J_\tau(g(\phi))$. Therefore it avoids the Hessian in the meta-gradient computation. The meta-gradient holds a comparable computational complexity as the policy gradient.
> >
> > >**Question 8. Does assumption 2 imply that the state space considered in the paper needs to be discrete and finite?**
> >
> > **Answer:** Thanks for the question. In this manuscript, the state space $\mathcal{S}$ could be either a discrete space or a bounded continuous space. If the state space $\mathcal{S}$ is discrete, $\nu^\pi_{\tau}(s)$ denotes the visitation probability on $s$. If the state space $\mathcal{S}$ is continuous, $\nu^\pi_{\tau}(s)$ denotes the visitation probability density on $s$.
> >
> > >**Question 9. Can the proposed algorithm framework if the max accumulated cost constraint is set to $0$ ?**
> >
> > **Answer:** The proposed algorithm can work when the maximal accumulated cost constraint is set to $0$. However, the theoretical results in Section 5 may not be valid for the case. In Assumption 1, we assume that the feasible set of problem (2) is not empty, i.e., there exists a softmax policy that is safe for all environments. However, when the maximal accumulated cost constraint is set to $0$, if the cost in some states is larger than $0$, the visitation probability to the state has to be strictly $0$ for a safe policy. Then, there does not exist a softmax policy whose visitation probability to any state is strictly $0$. In this case, Assumption 1 is not satisfied, then the theoretical results in Section 5 may not be valid for the case.

---

> ### Comment · Reviewer_TzFn · 2024-11-28
>
> Thank you for the detailed response and clarifications! The presentation and the rigor of the paper have been improved. I have no further questions.

---

### Official Review · Reviewer_GqFW · 2024-11-04

**Soundness:** 3
**Presentation:** 3
**Contribution:** 3
**Rating:** 6
**Confidence:** 3

**Summary:**

The paper proposed a safe meta RL framework to learn a meta policy which is safe to a new RL task. The key contributions include (i) theoretical analysis and show that anytime safety guarantee can be achieved if the initial policy is safe, (ii) theoretically analyze the tradeoff between safety guarantee and optimality, (iii) empirically validated its outperformance against other meta RL algorithms in computational efficiency and reward / safety performance.

**Strengths:**

1. This paper is a theoretically dense paper and its core strength lies in its theoretical derivations and insights. The paper did a good job in showing the safety guarantee (when initial policy is safe) and Theorem 1 provides key insight on the tradeoff between this safety guarantee and reward optimality.

2. The paper did a thorough study on the shortcomings of other safe meta RL paper and identified the key area which it can improve on (i.e. computational efficiency and anytime safety guarantee.

3. The empirical result shows that it convincingly outperforms other safe meta RL algorithms in terms of computational efficiency and reward / safety performance.

**Weaknesses:**

1. The experiments portion of this paper is relatively short (esp in main paper). I'd think including other experiments would further improve the paper. For example, trying out different values of $\delta_{c_i}$ (allowable constraint violation) and observe the tradeoff between reward and safety.

2. The allowable constraint violation $\delta_{c_i}$ seems like a hyper-parameter and I would appreciate further guidance on how to determine the appropriate value for a safety-constrained task. Perhaps performing experiment suggested in item (1) above could help.

3. The paper does point out the inherent shortcoming of CPO being computationally expensive and proposed a dual method for safe policy adaptation. However, the safe policy adaptation problem outlined in Eq4 & 5 seems rather similar to Lagrangian-based online safe RL algorithm, e.g. RCPO, PPO-Lagrangian. The authors might want to illustrate how is this dual method particularly novel.

4. To achieve anytime safety, the initial policy should already be safe. The paper could further illustrate how this is achieved. In Fig1, safe meta RL seems to start with safe policy in test env while MAML and meta-CRPO don't. I'm curious how this is achieved in practice.

5. In Fig4, meta-CPO (blue dashed line) is not present in the humanoid task. Is there any reason why this method is missing from humanoid task only?

**Questions:**

Please refer to the Weakness section and I'm more than happy to discuss if there's anything I misunderstood or missed out.

---

> ### Author Response · Authors · 2024-11-20
> **Response (1/2)**
>
> Thanks very much for your time and effort in reviewing our work. Thanks for your suggestions to make our manuscript better. We answer the weaknesses and questions as follows.
>
> >**Weakness 1. The experiments portion of this paper is relatively short (esp in the main paper). I think including other experiments would further improve the paper. For example, try out different values of (allowable constraint violation) and observe the trade-off between reward and safety.**
>
> **Answer:** Thanks for the suggestions. In the revised manuscript (Appendix D.4 and Figure 6), we conduct the experiments with different allowable constraint violation constant $\delta_{c_i}$ on two environments, including Half-cheetah and Car-Circle-Hazard. The experiment shows the trade-off between reward and safety.
>
> >**Weakness 2. The allowable constraint violation $\delta_{c_i} seems like a hyper-parameter and I would appreciate further guidance on how to determine the appropriate value for a safety-constrained task. Perhaps performing the experiment suggested in item (1) above could help.**
>
> **Answer:** Thanks for the suggestions. As stated in the answer to Weakness 1, we have added the experiments. The following is the guidance on how to choose the hyper-parameter $\delta_{c_i}$.
>
> As indicated in both theoretical results in Section 5.2 and the experimental results in Figure 6, we choose a large $\delta_{c_i}$ when the constraint satisfaction is not required to be strict, and a small $\delta_{c_i} \rightarrow 0$ when the constraint satisfaction is prioritized.
>
> >**Weakness 3. The paper does point out the inherent shortcoming of CPO being computationally expensive and proposes a dual method for safe policy adaptation. However, the safe policy adaptation problem outlined in Eq. 4 and 5 seems rather similar to Lagrangian-based online safe RL algorithm, e.g. RCPO, PPO-Lagrangian. The authors might want to illustrate how is this dual method particularly novel.**
>
> **Answer:** Eq. (4) and (5) solve the safe policy adaptation problem in (1) by the **dual method**. RCPO and PPO-Lagrangian solve the safe RL algorithm by the **primal-dual** method. Although both the proposed dual method for problem (1) and the primal-dual method in RCPO and PPO-Lagrangian are Lagrangian-based safe policy optimization algorithms, they are different. RCPO and PPO-Lagrangian are not suitable for this safe meta-RL problem and are much worse than the proposed method, even worse than CPO.
>
> Eq. (4) and (5) aim to solve the safe policy adaptation problem in (1). As mentioned in Section 3.1, the safe policy adaptation (1) holds several benefits similar to CPO, including the safety guarantee for a single policy optimization step (using data collected on a single policy) and the monotonic improvement. Moreover, we derive the closed-form solution under certain Lagrangian multipliers for the optimization problem (1). Based on the derived closed-form solution of (1) (shown in (3)), we can use the dual method shown in (4)(5) to solve the safe policy adaptation problem in (1), which significantly reduces the computational complexity during the meta-training.
>
> In contrast, RCPO and PPO-Lagrangian do not hold any of the benefits shown in CPO and the proposed algorithm. First, RCPO and PPO-Lagrangian use the gradient ascent steps on the Lagrangian, which do not have the safety guarantee and the monotonic improvement in each policy optimization step, and therefore cannot guarantee anytime safety in the meta-test stage. Moreover, there is no closed-form solution for the policy optimization step in RCPO and PPO-Lagrangian, and therefore cannot be solved by the dual method, which leads the high computational complexity during the meta-training.
>
> Thanks for the question and the literature recommendation. We have included the above discussion in the revised manuscript (Appendix C).

---

> > ### Author Response · Authors · 2024-11-20
> > **Response (2/2)**
> >
> > >**Weakness 4. To achieve anytime safety, the initial policy should already be safe. The paper could further illustrate how this is achieved. In Fig 1, safe meta RL seems to start with safe policy in test env while MAML and meta-CRPO don't. I'm curious how this is achieved in practice.**
> >
> > **Answer:** The learned meta-policy, which is the starting policy for the meta-test, is designed to be safe. In the proposed safe meta-RL algorithm, the optimization problem in (2) is solved for the meta-training stage. The constraints in the optimization problem of (2) ensure that the solved meta-policy is safe, i.e., the starting policy for the meta-test is safe. In Algorithm 2, we prioritize the minimization of the constraint violation of the meta-policy (shown in lines 10-12 in Algorithm 2). Please refer to Section 3.2 for the details.
> >
> > In contrast, MAML and meta-CRPO do not have such a mechanism in the meta-training. Thus, they cannot guarantee that the initial policy of the meta-test is safe.
> >
> > >**Weakness 5. In Fig 4, meta-CPO (blue dashed line) is not present in the humanoid task. Is there any reason why this method is missing from humanoid tasks only?**
> >
> > **Answer:**
> > Due to the high dimension of the Humanoid tasks, the meta-training of meta-CPO is too slow (10 times slower than the proposed method) in Humanoid tasks. It is extremely time-consuming (over one month) to run the meta-training of meta-CPO multiple times on humanoid tasks and draw its figure. So the result of meta-CPO is not shown in Fig 4. In contrast, the proposed method can deal with the high-dimensional problem.
> >
> >  We have included the above discussion in the revised manuscript (Appendix D.3).

---

> > > ### Comment · Reviewer_GqFW · 2024-11-26
> > >
> > > I thank the authors for their detailed response and the new set of results in the revised manuscript. I have no further questions.

---

### Official Review · Reviewer_9TCU · 2024-11-04

**Soundness:** 2
**Presentation:** 2
**Contribution:** 1
**Rating:** 3
**Confidence:** 4

**Summary:**

This paper investigates the problem of ensuring safety in meta-reinforcement learning (meta-RL) by proposing a framework that guarantees anytime safety during meta-testing. The approach is based on dual-method-based policy adaptation, which includes modules for safe policy adaptation and safe meta-policy training. It provides empirical results showcasing improvements over existing safe meta-RL methods in terms of optimality, safety guarantees, and computational efficiency.

**Strengths:**

1. The method achieves high computational efficiency, making it advantageous for scaling to complex tasks.

2. Empirical results demonstrate that the proposed method outperforms baseline approaches in terms of both optimality and safety across a variety of tasks.

**Weaknesses:**

1. The related work section lacks thorough investigation, e.g., some multi-task/multi-objective safe RL methods; these can be helpful for meta-safe RL.

2. The paper is not well-written and appears to rely heavily on language models for content generation, e.g. the abstract.

3. The method lacks novelty; based on my understanding, it does not present new contributions, including in the theoretical aspects. It extends primal-dual settings for meta-safe RL, similar to primal meta-safe RL (meta-CRPO).

**Questions:**

Why was a dual-method-based approach chosen over other constraint-handling techniques?  I guess that using any state-of-the-art safe RL baseline in meta-RL settings could also achieve good performance.

Could there be advantages to comparing it with alternatives, such as shielded RL?

---

> ### Author Response · Authors · 2024-11-20
> **Response (1/4)**
>
> We are grateful and indebted for the reviewer's time and effort invested to evaluate our manuscript, and for all the suggestions and reference recommendations to make our manuscript a better and stronger contribution. We answer the weaknesses and questions as follows.
>
> >**Weakness 1. The related work section lacks thorough investigation, e.g., some multi-task/multi-objective safe RL methods; these can be helpful for meta-safe RL.**
>
> **Answer:** Although all of meta-safe RL, multi-task safe RL, and multi-objective safe RL consider multiple tasks in safe RL environments, however, the most important distinction between meta-safe RL and multi-task/multi-objective safe RL is that the agent in meta-safe RL is required to adapt to a new and unknown environment under few-shot data collection. Therefore, the policy adaptation algorithm is the most important part of meta-safe RL. This manuscript designs a novel policy adaptation algorithm in (1) which holds several benefits for the few-shot policy adaptation that the existing methods do not hold. In contrast, the multi-task/multi-objective safe RL learns the policies for multiple tasks during the training stage, where the policy adaptation is not required. Therefore, the multi-task/multi-objective can borrow existing policy optimization methods and do not need to design a new one.
>
> Thanks for the comments. We have included the above discussion in Appendix A of the revised manuscript.
>
> >**Weakness 2. The paper is not well-written and appears to rely heavily on language models for content generation, e.g. the abstract.**
>
> **Answer:** We certify that we do not use language models to generate any content of the manuscript. We aim to keep the abstract as concise as possible.

---

> ### Author Response · Authors · 2024-11-20
> **Response (2/4)**
>
> >**Weakness 3. The method lacks novelty; based on my understanding, it does not present new contributions, including in the theoretical aspects. It extends primal-dual settings for meta-safe RL, similar to primal meta-safe RL (meta-CRPO).**
>
> **Answer:** The paper does not extend the **primal-dual method** in meta-CRPO to meta-safe RL. Instead, we design a new policy adaptation algorithm in problem (1) and solve it by the **dual method**. Although the primal-dual method and the dual method are both Lagrangian-based safe policy optimization algorithms, they are different, and the primal-dual method in meta-CRPO is much worse than the proposed dual method in the meta-safe RL problem.
>
> **The differences between the proposed method and meta-CRPO.**
> This paper designs a new policy adaptation algorithm in problem (1) and solves it by the dual method. The proposed algorithm holds (i) a safety guarantee for a single policy optimization step and (ii) a closed-form solution. The proposed policy adaptation algorithm is the first algorithm that simultaneously offers the two key properties.
> Based on the derived closed-form solution, we use the dual method to solve problem (1). Meta-CRPO uses the CRPO, a primal-dual-based method for policy adaptation. The method does not hold any of the two properties in our proposed methods. In particular, it does not have a closed-form solution and cannot be solved by the dual method. It uses the primal-dual method to solve safe RL, which cannot guarantee safety for a single policy optimization step.
> In this paper, the meta-policy training algorithm in (2) aims to maximize the expected accumulated reward of the policies adapted from the meta-policy. We derive a Hessian-free approach to optimize the meta-policy.
> In meta-CRPO, the meta-policy is learned by minimizing the distance between the meta-policy and the task-specific policy, which does not consider the optimality and the safety of the task-specific policies adapted from the meta-policy.
> In the following, we elaborate on the advantages of the proposed methods over existing meta-safe RL methods, including meta-CPO and meta-CRPO.
>
> **The advantages of the proposed method over meta-CRPO.** The proposed algorithms offer three key advantages over existing safe meta-RL methods, including meta-CPO and meta-CRPO.
> (i) **Superior optimality.** Our safe meta-policy training algorithm in (2) maximizes the expected accumulated reward of the policies adapted from the meta-policy. In contrast, the meta-training of meta-CRPO learns the meta-policy by minimizing the distance between the meta-policy and the task-specific policy, which does not consider the optimality and the safety of the task-specific policies adapted from the meta-policy.
> (ii) **Anytime safety guarantee** during the meta-test. The safe meta-policy training produces a safe initial meta-policy by imposing the safety constraint. The safe policy adaptation imposes a constraint on the upper bound of the total cost, and thus is guaranteed to produce a safe policy for each iteration when the initial policy is safe. By integrating these two modules, anytime safety is achieved. In contrast, meta-CRPO employs CRPO, a primal-dual-based method, for policy adaptation, which does not have any safe guarantee in a single policy adaptation step. Thus, anytime safety cannot be guaranteed in meta-CRPO.
> (iii) **High computational efficiency** in both the meta-test and meta-training stages. In the meta-test, the derivation of the close-formed solution of Problem (1) makes it efficient to be solved.
> In contrast, the meta-CRPO and meta-CPO require to solve a constrained optimization problem, which is more computationally expensive than the solution of problem (1). In the meta-training, the close-formed solution of the policy adaptation (1) is used to derive a Hessian-free meta-gradient and reduces the computation complexity of the proposed algorithm to approach that in the single-level optimization.
> In contrast, the meta-CPRO requires that the task-specific optimal policies have been learned, which is impractical when the number of training tasks is large.
> The meta-CPO uses the bi-level optimization to learn the meta-policy, which requires the computation of Hessian and Hessian inverse.
> We conduct experiments on seven scenarios including navigation tasks with collision avoidance and locomotion tasks to verify these advantages of the proposed algorithms.

---

> ### Author Response · Authors · 2024-11-20
> **Response (3/4)**
>
> **The theoretical contribution of the proposed method over meta-CRPO.** In terms of theoretical contribution, the paper is the first to derive a comprehensive theoretical analysis regarding near optimality and anytime safety guarantees for safe meta-RL.
> The theoretical contribution of this paper over meta-CRPO and meta-CPO is shown in Table 1. First, we establish the theoretical basis of the algorithm design that guarantees anytime safety, i.e., zero constraint violation for any policy used for exploration. Second, we derive a lower bound of the expected accumulated reward of the adapted policies compared to that of the task-specific optimal policies, which shows the near optimality of the proposed safe meta-RL framework.
> Finally, we demonstrate a trade-off between the optimality bound and constraint violation when the allowable constraint violation varies, which enables the algorithm to be adjusted to prioritize either safety or optimality.
> In meta-CPRO, the optimality bound is provided, but anytime safety is not guaranteed.
> Meta-CPO provides neither an optimality bound nor anytime safety.

---

> > ### Author Response · Authors · 2024-11-20
> > **Response (4/4)**
> >
> > >**Question 1. Why was a dual-method-based approach chosen over other constraint-handling techniques? I guess that using any state-of-the-art safe RL baseline in meta-RL settings could also achieve good performance.**
> >
> > **Answer:** As we state in the answer to Weakness 3, the proposed method is the first algorithm that simultaneously offers two key advantages, (i) the safety guarantee for every single policy optimization step (using data collected on a single policy) and (ii) holding a closed-form solution, which enables us to use the dual method to reduce the computational complexity of meta-policy training. The existing methods for safe policy optimization, including the primal-dual-based methods, e.g., CRPO, RCPO, PPO-Lagrangian, and the trust-region-based methods, e.g., CPO, do not hold these two advantages simultaneously, and therefore are not suitable for the safe meta-RL problem.
> >
> > We conduct experiments on seven scenarios including navigation tasks with collision avoidance and locomotion tasks to verify these advantages of the proposed algorithms.
> >
> > >**Question 2. Could there be advantages to comparing it with alternatives, such as shielded RL?**
> >
> > **Answer:** In this paper, we consider a safe-meta RL problem. During the meta-test, given an unknown environment with an unknown CMDP, the agent can sample few-shot data from the environment and adapt the policy.
> >
> > In shielded RL, the shield function for the agent is pre-trained for two cases: (i) the MDP is known; (ii) a large amount of data is sampled from an unknown MDP. Therefore, the shielded RL method cannot be used in the safe meta-RL problem.

---

> > > ### Comment · Reviewer_9TCU · 2024-11-25
> > >
> > > Thank you for the authors' response. After reviewing the response, considering other reviewers' comments, and re-checking the paper, I recognize that the study simply extends CRPO, a primal-based method, to meta-RL by replacing the primal optimization with dual optimization (See this [study algorithm 2](https://openreview.net/pdf?id=BbYu1wLwmj#page=6.66) and [CRPO algorithm 1](https://proceedings.mlr.press/v139/xu21a/xu21a.pdf#page=4.39)). I also agree with Reviewer bErq's feedback, and the manuscript overstates the method's performance. Therefore, I will maintain my score of 3: reject.

---

> > > > ### Author Response · Authors · 2024-11-25
> > > > **Response**
> > > >
> > > > Thanks for the reply.
> > > >
> > > > We politely disagree that "the study simply extends CRPO, a primal-based method, to meta-RL by replacing the primal optimization with dual optimization". We propose a new policy optimization algorithm in problem (1), which is specifically designed for the safe meta-RL problem. In particular, problem (1) along with its solution in Algorithm 2 is the first algorithm that holds (i) a safety guarantee for a single policy optimization step and (ii) a closed-form solution. The policy adaptation algorithm of CRPO lacks both a safety guarantee for individual policy optimization steps and a closed-form solution. Consequently, (a) it can not be solved by dual-method, which leads to high computational complexity, and (b) it cannot guarantee the anytime safety in the meta-test.
> > > >
> > > > We also disagree that "the manuscript overstates the method's performance". In terms of the algorithm design, we clearly show our motivations of why the proposed method can be better than the existing methods, i.e., three key advantages of the proposed method including (i) superior optimality, (ii) anytime safety guarantee, and (iii) high computational efficiency. In the experiment, we do fair and complete comparisons with all the existing safe meta-RL algorithms on seven environment settings. The proposed method significantly outperforms the baseline methods.

---

### Meta-Review · Area_Chair_Witb · 2024-12-19

**Metareview:**

Safe Meta-Reinforcement Learning via Dual-Method-Based Policy Adaptation: Near-Optimality and Anytime Safety Guarantee

Summary: The paper focuses on addressing the challenge of anytime safety in meta-reinforcement learning (meta-RL). The proposed framework integrates two modules: safe policy adaptation and safe meta-policy training, enabling the formulation of policies that are nearly optimal while guaranteeing safety constraints during exploration. By employing dual-method algorithms, the paper achieves computational efficiency and anytime safety, surpassing existing methods like Meta-CPO and Meta-CRPO in both theoretical guarantees and experimental results. Experiments across locomotion and navigation tasks validate the framework's performance, including reduced computational complexity and stricter adherence to safety constraints.

Comments: This paper received four expert reviews, with scores 3, 3, 6, 6, and the average score is 4.50. The reviewers acknowledge multiple positive aspects of the paper, including the theoretically grounded approach to address anytime safety in meta-RL. However, the reviewers have concerns about several weaknesses. More than one reviewer pointed out that the proposed dual-method-based approach, while effective, shows significant overlap with existing Lagrangian-based techniques in safe RL (e.g., RCPO, PPO-Lagrangian), and the novelty of the method is not clearly articulated. Reviewer bErq has given critical comments about assumptions used in developing and analyzing the proposed method. More than one reviewer commented that the experimental valuation can be greatly improved, including performing sensitivity analysis on the hyperparameters. One reviewer has also pointed out missing references, including significant contributions in multi-task or multi-objective safe RL, which could provide valuable context.

While this paper has several commendable aspects, such as its focus on anytime safety and computational efficiency, it suffers from a lack of clarity, incomplete theoretical justification, and limited novelty. Addressing these issues would significantly enhance the paper's quality and impact.

**Additional Comments On Reviewer Discussion:**

Please see the "Comments" in the meta-review.

---

### Decision · Program_Chairs · 2025-01-22

Reject